# Dopamine neuron morphology and output are differentially controlled by mTORC1 and mTORC2

Polina Kosillo[1†], Kamran M Ahmed[1†], Erin E Aisenberg[2], Vasiliki Karalis[1], Bradley M Roberts[3], Stephanie J Cragg[3], Helen S Bateup[1,2,4*]

[1]Department of Molecular and Cell Biology, University of California, Berkeley, Berkeley, United States; [2]Helen Wills Neuroscience Institute, University of California, Berkeley, Berkeley, United States; [3]Department of Physiology, Physiology, Anatomy and Genetics, University of Oxford, Oxford, United Kingdom; [4]Chan Zuckerberg Biohub, San Francisco, San Francisco, United States

**Abstract** The mTOR pathway is an essential regulator of cell growth and metabolism. Midbrain dopamine neurons are particularly sensitive to mTOR signaling status as activation or inhibition of mTOR alters their morphology and physiology. mTOR exists in two distinct multiprotein complexes termed mTORC1 and mTORC2. How each of these complexes affect dopamine neuron properties, and whether they have similar or distinct functions is unknown. Here, we investigated this in mice with dopamine neuron-specific deletion of *Rptor* or *Rictor*, which encode obligatory components of mTORC1 or mTORC2, respectively. We find that inhibition of mTORC1 strongly and broadly impacts dopamine neuron structure and function causing somatodendritic and axonal hypotrophy, increased intrinsic excitability, decreased dopamine production, and impaired dopamine release. In contrast, inhibition of mTORC2 has more subtle effects, with selective alterations to the output of ventral tegmental area dopamine neurons. Disruption of both mTOR complexes leads to pronounced deficits in dopamine release demonstrating the importance of balanced mTORC1 and mTORC2 signaling for dopaminergic function.

**\*For correspondence:**
bateup@berkeley.edu

[†]These authors contributed equally to this work

**Competing interest:** The authors declare that no competing interests exist.

## Editor's evaluation

This manuscript by Kosillo and colleagues presents a series of carefully carried out experiments evaluating the impact of perturbing the mTORC1 and mTORC2 protein complexes selectively in mouse dopamine neurons. By utilizing dopamine neuron-specific Raptor and Rictor cKO mice, this paper elucidated which of these mTOR complexes are responsible for the regulation of dopamine neuronal functions, revealing the importance of mTORC1/2 signaling for the structure and function of dopamine neurons. This paper provided comprehensive data including structural, physiological, and biochemical alterations by genetic deletion of Raptor/Rictor in dopamine neurons.

## Introduction

The mechanistic target of rapamycin (mTOR) is an evolutionarily conserved kinase that serves as a central coordinator of cellular metabolism and regulator of anabolic and catabolic processes (*Saxton and Sabatini, 2017*). Balanced mTOR signaling is required for proper cell growth and function, while dysregulation of mTOR signaling is associated with various diseases (*Crino, 2016*; *Saxton and Sabatini, 2017*). Within the nervous system, mTOR fulfills distinct functions at different developmental stages. During embryonic development, mTOR regulates progenitor cell proliferation, differentiation,

and neuronal migration (*Blair and Bateup, 2020*; *Magri and Galli, 2013*; *Switon et al., 2017*). In mature neurons, mTOR controls neuronal morphology, physiology, and synaptic properties (*Costa-Mattioli and Monteggia, 2013*; *Hoeffer and Klann, 2010*; *Switon et al., 2017*). Consequently, dysregulation of mTOR signaling has a profound impact on nervous system function and several neurodevelopmental, psychiatric and neurodegenerative disorders are directly caused by or associated with altered mTOR activity (*Costa-Mattioli and Monteggia, 2013*; *Karalis et al., 2021b*; *Lipton and Sahin, 2014*).

While mTOR is present and active in all cells, different types of neurons show differential sensitivity to alterations in mTOR signaling (*Benthall et al., 2021*; *Benthall et al., 2018*; *Yang et al., 2012*). Midbrain dopamine (DA) neurons residing in the substantia nigra pars compacta (SNc) and ventral tegmental area (VTA) are particularly sensitive to mTOR signaling status, which may be a result of their unique physiology and high metabolic demands (*Matsuda et al., 2009*; *Pacelli et al., 2015*). For example, adult deletion of *Mtor* from VTA DA neurons leads to decreased DA release in the nucleus accumbens (NAc), altered DA transporter (DAT) expression, and attenuated synaptic and behavioral responses to cocaine (*Liu et al., 2018b*). Treatment with the mTOR inhibitor rapamycin, applied directly to striatal slices, decreases DA axon terminal size and depresses evoked DA release (*Hernandez et al., 2012*). By contrast, chronic activation of mTOR signaling via deletion of the genes encoding the upstream negative regulators Pten or Tsc1, leads to DA neuron hypertrophy and increased DA synthesis, with differential effects on DA release (*Diaz-Ruiz et al., 2009*; *Kosillo et al., 2019*).

In Parkinson's disease (PD) models, both partial mTOR inhibition and mTOR activation can be neuroprotective to degenerating DA neurons via different mechanisms. These include suppression of pro-apoptotic protein synthesis, activation of autophagy, enhancement of neuronal survival, and increased axon growth (*Zhu et al., 2019*). Therefore, up or down-regulation of mTOR signaling can impact multiple aspects of DA neuron biology. Since perturbations in mTOR signaling are implicated in multiple neurodevelopmental, neurodegenerative and psychiatric disorders (*Dayas et al., 2012*; *Kosillo and Bateup, 2021*; *Lan et al., 2017*) that affect dopaminergic function, it is important to understand how mTOR controls DA neuron cytoarchitecture and physiology.

mTOR participates in two multi-protein complexes termed mTORC1 and mTORC2, which have both shared and unique components (*Liu and Sabatini, 2020*; *Switon et al., 2017*). These complexes have distinct upstream activators and downstream targets, although some signaling crosstalk has been observed (*Liu and Sabatini, 2020*; *Xie and Proud, 2014*). Manipulations used to study mTOR signaling are often not specific to one complex. Rapamycin treatment can interfere with both mTORC1 and mTORC2 signaling (*Karalis and Bateup, 2021a*; *Sarbassov et al., 2006*), deletion of *Pten* activates both mTORC1 and mTORC2 (*Chen et al., 2019*), and loss of Tsc1 activates mTORC1 but suppresses mTORC2 signaling (*Carson et al., 2012*; *Karalis et al., 2021b*; *Meikle et al., 2008*). Consequently, from the current literature it is difficult to disentangle which mTOR complex is responsible for the observed effects. For mTOR-related diseases, it will be important to define which mTOR complex is most relevant for pathophysiology and which should be targeted in a potential therapeutic approach.

Several studies have investigated how manipulation of mTORC1 or mTORC2 signaling affects DA neuron properties. A previous study showed that activation of mTORC1 due to loss of Tsc1 caused DA neuron hypertrophy and increased dopamine synthesis (*Kosillo et al., 2019*). However, DA-Tsc1 KO mice had profound impairments in striatal DA release, which were likely driven by abnormalities in axon terminal architecture. Complete removal of the obligatory mTORC1 component Raptor and suppression of mTORC1 activity in Tsc1 KO DA neurons led to opposite changes in cell body size. However, DA release was equally compromised. The mechanism by which suppression of mTORC1 caused DA release deficits is unknown.

While mTORC1 is a known regulator of cell size (*Fingar et al., 2002*), it was shown that mTORC2 and not mTORC1 is responsible for somatic hypotrophy in VTA DA neurons in response to chronic morphine (*Mazei-Robison et al., 2011*). In this study, the morphine effects on VTA DA neuron soma size and intrinsic excitability could be phenocopied by deletion of *Rictor*, a necessary component of mTORC2 (*Sarbassov et al., 2004*), and prevented by *Rictor* overexpression (*Mazei-Robison et al., 2011*). Other studies showed alterations in DA tissue content and DAT function in mice with *Rictor* deleted from the majority of neurons or from catecholaminergic neurons, specifically (*Dadalko et al.,*

*2015a*; *Dadalko et al., 2015b*; *Siuta et al., 2010*). Therefore, open questions remain regarding the specific DA neuron properties that are regulated by mTORC1 and mTORC2 and whether these complexes have similar or distinct functions.

Here, we addressed this by directly comparing how DA neuron-specific deletion of *Rptor* or *Rictor* affects key cellular properties of SNc and VTA DA neurons. We find that disruption of mTORC1 strongly impacts DA neuron structure and function leading to global cellular hypotrophy, increased intrinsic excitability, reduced DA synthesis, and impaired DA release. By contrast, suppression of mTORC2 results in more mild morphological changes and selectively increases the excitability but reduces the output of VTA DA neurons. Global inhibition of mTOR signaling by concomitant deletion of *Rptor* and *Rictor* leads to major impairments in dopamine release, demonstrating that mTOR signaling is critical for dopaminergic function.

## Results

### Somatodendritic architecture of DA neurons is altered by mTORC1 or mTORC2 inhibition

The activity of mTORC1 or mTORC2 can be suppressed by deletion of the genes encoding their respective obligatory components Raptor or Rictor (*Figure 1a*). To achieve DA neuron-specific inhibition of each mTOR complex, we crossed *Slc6a3*[IREScre] mice, which express Cre from the *Slc6a3* (DAT) locus (*Bäckman et al., 2006*), to *Rptor*[fl/fl] (*Peterson et al., 2011*; *Sengupta et al., 2010*) or *Rictor*[fl/fl] (*Magee et al., 2012*; *Tang et al., 2012*) conditional knock-out (KO) mice (*Figure 1—figure supplement 1a,b*). We first assessed whether *Rptor* or *Rictor* deletion affected the number of DA neurons in the midbrain. In adult mice, stereological analysis revealed no difference in the number of midbrain DA neurons labeled with a tdTomato Cre reporter (Ai9) (*Madisen et al., 2010*) between *Rictor*[fl/fl];*Slc6a3*[IREScre/+];*ROSA26*[Ai9/+] ("DA-Rictor KO") mice and their respective wild-type (WT) littermate controls (*Figure 1—figure supplement 1c-i*). By contrast, deletion of *Rptor* (*Rptor*[fl/fl];*Slc6a3*[IREScre/+];*ROSA26*[Ai9/+]; "DA-Raptor KO") led to a significant reduction in the number of tdTomato+ DA neurons in both the SNc and VTA, while overall midbrain architecture was preserved (*Figure 1—figure supplement 1j-p*). Given that Cre expression in this model turns on in late embryogenesis after DA neurons are born, this suggests that chronic inhibition of mTORC1 may affect the subsequent development or survival of DA neurons.

SNc and VTA DA neurons are distinct in terms of their synaptic inputs and projection targets (*Beier et al., 2015*; *Lammel et al., 2011*; *Watabe-Uchida et al., 2012*), morphological and electrophysiological properties (*Giguère et al., 2019*; *Lammel et al., 2008*), and vulnerability to disease (*Brichta and Greengard, 2014*; *Gantz et al., 2018*). We therefore examined the impact of mTOR signaling perturbations on SNc and VTA DA neuron populations separately. We observed the expected inhibition of mTORC1 signaling in DA-Raptor KO mice, shown by strongly reduced phosphorylation of S6 (p-S6) and decreased soma size of both SNc and VTA DA neurons (*Figure 1b–m*). Since reductions in Raptor can alter dendritic morphology in other neuron types (*Angliker et al., 2015*; *Urbanska et al., 2012*), we filled individual midbrain DA neurons with neurobiotin (*Figure 1n*) and performed 3D reconstructions to assess their dendritic morphology. Consistent with other cell types, DA-Raptor KO neurons exhibited significantly reduced dendritic complexity and total dendrite length, with the most pronounced changes in SNc neurons (*Figure 1o–t*). However, the number of primary dendrites was not altered (*Figure 1u and v*). Thus, constitutive mTORC1 suppression leads to significant hypotrophy of midbrain DA neurons characterized by reduced soma size and dendritic arborization.

We have previously shown that Cre-dependent deletion of *Rictor* reduces p-473 Akt levels in cultured hippocampal neurons, indicative of suppressed mTORC2 activity (*Karalis et al., 2021b*). Due to technical limitations of the antibody, we were not able to assess p-473 at the level of individual neurons in brain sections. To assess mTORC1 signaling status in DA-Rictor KO mice, we measured p-S6 levels and found a small but significant reduction in p-S6 in both SNc and VTA DA neurons compared to WT controls (*Figure 2a–i and k*). Although p70S6K and S6 are not direct phosphorylation targets of mTORC2 (see *Figure 1a*), this observation is consistent with previous reports that mTORC2 suppression can lead to reduced activity of mTORC1 (*McCabe et al., 2020*; *Urbanska et al., 2012*), suggesting some crosstalk between mTORC1 and mTORC2. Consistent with their reduced S6 phosphorylation, DA-Rictor KO neurons in the SNc and VTA exhibited a small but significant reduction

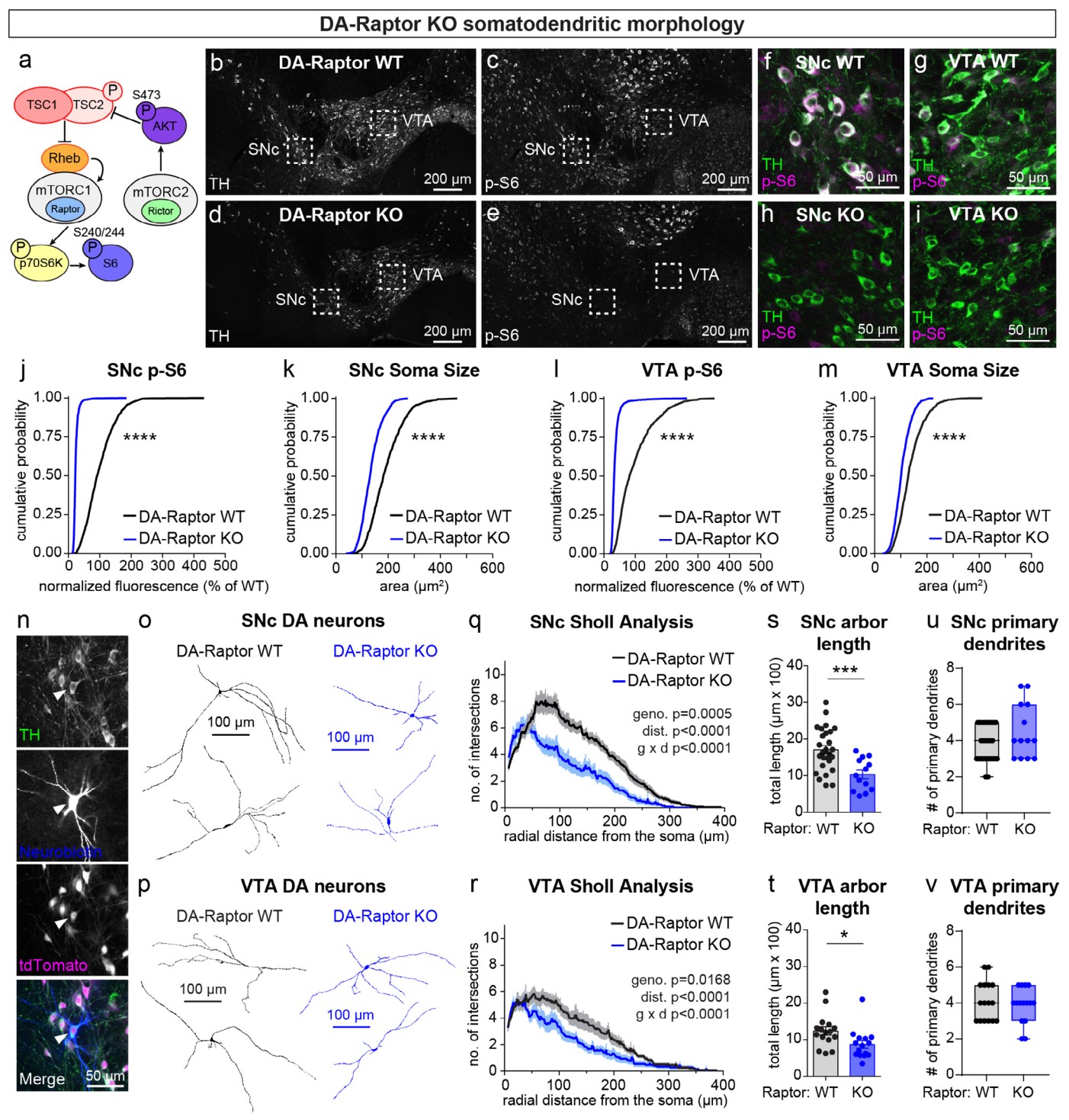

**Figure 1.** mTORC1 suppression causes somatodendritic hypotrophy of DA neurons. (**a**) Simplified mTOR signaling schematic showing mTORC1 and mTORC2 with their respective obligatory components Raptor and Rictor. Ribosomal protein S6 is phosphorylated on Ser240/244 by p70S6K, a direct phosphorylation target of mTORC1. AKT is phosphorylated on Ser473 by mTORC2. AKT is an upstream regulator of the Tsc1/2 complex, which negatively regulates mTORC1 activity via the small GTPase Rheb. (**b–e**) Representative confocal images of midbrain sections from DA-Raptor WT (**b,c**) and DA-Raptor KO (**d,e**) mice labeled with antibodies against tyrosine hydroxylase (TH) and phosphorylated S6 (p-S6, Ser240/244); scale bars = 200 μm. (**f–i**) Higher magnification merged images of the SNc (**f,h**) and VTA (**g,i**) boxed regions from panels b-e; scale bars = 50 μm. (**j,k**) Cumulative distributions of SNc DA neuron p-S6 levels (**j**) and soma area (**k**). DA-Raptor WT in black: n=1024 neurons from three mice, DA-Raptor KO in blue: n=1045 neurons from three mice; ****p<0.0001, Kolmogorov–Smirnov tests. (**l,m**) Cumulative distributions of VTA DA neuron p-S6 levels (**l**) and soma

*Figure 1 continued on next page*

*Figure 1 continued*

area (**m**). DA-Raptor WT in black: n=1389 neurons from three mice, DA-Raptor KO in blue: n=1526 neurons from three mice; ****p<0.0001, Kolmogorov–Smirnov tests. (**n**) Representative confocal images of a midbrain section containing a triple-labelled DA neuron (arrowhead; TH, neurobiotin and tdTomato Cre reporter-positive) used for dendritic arbor reconstruction and analysis. (**o,p**) Reconstructions of the dendrites and cell body of SNc (**o**) and VTA (**p**) DA neurons of the indicated genotypes. (**q**) Sholl analysis of SNc DA neuron dendritic arbors. Dark colored lines are the mean, lighter color shading is the SEM. DA-Raptor WT in black: n=27 neurons from 9 mice, DA-Raptor KO in blue: n=13 neurons from 6 mice. Two-way ANOVA p values are shown. (**r**) Sholl analysis of VTA DA neuron dendritic arbors. Dark colored lines are the mean, lighter color shading is the SEM. DA-Raptor WT in black: n=16 neurons from 6 mice, DA-Raptor KO in blue: n=15 neurons from 6 mice. Two-way ANOVA p values are shown. (**s**) Mean ± SEM total dendritic arbor length per SNc neuron. DA-Raptor WT: n=27 neurons from 9 mice, DA-Raptor KO: n=13 neurons from 6 mice; ***p=0.0003, Welch's two-tailed t-test. (**t**) Mean ± SEM total dendritic arbor length per VTA neuron. DA-Raptor WT: n=16 neurons from 6 mice, DA-Raptor KO: n=15 neurons from 6 mice; *p=0.0235, Welch's two-tailed t-test. (**u**) Box plots of the number of primary dendrites per SNc neuron for each genotype (boxes represent the inter-quartile range (25–75%), lines denote the median, whiskers represent min to max values). DA-Raptor WT: n=27 neurons from 9 mice, DA-Raptor KO: n=13 neurons from 6 mice; p=0.2078, Mann-Whitney test. (**v**) Box plots of the number of primary dendrites per VTA neuron for each genotype (boxes represent the inter-quartile range (25–75%), lines denote the median, whiskers represent min to max values). DA-Raptor WT: n=16 neurons from 6 mice, DA-Raptor KO: n=15 neurons from 6 mice; p=0.6420, Mann-Whitney test. For panels s-v, dots represent values from individual neurons. See also *Figure 1—figure supplement 1*.

The online version of this article includes the following figure supplement(s) for figure 1:

**Figure supplement 1.** Deletion of *Rptor*, but not *Rictor*, reduces DA neuron number (related to *Figures 1 and 2*).

in soma size (*Figure 2j and l*). Unlike mTORC1 suppression, inhibition of mTORC2 increased the complexity of proximal dendrites within ~50–100 µm of the soma, as measured by Sholl analysis (*Figure 2m–p*). However, there were no significant changes in the number of primary dendrites or total dendritic length (*Figure 2q–t*). Together, these data show that mTORC2 suppression has both similar (decreased soma size) and distinct (increased proximal dendrite branching) effects on somato-dendritic morphology as compared to mTORC1 inhibition.

## Striatal dopamine axon density is reduced by mTORC1 or mTORC2 inhibition

We previously showed using electron microscopy (EM) that constitutive mTORC1 activation due to loss of Tsc1 causes a significant enlargement of DA axon terminals, which is most pronounced in the SNc-innervated dorsolateral striatum (DLS) (*Kosillo et al., 2019*). In DA-Tsc1 KO mice (*Tsc1^{fl/fl};Slc6a3^{IRE-Scre/+}*), DA release was strongly reduced despite increased DA synthesis, suggesting that alterations in axon terminal structure may be an important contributing factor to DA releasability. To determine how suppression of mTORC1 or mTORC2 signaling affects the structural properties of DA axons, we developed a workflow to quantify DA axon density and size. These parameters are challenging to measure accurately with conventional microscopy due to the high density and small diameter of dopamine axons within the striatum. To overcome this, we employed protein-retention expansion microscopy (ProExM), whereby physical enlargement of the tissue effectively increases the resolution of light microscopy (*Tillberg et al., 2016*). We combined ProExM with light sheet fluorescence imaging and examined global DA axon architecture in DA-Raptor and DA-Rictor KO mice bred to the Ai9 tdTomato Cre reporter line (*Figure 3a*, *Figure 3—figure supplement 1a*, *Video 1*). To control for variations in tdTomato expression along DA axon segments, across striatal sub-regions, and between genotypes, automated segmentation of DA axons from light sheet volumes was performed using a 3D convolutional neural network, TrailMap (*Friedmann et al., 2020*). We first validated our DA axon segmentation pipeline in DA-Tsc1 KO mice. Consistent with EM analysis, quantification of TrailMap-segmented images revealed an increase in the total axon volume of striatal DA-Tsc1 KO projections, which was due to an increase in both axonal density and the radius of individual axon segments (*Figure 3—figure supplement 1b-f*, *Videos 2 and 3*).

We applied the validated ProExM-TrailMap pipeline to striatal slices from DA-Raptor-KO;Ai9 mice and found that, consistent with their somatodendritic hypotrophy, DA-Raptor KO axons in the DLS and NAc core exhibited a significant decrease in axon volume driven by a reduction in both axon density and radius (*Figure 3b–e*, *Videos 4 and 5*). These phenotypes are opposite to what we observed in DA-Tsc1 KO slices, demonstrating that mTORC1 signaling bi-directionally controls striatal DA axon density and size. We further noted that the striatal tdTomato fluorescence intensity in DA-Raptor KO mice showed regional heterogeneity, appearing as a patchwork pattern, which was particularly

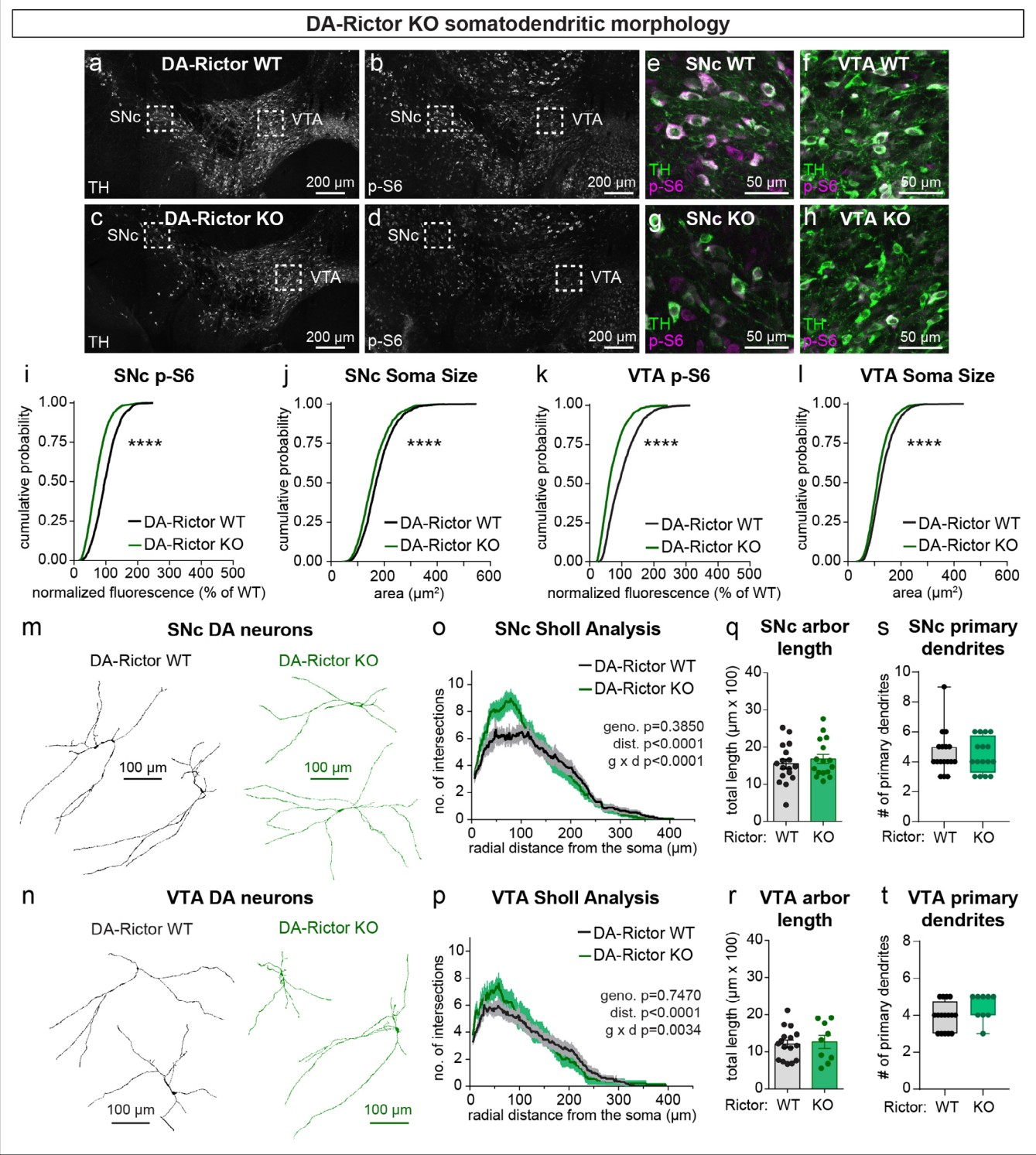

**Figure 2.** mTORC2 inhibition reduces DA neuron soma size and increases proximal dendrite branching. (**a–d**) Representative confocal images of midbrain sections from DA-Rictor WT (**a,b**) and DA-Rictor KO (**c,d**) mice labeled with antibodies against tyrosine hydroxylase (TH) and phosphorylated S6 (p-S6, Ser240/244); scale bars = 200 µm. (**e–h**) Higher magnification merged images of the SNc (**e,g**) and VTA (**h,i**) boxed regions from panels b-e; scale bars = 50 µm. (**i,j**) Cumulative distributions of SNc DA neuron p-S6 levels (**i**) and soma area (**j**). DA-Rictor WT in black: n=1280 neurons from three mice, DA-Rictor KO in green: n=1550 neurons from four mice; ****p<0.0001, Kolmogorov–Smirnov tests. (**k,l**) Cumulative distributions of VTA DA neuron p-S6 levels (**k**) and soma area (**l**). DA-Rictor WT in black: n=1968 neurons from three mice, DA-Rictor KO in green: n=2370 neurons from four mice; ****p<0.0001, Kolmogorov–Smirnov tests. (**m,n**) Reconstructions of the dendrites and cell body of SNc (**m**) and VTA (**n**) DA neurons of the indicated genotypes. (**o**) Sholl analysis of SNc DA neuron dendritic arbors. Dark colored lines are the mean, lighter color shading is the SEM. DA-Rictor WT in

*Figure 2 continued on next page*

## eLife Research article

Neuroscience

*Figure 2 continued*

black: n=17 neurons from 6 mice, DA-Rictor KO in green: n=16 neurons from 6 mice. Two-way ANOVA p values are shown. (**p**) Sholl analysis of VTA DA neuron dendritic arbors. Dark colored lines are the mean, lighter color shading is the SEM. DA-Rictor WT in black: n=16 neurons from 7 mice, DA-Rictor KO in green: n=9 neurons from 5 mice. Two-way ANOVA p values are shown. (**q**) Mean ± SEM total dendritic arbor length per cell. DA-Rictor WT: n=17 neurons from 6 mice, DA-Rictor KO: n=16 neurons from 6 mice; p=0.4633, Welch's two-tailed t-test. Dots represent values from individual neurons. (**r**) Mean ± SEM total dendritic arbor length per cell. DA-Rictor WT: n=16 neurons from 7 mice, DA-Rictor KO: n=9 neurons from 5 mice; p=0.7907, Welch's two-tailed t-test. Dots represent values from individual neurons. (**s**) Box plots of the number of primary dendrites per SNc neuron for each genotype (boxes represent the inter-quartile range (25–75%), lines denote the median, whiskers represent min to max values). DA-Rictor WT: n=17 neurons from 6 mice, DA-Rictor KO: n=16 neurons from 6 mice; p=0.9938, Mann-Whitney test. (**t**) Box plots of the number of primary dendrites per VTA neuron for each genotype (boxes represent the inter-quartile range (25–75%), lines denote the median, whiskers represent min to max values). DA-Rictor WT: n=16 neurons from 7 mice, DA-Rictor KO: n=9 neurons from 5 mice; p=0.1634, Mann-Whitney test. For panels q-t, dots represent values from individual neurons.

apparent within the DLS (*Figure 3f*). The striatal 'patches' with brighter tdTomato signal in DA-Raptor KO mice aligned with striatal patches (also called striosomes) defined by high mu opioid receptor (MOR) immunostaining using conventional confocal microscopy (*Figure 3—figure supplement 1g*). The total area of patches, as defined by MOR staining, was similar between DA-Raptor WT and KO mice, suggesting that the global patch-matrix striatal compartmentalization was intact (*Figure 3—figure supplement 1h*). However, the ratio of tdTomato fluorescence in MOR-delineated patch versus matrix regions was significantly higher in DA-Raptor KO slices compared to WT (*Figure 3—figure supplement 1i*), suggesting differential innervation of these regions in KO mice.

To further analyze the DA axon properties in the patch versus matrix compartments of DA-Raptor KO mice, we separately imaged these regions in ProExM-expanded DLS slices using lightsheet microscopy (*Figure 3g* and *Video 5*). The volume of DA axons within matrix regions was significantly reduced compared to patch regions in DA-Raptor KO slices. This was due to a selective reduction in axon density, as axon radius was similar across both compartments (*Figure 3h–j*). Given the relatively uniform tdTomato fluorescence across the striatum in WT mice (see *Figure 3—figure supplement 1g*), we could not reliably distinguish patch versus matrix in the ProExM samples. Together, these results indicate that mTORC1 inhibition reduces the size and density of DA axons throughout the striatum, with a more pronounced deficit in axonal innervation of the matrix compartment.

We performed ProExM-TrailMap on striatal slices from DA-Rictor-KO mice and found that DA-Rictor KO projections in the DLS and NAc core exhibited a small decrease in axon volume that was driven by a change in axon density but not radius (*Figure 3k–n*, *Videos 6 and 7*). Similar to WT mice, DA-Rictor KO axons showed relatively uniform innervation of the striatum, without clearly discernable regional differences in the tdTomato signal (*Figure 3o*). Taken together, the analysis of DA axons using ProExM-TrailMap demonstrates that constitutive mTORC1 suppression leads to DA axonal hypotrophy characterized by a concurrent reduction in axon density and radius. In contrast, mTORC2 suppression results in a more moderate decrease in axonal volume due to a selective reduction in axon density (*Figure 3p–r*).

## Electrophysiological properties of DA neurons are differentially altered by mTORC1 or mTORC2 suppression

Given the significant changes in DA neuron morphology observed in DA-Raptor KO mice, and to a lesser extent DA-Rictor KO mice, we performed whole-cell recordings to determine how these structural changes affected intrinsic membrane properties and excitability. Consistent with the somatodendritic and axonal hypotrophy of DA-Raptor KO neurons, we observed a large decrease in membrane capacitance and increase in membrane resistance in SNc DA-Raptor KO cells compared to controls (*Figure 4a–b*), with no change in the resting membrane potential (*Table 1*). In VTA neurons, deletion of *Rptor* decreased capacitance (*Figure 4f*) but did not significantly alter membrane resistance (*Figure 4g*) compared to WT cells.

Upon hyperpolarization, SNc DA neurons show a prominent sag response driven by hyperpolarization-activated ($I_h$) current and a rebound response that depends on $I_h$, type-A potassium current ($I_A$) and T-type calcium channels (*Amendola et al., 2012*; *Evans et al., 2017*). This sag-rebound signature distinguishes SNc DA neurons from the neighboring VTA DA cells (*Lammel et al., 2008*). In response to a negative current step (–100 pA), we observed a significant reduction in both the sag and rebound

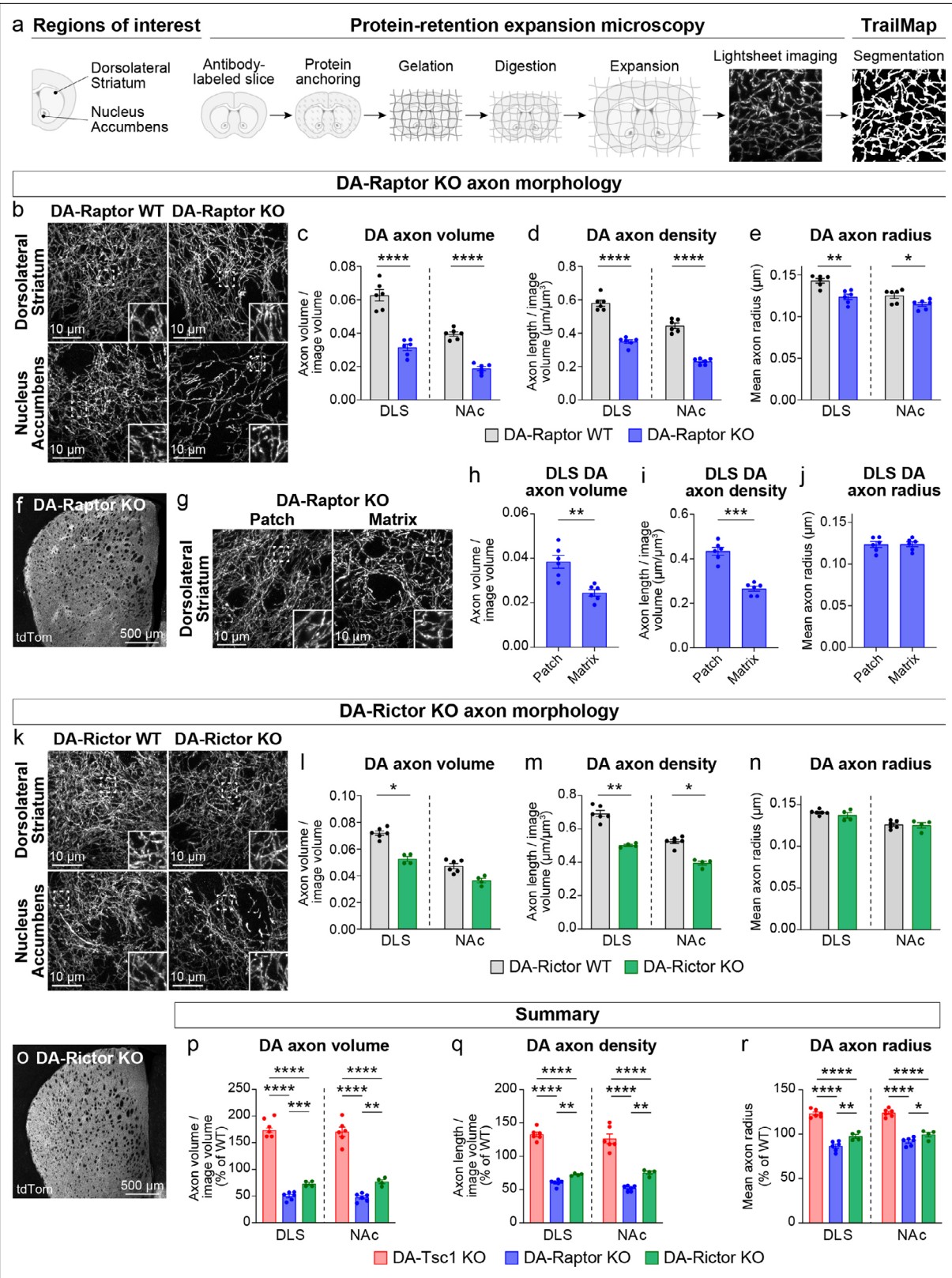

**Figure 3.** Dopamine axon density is reduced by inhibition of mTORC1 or mTORC2. (**a**) Outline of the expansion microscopy and analysis workflow. Striatal slices containing tdTomato-labeled DA axons were labeled with an anti-RFP antibody and processed for protein-retention expansion microscopy (ProExM). Regions of interest containing dorsolateral striatum (DLS) or nucleus accumbens core (NAc) were imaged with a light sheet fluorescence microscope and DA axons were segmented using TrailMap. (**b**) Representative light sheet images of ProExM-processed striatal DA axons from DA-

*Figure 3 continued on next page*

*Figure 3 continued*

Raptor WT (left panels) and DA-Raptor KO (right panels) mice. Scale bars are normalized by expansion factor. Insets show high magnification images of boxed regions from the same panels. (**c–e**) Mean ± SEM DA axon volume (**c**), density (**d**), and radius (**e**) in the DLS and NAc regions from DA-Raptor WT and DA-Raptor KO mice. n=6 slices from 3 mice per genotype (values are the average of 4 images per slice). Axon volume (**c**), ****p<0.0001 WT vs KO in DLS, ****p<0.0001 WT vs KO in NAc. Axon density (**d**), ****p<0.0001 WT vs KO in DLS, ****p<0.0001 WT vs KO in NAc. Axon radius (**e**), **p=0.0011 WT vs KO in DLS, *p=0.0279 WT vs KO in NAc. Welch's unpaired t-tests. (**f**) Representative confocal image of an unexpanded DA-Raptor KO striatal section, visualized by Cre-dependent tdTomato (tdTom). (**g**) Representative light sheet images of ProExM-processed DLS patch and matrix regions from DA-Raptor KO mice showing tdTomato-labeled DA axons, scale bars = 10 µm (normalized by expansion factor). Insets show higher magnification images of boxed regions from the same panels. (**h–j**) Mean ± SEM DA axon volume (**h**), density (**i**), and radius (**j**) in the DLS patch and matrix compartments from DA-Raptor KO mice. Axon volume (**h**), **p=0.0019. Axon density (**i**), ***p=0.0003. Axon radius (**j**), p=0.7536. Paired t-tests. n=6 slices from 3 mice (values are the average of four images per slice). (**k**) Representative light sheet images of ProExM-processed striatal DA axons from DA-Rictor WT (left panels) and DA-Rictor KO (right panels) mice. Scale bars are normalized by expansion factor. Insets show high magnification images of boxed regions from the same panels. (**l-n**) Mean ± SEM DA axon volume (**l**), density (**m**), and radius (**n**) in the DLS and NAc regions from DA-Rictor WT and DA-Rictor KO mice. For DA-Rictor WT, n=6 slices from 3 mice. For DA-Rictor KO, n=4 slices from 2 mice (values are the average of 4 images per slice). Axon volume (**l**), *p=0.0351 WT vs KO in DLS, p=0.1164 WT vs KO in NAc. Axon density (**m**), **p=0.0068 WT vs KO in DLS, *p=0.0184 WT vs KO in NAc. Axon radius (**n**), p=0.8059 WT vs KO in DLS, p=0.6195 WT vs KO in NAc. Welch's unpaired t-tests. (**o**) Representative confocal image of an unexpanded DA-Rictor KO striatal section, visualized by Cre-dependent tdTomato. (**p–r**) Summary data across all genotypes examined by ProExM-TrailMap. Mean ± SEM DA axon volume (**p**), density (**q**), and radius (**r**) in the DLS and NAc regions from DA-Tsc1 KO, DA-Raptor KO, and DA-Rictor KO mice expressed as a percentage of WT values for each genotype. Axon volume (**p**), DLS: p<0.0001, one-way ANOVA; Holm-Sidak's multiple comparison tests, ****p<0.0001 DA-Tsc1 KO vs DA-Raptor KO, ****p<0.0001 DA-Tsc1 KO vs DA-Rictor KO, ***p=0.0010 DA-Raptor KO vs DA-Rictor KO. NAc: p<0.0001, one-way ANOVA; Holm-Sidak's multiple comparison tests, ****p<0.0001 DA-Tsc1 KO vs DA-Raptor KO, ****p<0.0001 DA-Tsc1 KO vs DA-Rictor KO, **p=0.0076 DA-Raptor KO vs DA-Rictor KO. Axon density (**q**), DLS: p<0.0001, one-way ANOVA; Holm-Sidak's multiple comparison tests, ****p<0.0001 DA-Tsc1 KO vs DA-Raptor KO, ****p<0.0001 DA-Tsc1 KO vs DA-Rictor KO, **p<0.0087 DA-Raptor KO vs DA-Rictor KO. NAc: p<0.0001, one-way ANOVA; Holm-Sidak's multiple comparison tests, ****p<0.0001 DA-Tsc1 KO vs DA-Raptor KO, ****p<0.0001 DA-Tsc1 KO vs DA-Rictor KO, **p=0.0070 DA-Raptor KO vs DA-Rictor KO. Axon radius (**r**), DLS: p<0.0001, one-way ANOVA; Holm-Sidak's multiple comparison tests, ****p<0.0001 DA-Tsc1 KO vs DA-Raptor KO, ****p<0.0001 DA-Tsc1 KO vs DA-Rictor KO, **p=0.0038 DA-Raptor KO vs DA-Rictor KO. NAc: p<0.0001, one-way ANOVA; Holm-Sidak's multiple comparison tests, ****p<0.0001 DA-Tsc1 KO vs DA-Raptor KO, ****p<0.0001 DA-Tsc1 KO vs DA-Rictor KO, *p=0.0210 DA-Raptor KO vs DA-Rictor KO. See also *Figure 3—figure supplement 1* and *Videos 1–7*.

The online version of this article includes the following figure supplement(s) for figure 3:

**Figure supplement 1.** Development of a ProExM-TrailMap pipeline for the analysis of DA axon morphology (related to *Figure 3*).

responses of DA-Raptor KO SNc neurons compared to WT controls (*Figure 4c–e*). We plotted this as a percentage of the maximum hyperpolarization from baseline to account for the increased membrane resistance of DA-Raptor KO neurons, which influenced their degree of hyperpolarization (*Figure 4d–e* and *Table 1*). In contrast to SNc neurons, mTORC1 suppression in VTA neurons had no impact on responses to hyperpolarizing current (*Figure 4h–j*).

We next examined responses to depolarizing current steps in DA-Raptor KO neurons to

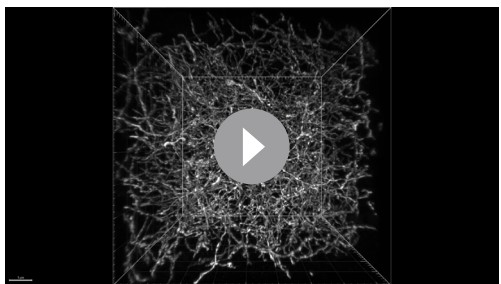

**Video 1.** 3D rendering of tdTomato-labeled DA axons in the dorsolateral striatum. 3D rendering of tdTomato-labeled Raptor KO DA axons (*Rptor*^fl/fl^;*Slc6a3*^IREScre/+^;*ROSA26*^Ai9/+^ mouse) in the dorsolateral striatum (DLS).Tissue sections were expanded ~4 x using ProExM. Video shows a 41.87 × 41.87 x 50.52 µm volume generated from z-stack lightsheet images (normalized by expansion factor).

https://elifesciences.org/articles/75398/figures#video1

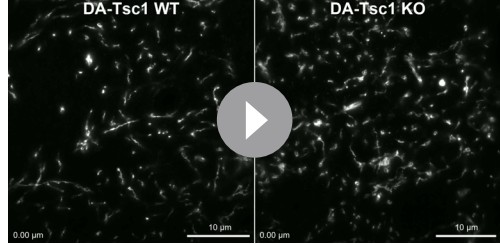

**Video 2.** Video of z-stack lightsheet images of tdTomato-labeled Tsc1 WT and Tsc1 KO DA axons in the nucleus accumbens core. Striatal sections containing tdTomato-labeled DA axons from DA-Tsc1 WT (*Tsc1*^+/+^;*Slc6a3*^IREScre/+^;*ROSA26*^Ai9/+^) and DA-Tsc1 KO (*Tsc1*^fl/fl^;*Slc6a3*^IREScre/+^;*ROSA26*^Ai9/+^) mice were expanded with ProExM and imaged on a lightsheet microscope. Video shows 39.32 × 39.32 x 25.13 µm (DA-Tsc1 WT) and 40.84 × 40.84 x 26.10 µm (DA-Tsc1 KO) z-stacks (normalized by expansion factor) from the nucleus accumbens core (NAc).

https://elifesciences.org/articles/75398/figures#video2

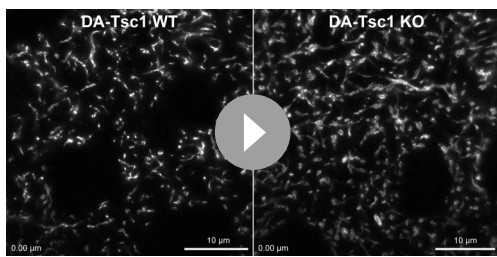

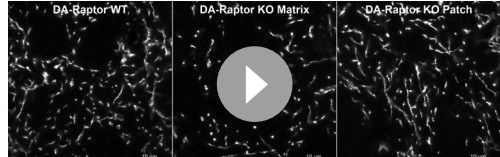

**Video 3.** Video of z-stack lightsheet images of tdTomato-labeled Tsc1 WT and Tsc1 KO DA axons in the dorsolateral striatum. Striatal sections containing tdTomato-labeled DA axons from DA-Tsc1 WT (*Tsc1*[+/+]; *Slc6a3*[IREScre/+];*ROSA26*[Ai9/+]) and DA-Tsc1 KO (*Tsc1*[fl/fl]; *Slc6a3*[IREScre/+];*ROSA26*[Ai9/+]) mice were expanded with ProExM and imaged on a lightsheet microscope. Video shows 38.64 × 38.64 x 32.07 µm (DA-Tsc1 WT) and 41.25 × 41.25 x 34.25 µm (DA-Tsc1 KO) z-stacks (normalized by expansion factor) from the DLS.
https://elifesciences.org/articles/75398/figures#video3

**Video 5.** Video of z-stack lightsheet images of tdTomato-labeled Raptor WT and Raptor KO DA axons in the dorsolateral striatum. Striatal sections containing tdTomato-labeled DA axons from DA-Raptor WT (*Rptor*[+/+];*Slc6a3*[IREScre/+];*ROSA26*[Ai9/+]) and DA-Raptor KO (*Rptor*[fl/fl];*Slc6a3*[IREScre/+];*ROSA26*[Ai9/+]) mice were expanded with ProExM and imaged on a lightsheet microscope. Video shows 40.97 × 40.97 x 35.60 µm (DA-Raptor WT) and 41.87 × 41.87 x 36.39 µm (DA-Raptor KO) z-stacks (normalized by expansion factor) from the DLS. For DA-Raptor KO sections, putative patch and matrix regions were imaged separately.
https://elifesciences.org/articles/75398/figures#video5

assess their intrinsic excitability (*Figure 4k–p*). We found a significant decrease in the minimum current required to evoke an action potential (rheobase) in DA-Raptor KO neurons compared to WT (*Figure 4l and o*). The input-output relationship of SNc and VTA DA-Rptor KO neurons to current steps of increasing amplitude was also altered such that they had increased firing frequency compared to WT cells at currents <150 pA, but exhibited substantial depolarization block and reduced firing rates at currents >300 pA (*Figure 4m and p*). Thus, constitutive mTORC1 inhibition increased the intrinsic excitability of both SNc and VTA neurons at lower current amplitudes, while firing was impaired in response to large depolarizing currents.

We measured the properties of DA-Rictor KO neurons (*Table 2*) and found that there was a small but significant decrease in capacitance for both SNc and VTA neurons (*Figure 5a and f*), consistent with their modest decrease in soma size (see *Figure 2*). VTA, but not SNc, DA-Rictor KO neurons also exhibited a significant increase in membrane resistance (*Figure 5b and g*), with no change in resting membrane potential (*Table 2*). In contrast to loss of *Rptor*, deletion of *Rictor* did not affect sag-rebound responses in either SNc (*Figure 5c–e*) or VTA neurons (*Figure 5h–j*). We assessed intrinsic excitability and found that DA-Rictor

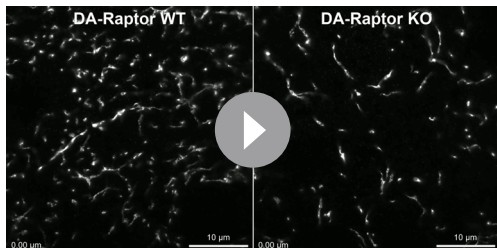

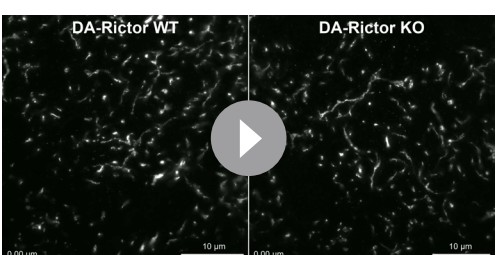

**Video 4.** Video of z-stack lightsheet images of tdTomato-labeled Raptor WT and Raptor KO DA axons in the nucleus accumbens core. Striatal sections containing tdTomato-labeled DA axons from DA-Raptor WT (*Rptor*[+/+];*Slc6a3*[IREScre/+];*ROSA26*[Ai9/+]) and DA-Raptor KO (*Rptor*[fl/fl];*Slc6a3*[IREScre/+];*ROSA26*[Ai9/+]) mice were expanded with ProExM and imaged on a lightsheet microscope. Video shows 41.65 × 41.65 x 35.45 µm (DA-Raptor WT) and 42.08 × 42.08 x 35.81 µm (DA-Raptor KO) z-stacks (normalized by expansion factor) from the NAc.
https://elifesciences.org/articles/75398/figures#video4

**Video 6.** Video of z-stack lightsheet images of tdTomato-labeled Rictor WT and Rictor KO DA axons in the nucleus accumbens core. Striatal sections containing tdTomato-labeled DA axons from DA-Rictor WT (*Rictor*[+/+];*Slc6a3*[IREScre/+];*ROSA26*[Ai9/+]) and DA-Rictor KO (*Rictor*[fl/fl];*Slc6a3*[IREScre/+];*ROSA26*[Ai9/+]) mice were expanded with ProExM and imaged on a lightsheet microscope. Video shows 39.19 × 39.19 x 36.52 µm (DA-Rictor WT) and 42.58 × 42.58 x 39.67 µm (DA-Rictor KO) z-stacks (normalized by expansion factor) from the NAc.
https://elifesciences.org/articles/75398/figures#video6

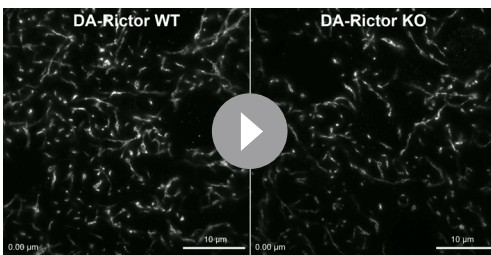

**Video 7.** Video of z-stack lightsheet images of tdTomato-labeled Rictor WT and Rictor KO DA axons in the dorsolateral striatum. Striatal sections containing tdTomato-labeled DA axons from DA-Rictor WT (*Rictor*[+/+];*Slc6a3*[IREScre/+];*ROSA26*[Ai9/+]) and DA-Rictor KO (*Rictor*[fl/fl];*Slc6a3*[IREScre/+];*ROSA26*[Ai9/+]) mice were expanded with ProExM and imaged on a lightsheet microscope. Video shows 39.62 × 39.62 x 30.17 μm (DA-Rictor WT) and 44.05 × 44.05 x 33.54 μm (DA-Rictor KO) z-stacks (normalized by expansion factor) from the DLS. https://elifesciences.org/articles/75398/figures#video7

KO SNc neurons had no change in rheobase and only a small shift in the input-output curve, which trended towards decreased excitability with higher current injections (*Figure 5k–m*). In contrast, DA-Rictor KO VTA neurons exhibited a significant decrease in rheobase and increased firing in response to current steps between 75 and 200 pA (*Figure 5n–p*). Together, these data show that mTORC1 suppression alters multiple aspects of DA neuron physiology with the most pronounced effects in SNc DA neurons, while mTORC2 inhibition primarily affects the excitability of VTA neurons.

## Evoked striatal DA release is differentially impacted by mTORC1 or mTORC2 inhibition

To determine how DA release is impacted by loss of Raptor or Rictor, we used fast scan cyclic voltammetry (FCV) to measure evoked DA release within seven striatal subregions. We examined multiple subregions since these areas are differentially innervated by DA neuron subpopulations and participate in distinct circuits and behavioral functions (*de Jong et al., 2019*; *Haber, 2014*; *Kramer et al., 2018*; *Lerner et al., 2015*; *Poulin et al., 2018*). The dorsal striatum (sites 1–6, *Figure 6a*) is preferentially innervated by SNc DA neurons, while the ventral striatum, or NAc (site 7, *Figure 6a*), is primarily targeted by VTA DA cells. Constitutive mTORC1 suppression in DA-Raptor KO mice resulted in a profound impairment in electrically evoked DA release across all regions sampled (*Figure 6a–c* and *Figure 6—figure supplement 1*). We observed a more than 60% reduction in peak evoked DA release with both single pulse (*Figure 6b and c*) and high frequency burst (*Figure 6—figure supplement 1*) stimulation across dorsal and ventral striatal sampling sites. This level of impairment in evoked DA release is generally consistent with that observed in the dorsal striatum of DA-Tsc1 KO mice, in which mTORC1 is hyperactive (*Kosillo et al., 2019*). Thus, DA release impairments can occur either with constitutive mTORC1 activation or suppression.

We compared the kinetics of DA re-uptake via DAT and found a significant reduction in re-uptake rate in the dorsal striatum (sites 1–6, *Figure 6d and e*) but not the NAc (site 7, *Figure 6f and g*) of DA-Raptor KO mice, suggesting a region-specific decrease in DAT expression and/or activity. Baseline DAT expression is higher in the dorsal compared to ventral striatum (*Condon et al., 2019*; *Rice and Cragg, 2008*; *Sulzer et al., 2016*), which presents an additional measure of control for adjusting DA signal duration and/or amplitude via changes in DAT. The decreased DA re-uptake rate in DA-Raptor KO mice may serve a compensatory function to boost dopaminergic signaling in response to compromised DA release. This is consistent with the decreased DA release and re-uptake observed in mice with conditional deletion of *Mtor* from VTA neurons (*Liu et al., 2018b*).

Constitutive mTORC2 suppression in DA-Rictor KO mice resulted in a different profile of striatal DA release. In DA-Rictor KO mice, we found a moderate reduction in peak evoked DA release specifically in the ventral half of the striatum and NAc (sites 4–7), while DA release in the dorsal-most regions (sites 1–3) was unchanged (*Figure 7a–c* and *Figure 7—figure supplement 1*). On average, we observed an approximately 25% reduction in peak evoked DA release both with single pulse (*Figure 7b and c*) and high frequency burst (*Figure 7—figure supplement 1*) stimulation at sampling sites 4–7, which receive proportionately denser innervation from the VTA compared to the SNc (*Chen et al., 2021*). The observation that release impairments in DA-Rictor KO mice were selectively present in ventral regions is consistent with the more pronounced electrophysiological changes we found in VTA neurons compared to SNc neurons with *Rictor* deletion (see *Figure 5*). We observed no change in DA re-uptake kinetics in either the dorsal striatum (*Figure 7d and e*) or NAc (*Figure 7f and g*) of DA-Rictor KO slices.

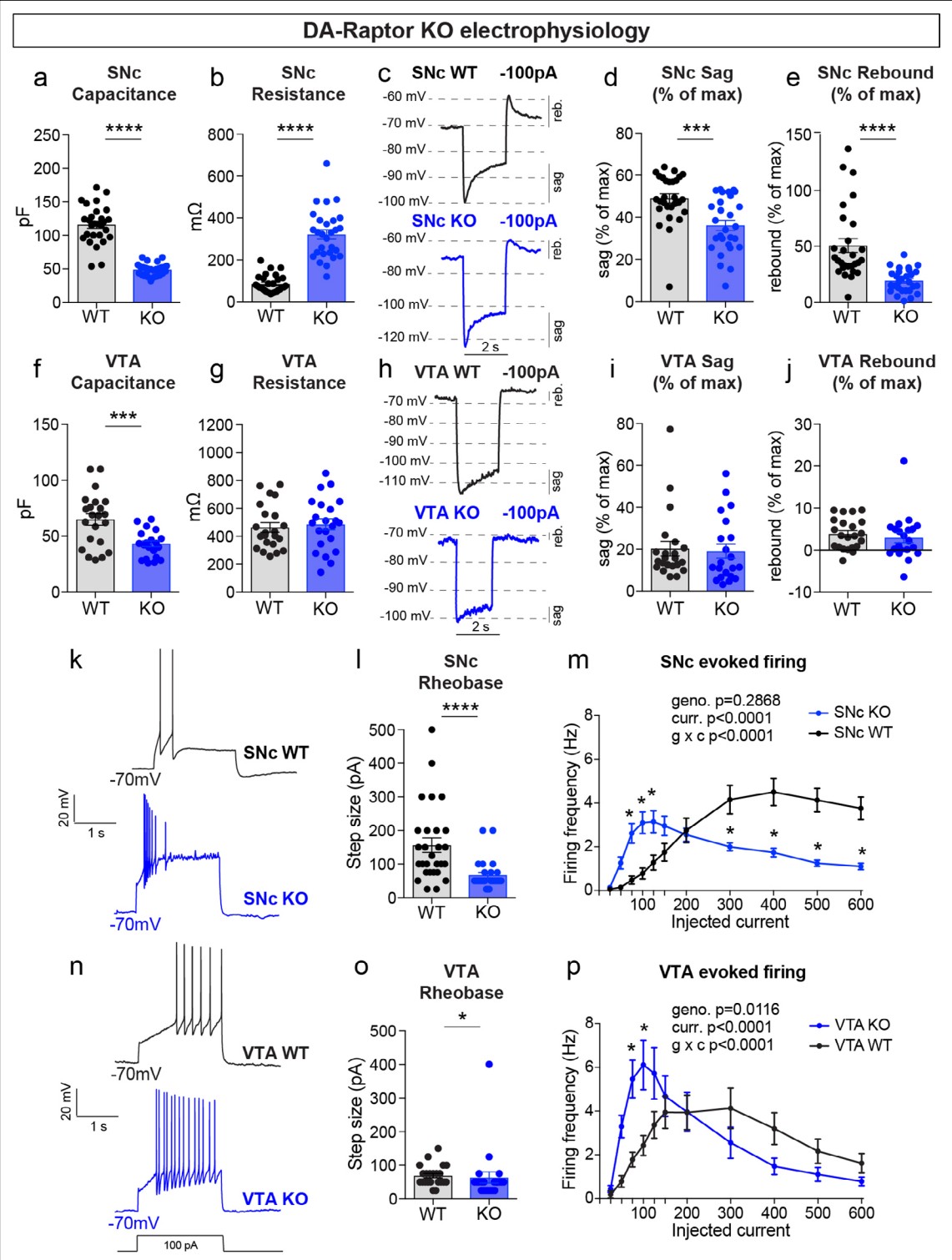

**Figure 4.** mTORC1 suppression increases the excitability of SNc and VTA DA neurons. (**a,b**) Mean ± SEM membrane capacitance (**a**) and membrane resistance (**b**) of SNc DA neurons. DA-Raptor WT in black: n=28 neurons from eight mice, DA-Raptor KO in blue: n=28 neurons from six mice. Capacitance (**a**), ****p<0.0001, unpaired two-tailed t-test. Resistance (**b**), ****p<0.0001, Mann–Whitney test. (**c**) Example current-clamp recordings from SNc DA neurons of the indicated genotypes in response to a −100 pA current step. reb.=rebound. (**d,e**) Mean ± SEM sag (**d**) and rebound (**e**) amplitude expressed as a percentage of the maximum hyperpolarization from baseline in SNc DA neurons. DA-Raptor WT in black: n=28 neurons from six mice, DA-Raptor KO in blue: n=28 neurons from six mice. Sag (**d**), ***p=0.0003. Rebound (**e**), ****p<0.0001. Mann–Whitney tests. (**f,g**) Mean ± SEM membrane capacitance (**f**) and membrane resistance (**g**) of VTA DA neurons. DA-Raptor WT in black: n=22 neurons from eight mice, DA-Raptor KO in blue: n=22

*Figure 4 continued on next page*

*Figure 4 continued*

neurons from six mice. Capacitance (**f**), ***p=0.0005. Resistance (**g**), p=0.6769. Unpaired two-tailed t-tests. (**h**) Example current-clamp recordings from VTA DA neurons of the indicated genotypes in response to a −100 pA current step. (**i,j**) Mean ± SEM sag (**i**) and rebound (**j**) amplitude expressed as a percentage of the maximum hyperpolarization from baseline in VTA DA neurons. DA-Raptor WT in black: n=22 neurons from six mice, DA-Raptor KO in blue: n=21 neurons from six mice. Sag (**i**), p=0.3660. Rebound (**j**), p=0.2513. Mann–Whitney tests. (**k**) Examples of action potential firing elicited with a+100 pA current step in SNc DA neurons of the indicated genotypes. (**l**) Mean ± SEM rheobase of SNc DA neurons calculated as the current at which action potentials were first elicited. DA-Raptor WT in black: n=28 neurons from eight mice, DA-Raptor KO in blue: n=27 neurons from six mice, ****p<0.0001, Mann–Whitney test. (**m**) Input-output curves showing the firing frequency of SNc DA neurons in response to positive current steps of increasing amplitude. Data are displayed as mean ± SEM. DA-Raptor WT in black: n=28 neurons from eight mice, DA-Raptor KO in blue: n=28 neurons from six mice. Two-way ANOVA p values are shown. Sidak's multiple comparisons tests: $p_{25pA}$ >0.9999, $p_{50pA}$=0.3649, *$p_{75pA}$=0.0009, *$p_{100pA}$=0.0002, *$p_{125pA}$=0.0069, $p_{150pA}$=0.2497, $p_{200pA}$ >0.9999, *$p_{300pA}$=0.0009, *$p_{400pA-600pA}$ <0.0001. (**n**) Examples of action potential firing elicited with a+100 pA current step in VTA DA neurons of the indicated genotypes. (**o**) Mean ± SEM rheobase of VTA DA neurons calculated as the current at which action potentials were first elicited. DA-Raptor WT in black: n=22 neurons from eight mice, DA-Raptor KO in blue: n=22 neurons from six mice, *p=0.0190, Mann–Whitney test. (**p**) Input-output curves showing the firing frequency of VTA DA neurons in response to current steps of increasing amplitude. Data are displayed as mean ± SEM. DA-Raptor WT in black: n=22 neurons from eight mice, DA-Raptor KO in blue: n=22 neurons from six mice. Two-way ANOVA p values are shown. Sidak's multiple comparisons tests: $p_{25pA}$ >0.9999, $p_{50pA}$=0.0902, *$p_{75pA}$=0.0014, *$p_{100pA}$=0.0014, $p_{125pA}$=0.1398, $p_{150pA}$=0.9983, $p_{200pA}$ >0.9999, $p_{300pA}$=0.6959, $p_{400pA}$=0.5691, $p_{500pA}$=0.9718, $p_{600pA}$=0.9963. For all bar graphs, dots represent values for individual neurons. See also *Table 1*.

Together these data demonstrate that constitutive mTORC1 suppression results in a ubiquitous impairment in evoked striatal DA release and a likely compensatory decrease in DA clearance rate in the dorsal striatum. mTORC2 inhibition causes a milder impairment in DA release, which specifically affects ventral regions that are targeted by VTA projections.

## Inhibition of mTORC1 but not mTORC2 impairs DA synthesis

To investigate whether alterations in DA synthesis or vesicle packaging could account for the release impairments in DA-Raptor or DA-Rictor KO mice, we harvested striatal tissue and measured protein levels of the vesicular monoamine transporter 2 (VMAT-2), which is responsible for vesicular loading of DA, and tyrosine hydroxylase (TH), which is the rate-limiting enzyme in DA synthesis. In striatal lysates from DA-Raptor KO mice, we found no change in VMAT-2 expression compared to WT controls (*Figure 8a and b*). However, DA-Raptor KO mice showed a significant downregulation of striatal TH protein expression (*Figure 8a and b*), consistent with their impaired DA release. TH levels were also reduced in DA cell bodies in both the SNc (*Figure 8d–f*) and VTA (*Figure 8g–i*). Together, these data suggest reduced DA synthesis capacity resulting from constitutive mTORC1 suppression. To examine DA synthesis and content directly, we harvested dorsal and ventral striatal tissue and measured DA levels using high performance liquid chromatography (HPLC). We found a large reduction in the total tissue DA content across both regions in DA-Raptor KO mice compared to controls, while the ratio of DA to its primary metabolite DOPAC was unchanged (*Figure 8j and k*). This profile is indicative of reduced DA production in DA-Raptor KO mice, rather than increased DA turnover.

Given the altered re-uptake kinetics observed in DA-Raptor KO mice in the FCV experiments, we examined striatal protein levels of DAT and phosphorylated DAT at Thr53 (p-DAT), which promotes transporter function (*Foster et al., 2012*). We found reduced total DAT levels in the striatum of DA-Raptor KO mice (*Figure 8a and b*), consistent with the decreased re-uptake rate observed by FCV. Despite decreased DAT levels, the remaining protein was highly phosphorylated in DA-Raptor KO mice (*Figure 8c*). Phosphorylation of DAT at T53 is typically thought to promote DAT activity. However, since we observed slower DA re-uptake rate in the DLS of DA-Raptor KO mice (see *Figure 6d and e*), our results indicate that the enhanced phosphorylation of DAT was not able to functionally compensate for the large reduction in total DAT expression in this condition. Together these data show that constitutive mTORC1 suppression leads to downregulation of TH levels in both the somatic and axonal compartments, resulting in reduced DA synthesis capacity and tissue content. DA-Raptor KO mice also have reduced axonal DAT expression, which may serve to prolong DA actions in response to limited neurotransmitter availability and release.

We next investigated whether changes in DA synthesis or packaging might be responsible for the altered DA release in DA-Rictor KO mice. Similar to *Rptor* deletion, loss of *Rictor* did not affect expression of VMAT-2 within the striatum (*Figure 9a and b*). However, in contrast to mTORC1 inhibition, mTORC2 suppression did not strongly affect striatal TH, DAT, or p-DAT levels (*Figure 9a–c*). In the midbrain, we found a small shift in the cumulative distribution of TH levels in individual SNc neurons

**Table 1.** Summary of the electrophysiology properties of DA-Raptor WT and DA-Raptor KO SNc and VTA DA neurons.

| Properties | DA-Raptor WT (SNc) | | | | DA-Raptor KO (SNc) | | | | WT vs KO |
|---|---|---|---|---|---|---|---|---|---|
| | Mean | SEM | n (cells) | n (mice) | Mean | SEM | n (cells) | n (mice) | p-value/ test |
| Series resistance (mOhms) | 3.269 | 0.308 | 28 | 6 | 3.494 | 0.213 | 28 | 6 | 0.0965 Mann-Whitney |
| Membrane resistance (mOhms) | 85.79 | 7.869 | 28 | 6 | 321.0 | 22.20 | 28 | 6 | **<0.0001** Mann-Whitney |
| Membrane capacitance (pF) | 115.8 | 5.365 | 28 | 6 | 48.92 | 1.661 | 28 | 6 | **<0.0001** unpaired t-test |
| Resting membrane potential (mV) | −56.13 | 0.904 | 28 | 6 | −53.29 | 1.372 | 28 | 6 | 0.0902 unpaired t-test |
| Rheobase (current when first action potentials occur, pA) | 156.3 | 21.54 | 28 | 6 | 66.67 | 8.333 | 27 | 6 | **<0.0001** Mann-Whitney |
| Action potential (AP) threshold (mV) | −33.45 | 0.976 | 28 | 6 | −32.56 | 1.313 | 27 | 6 | 0.5885 unpaired t-test |
| Action potential peak (maximum membrane potential, mV) | 24.85 | 1.279 | 28 | 6 | 15.55 | 1.478 | 27 | 6 | **<0.0001** unpaired t-test |
| Action potential height (change in membrane potential from the start of the AP to maximum depolarization, mV) | 66.44 | 1.556 | 28 | 6 | 57.74 | 1.468 | 27 | 6 | **0.0002** unpaired t-test |
| Afterhyperpolarization (minimum membrane potential after the AP, mV) | −61.70 | 1.645 | 28 | 6 | −56.17 | 1.365 | 27 | 6 | **0.0127** unpaired t-test |
| Afterhyperpolarization (change in membrane potential from the start of the AP to maximum hyperpolarization, mV) | 20.11 | 1.171 | 28 | 6 | 15.06 | 1.011 | 27 | 6 | **0.0020** unpaired t-test |
| Maximum hyperpolarization in response to −100 pA (from ~−70 mV in response to a 2 second −100 pA current step, mV) | −91.04 | 1.841 | 28 | 6 | −124.4 | 2.271 | 28 | 6 | **<0.0001** unpaired t-test |
| Sag component in response to −100 pA (maximum hyperpolarization minus the steady state membrane potential in the last 50ms of the current step, mV) | 11.49 | 1.060 | 28 | 6 | 19.38 | 1.283 | 28 | 6 | **<0.0001** unpaired t-test |
| Sag component expressed as a percentage (sag component as a percentage of the total step size, calculated as the difference between the max hyperpolarization and baseline potential, %) | 48.99 | 2.219 | 28 | 6 | 36.18 | 2.472 | 28 | 6 | **0.0003** Mann-Whitney |
| Rebound depolarization in response to −100 pA (baseline membrane potential minus the maximum depolarization within 500ms of the end of the current step, mV) | 11.32 | 1.350 | 28 | 6 | 10.15 | 1.030 | 28 | 6 | 0.4946 unpaired t-test |
| Rebound depolarization expressed as a percentage (rebound as a percentage of the total step size, calculated as the difference between the max hyperpolarization and baseline potential, %) | 50.77 | 6.156 | 28 | 6 | 19.28 | 1.961 | 28 | 6 | **<0.0001** Mann-Whitney |
| Properties | DA-Raptor WT (VTA) | | | | DA-Raptor KO (VTA) | | | | WT vs KO |
| | Mean | SEM | n (cells) | n (mice) | Mean | SEM | n (cells) | n (mice) | p-value/ test |
| Series resistance (mOhms) | 4.216 | 0.282 | 22 | 8 | 4.472 | 0.337 | 22 | 6 | 0.8617 Mann-Whitney |
| Membrane resistance (mOhms) | 463.7 | 34.41 | 22 | 8 | 485.9 | 40.15 | 22 | 6 | 0.6769 unpaired t-test |
| Membrane capacitance (pF) | 64.88 | 5.117 | 22 | 8 | 43.21 | 2.581 | 22 | 6 | **0.0005** unpaired t-test |
| Resting membrane potential (mV) | −52.03 | 2.130 | 22 | 8 | −51.42 | 3.046 | 22 | 6 | 0.8697 unpaired t-test |
| Rheobase (current when first action potentials occur, pA) | 69.32 | 6.565 | 22 | 8 | 63.64 | 16.75 | 22 | 6 | **0.0190** Mann-Whitney |
| Action potential threshold (mV) | −26.06 | 1.458 | 22 | 8 | −32.35 | 1.391 | 22 | 6 | **0.0039** Mann-Whitney |
| Action potential peak (maximum membrane potential, mV) | 17.65 | 2.180 | 22 | 8 | 10.50 | 1.708 | 22 | 6 | **0.0134** unpaired t-test |
| Action potential height (change in membrane potential from the start of the AP to maximum depolarization, mV) | 54.72 | 2.378 | 22 | 8 | 53.49 | 1.735 | 22 | 6 | 0.6766 unpaired t-test |
| Afterhyperpolarization (minimum membrane potential after the AP, mV) | −51.09 | 1.619 | 22 | 8 | −54.24 | 0.875 | 22 | 6 | 0.0942 unpaired t-test |

*Table 1 continued on next page*

*Table 1 continued*

| Properties | DA-Raptor WT (VTA) | | | | DA-Raptor KO (VTA) | | | | WT vs KO |
|---|---|---|---|---|---|---|---|---|---|
| | Mean | SEM | n (cells) | n (mice) | Mean | SEM | n (cells) | n (mice) | p-value/ test |
| Afterhyperpolarization (change in membrane potential from the start of the AP to maximum hyperpolarization, mV) | 14.02 | 0.826 | 22 | 8 | 11.83 | 0.703 | 22 | 6 | 0.0983 Mann-Whitney |
| Maximum hyperpolarization in response to –100 pA (from ~–70 mV in response to a 2 second –100 pA current step, mV) | –128.9 | 4.016 | 22 | 8 | –148.4 | 4.986 | 21 | 6 | **0.0038** unpaired t-test |
| Sag component in response to –100 pA (maximum hyperpolarization minus the steady state membrane potential in the last 50ms of the current step, mV) | 14.34 | 3.756 | 22 | 8 | 15.78 | 3.329 | 21 | 6 | 0.9329 Mann-Whitney |
| Sag component expressed as a percentage (sag component as a percentage of the total step size, calculated as the difference between the max hyperpolarization and baseline potential, %) | 20.33 | 3.465 | 22 | 8 | 19.18 | 3.388 | 21 | 6 | 0.3660 Mann-Whitney |
| Rebound depolarization in response to –100 pA (baseline membrane potential minus the maximum depolarization within 500ms of the end of the current step, mV) | 2.414 | 0.595 | 22 | 8 | 2.027 | 0.753 | 21 | 6 | 0.6219 Mann-Whitney |
| Rebound depolarization expressed as a percentage (rebound as a percentage of the total step size, calculated as the difference between the max hyperpolarization and baseline potential, %) | 3.857 | 0.777 | 22 | 8 | 2.944 | 1.177 | 21 | 6 | 0.2513 Mann-Whitney |

in DA-Rictor KO mice compared to controls (*Figure 9d–f*). However, the average somatic TH level per mouse was not different between genotypes (WT, 100.00%+/-2.50; KO, 101.10%+/-6.58, p=0.9905, Welch's t-test, normalized values). In VTA neurons, deletion of *Rictor* led to a small but significant decrease in somatic TH levels (*Figure 9g–i*). This small reduction in TH did not translate into significant changes in the tissue content of DA or DOPAC in either the dorsal or ventral striatum of DA-Rictor KO mice (*Figure 9j and k*). Together these data show that reduced DA availability may not account for the DA release deficits in DA-Rictor KO mice.

## Concurrent deletion of *Rptor* and *Rictor* exacerbates deficits in DA production, release and re-uptake

Given our observation that deletion of *Rictor* led to decreased p-S6 levels in DA neurons and that DA-Rictor KO mice shared several phenotypes with DA-Raptor KO mice, we asked whether the effects of *Rictor* deletion were mediated by mTORC1 suppression. To test this, we generated double KO mice in which both *Rptor* and *Rictor* were selectively deleted from DA neurons (*Rptor*$^{fl/fl}$;*Rictor*$^{fl/fl}$;*Sl-c6a3*$^{IREScre/+}$;*ROSA26*$^{Ai9/+}$, *Figure 10a–c*). We compared mice with deletion of both *Rptor* and *Rictor* (DA-Raptor KO;Rictor KO, 'Double KO') to mice with deletion of *Rptor* alone (DA-Raptor KO;Rictor WT). With this strategy, we could investigate whether loss of Rictor induced further changes in DA neuron properties beyond those caused by Raptor loss. We found that deletion of *Rictor* did not further decrease p-S6 in DA neurons at our level of detection (*Figure 10b–f and i–k*). This suggests that the reduction in p-S6 in DA-Rictor KO mice (see *Figure 2i and k*) was likely due to suppression of mTORC1 pathway activity. Soma size was slightly reduced in SNc, but not VTA, neurons from the Double KO mice (*Figure 10g and l*). Compared to *Rptor* deletion alone, concurrent deletion of *Rictor* and *Rptor* further reduced TH expression in both SNc (*Figure 10h*) and VTA (*Figure 10m*) neurons.

To determine how DA neurotransmission was affected by combined suppression of mTORC1 and mTORC2 signaling, we measured evoked DA release with FCV. We found that compared to *Rptor* loss alone, deletion of *Rptor* and *Rictor* resulted in a further reduction in peak-evoked DA levels (*Figure 10n–p*). Evoked DA release was reduced across all striatal sampling sites by ~20% in Double KO mice compared to DA-Raptor KO;Rictor WT mice (*Figure 10o and p*). Thus, concurrent mTORC1 and mTORC2 inhibition resulted in additional release deficits at dorsal striatum sites 1–3, which were unaffected by mTORC2 suppression alone (see *Figure 7a–c*). We examined the kinetics of DA re-uptake and found a significant reduction in the re-uptake rate in both the dorsal striatum (*Figure 10q and r*) and the NAc core (*Figure 10s, t*) of Double KO mice. In this case, combined mTORC1 and mTORC2 suppression caused slower [DA]$_o$ uptake in the NAc core, which was not

**Table 2.** Summary of the electrophysiology properties of DA-Rictor WT and DA-Rictor KO SNc and VTA DA neurons.

| Properties | DA-Rictor WT (SNc) | | | | DA-Rictor KO (SNc) | | | | WT vs KO |
|---|---|---|---|---|---|---|---|---|---|
| | Mean | SEM | n (cells) | n (mice) | Mean | SEM | n (cells) | n (mice) | p-value/ test |
| Series resistance (mOhms) | 2.979 | 0.249 | 21 | 4 | 2.573 | 0.171 | 27 | 8 | 0.1249 Mann-Whitney |
| Membrane resistance (mOhms) | 110.9 | 14.05 | 21 | 4 | 147.9 | 14.52 | 27 | 8 | 0.0772 Mann-Whitney |
| Membrane capacitance (pF) | 114.4 | 6.016 | 21 | 4 | 85.41 | 5.165 | 27 | 8 | **0.0006** Mann-Whitney |
| Resting membrane potential (mV) | −55.46 | 1.337 | 21 | 4 | −52.27 | 1.565 | 27 | 8 | 0.1419 unpaired t-test |
| Rheobase (current when first action potentials occur, pA) | 101.2 | 11.11 | 21 | 4 | 112.0 | 12.96 | 27 | 8 | 0.7414 Mann-Whitney |
| Action potential threshold (mV) | −32.94 | 1.520 | 21 | 4 | −31.34 | 0.974 | 27 | 8 | 0.9180 Mann-Whitney |
| Action potential peak (maximum membrane potential, mV) | 25.46 | 1.647 | 21 | 4 | 23.61 | 1.429 | 27 | 8 | 0.3964 unpaired t-test |
| Action potential height (change in membrane potential from the start of the AP to maximum depolarization, mV) | 66.31 | 1.253 | 21 | 4 | 62.61 | 1.616 | 27 | 8 | 0.1590 Mann-Whitney |
| Afterhyperpolarization (minimum membrane potential after the AP, mV) | −59.49 | 2.276 | 21 | 4 | −58.46 | 1.416 | 27 | 8 | 0.5467 Mann-Whitney |
| Afterhyperpolarization (change in membrane potential from the start of the AP to maximum hyperpolarization, mV) | 18.65 | 1.517 | 21 | 4 | 19.45 | 1.063 | 27 | 8 | 0.6582 unpaired t-test |
| Maximum hyperpolarization in response to −100 pA (from ~−70 mV in response to a 2 second −100 pA current step, mV) | −94.91 | 2.162 | 21 | 4 | −101.0 | 2.079 | 27 | 8 | **0.0490** unpaired t-test |
| Sag component in response to −100 pA (maximum hyperpolarization minus the steady state membrane potential in the last 50ms of the current step, mV) | 11.90 | 0.992 | 21 | 4 | 14.74 | 1.129 | 27 | 8 | 0.0772 Mann-Whitney |
| Sag component expressed as a percentage (sag component as a percentage of the total step size, calculated as the difference between the max hyperpolarization and baseline potential, %) | 45.96 | 2.701 | 21 | 4 | 43.63 | 2.172 | 27 | 8 | 0.5006 unpaired t-test |
| Rebound depolarization in response to −100 pA (baseline membrane potential minus the maximum depolarization within 500ms of the end of the current step, mV) | 14.17 | 4.150 | 21 | 4 | 12.02 | 1.363 | 27 | 8 | 0.8049 Mann-Whitney |
| Rebound depolarization expressed as a percentage (rebound as a percentage of the total step size, calculated as the difference between the max hyperpolarization and baseline potential, %) | 52.27 | 12.87 | 21 | 4 | 37.70 | 4.586 | 27 | 8 | 0.4212 Mann-Whitney |

| Properties | DA-Rictor WT (VTA) | | | | DA-Rictor KO (VTA) | | | | WT vs KO |
|---|---|---|---|---|---|---|---|---|---|
| | Mean | SEM | n (cells) | n (mice) | Mean | SEM | n (cells) | n (mice) | p-value/ test |
| Series resistance (mOhms) | 3.608 | 0.303 | 22 | 4 | 4.239 | 0.446 | 20 | 6 | 0.2418 unpaired t-test |
| Membrane resistance (mOhms) | 485.9 | 46.92 | 22 | 4 | 606.2 | 43.56 | 20 | 6 | **0.0418** Mann-Whitney |
| Membrane capacitance (pF) | 60.06 | 3.977 | 22 | 4 | 47.28 | 4.014 | 20 | 6 | **0.0014** Mann-Whitney |
| Resting membrane potential (mV) | −49.72 | 1.700 | 22 | 4 | −50.45 | 2.267 | 20 | 6 | 0.9305 Mann-Whitney |
| Rheobase (current when first action potentials occur, pA) | 100.0 | 20.76 | 20 | 4 | 57.50 | 8.331 | 20 | 6 | **0.0299** Mann-Whitney |
| Action potential threshold (mV) | −26.61 | 2.422 | 20 | 4 | −24.32 | 1.568 | 19 | 6 | 0.4376 unpaired t-test |
| Action potential peak (maximum membrane potential, mV) | 19.25 | 2.135 | 20 | 4 | 14.25 | 2.431 | 19 | 6 | 0.1298 unpaired t-test |

*Table 2 continued on next page*

*Table 2 continued*

| Properties | DA-Rictor WT (VTA) | | | | DA-Rictor KO (VTA) | | | | WT vs KO |
|---|---|---|---|---|---|---|---|---|---|
| | Mean | SEM | n (cells) | n (mice) | Mean | SEM | n (cells) | n (mice) | p-value/ test |
| Action potential height (change in membrane potential from the start of the AP to maximum depolarization, mV) | 60.49 | 2.102 | 20 | 4 | 54.17 | 2.303 | 19 | 6 | **0.0496** unpaired t-test |
| Afterhyperpolarization (minimum membrane potential after the AP, mV) | −54.66 | 1.367 | 20 | 4 | −52.49 | 0.858 | 19 | 6 | 0.1935 unpaired t-test |
| Afterhyperpolarization (change in membrane potential from the start of the AP to maximum hyperpolarization, mV) | 14.15 | 1.135 | 20 | 4 | 12.57 | 0.883 | 19 | 6 | 0.2829 unpaired t-test |
| Maximum hyperpolarization in response to −100 pA (from ~−70 mV in response to a 2 second −100 pA current step, mV) | −135.2 | 5.016 | 22 | 4 | −143.5 | 4.862 | 20 | 6 | 0.2433 unpaired t-test |
| Sag component in response to −100 pA (maximum hyperpolarization minus the steady state membrane potential in the last 50ms of the current step, mV) | 14.20 | 1.589 | 22 | 4 | 16.73 | 3.457 | 20 | 6 | 0.7178 Mann-Whitney |
| Sag component expressed as a percentage (sag component as a percentage of the total step size, calculated as the difference between the max hyperpolarization and baseline potential, %) | 22.13 | 2.189 | 22 | 4 | 20.35 | 3.093 | 20 | 6 | 0.3888 Mann-Whitney |
| Rebound depolarization in response to −100 pA (baseline membrane potential minus the maximum depolarization within 500ms of the end of the current step, mV) | 3.42 | 0.573 | 22 | 4 | 2.720 | 0.661 | 20 | 6 | 0.4252 unpaired t-test |
| Rebound depolarization expressed as a percentage (rebound as a percentage of the total step size, calculated as the difference between the max hyperpolarization and baseline potential, %) | 6.044 | 1.290 | 22 | 4 | 3.515 | 0.808 | 20 | 6 | 0.1638 Mann-Whitney |

observed with either mTORC1 (see *Figure 6f and g*) or mTORC2 (see *Figure 7f and g*) inhibition alone. Together these data suggest that combined inhibition of mTORC1 and mTORC2 signaling leads to widespread disruption of DA neuron output, highlighting the importance of mTOR signaling for DA neuron function.

## Discussion

DA modulates the activity of neural circuits throughout the brain and dysregulation of DA signaling is linked to a variety of neuropsychiatric and neurodegenerative disorders. The mTOR signaling network is a central regulator of multiple aspects of DA neuron growth and metabolism. Specifically, mTOR signaling has been shown to be important for therapeutic responses in PD, adaptations to drugs of abuse, and DA system changes induced by ASD-linked mutations (*Dadalko et al., 2015b*; *Kim et al., 2012*; *Kosillo and Bateup, 2021*; *Kosillo et al., 2019*; *Liu et al., 2018b*; *Malagelada et al., 2010*; *Mazei-Robison et al., 2011*; *Neasta et al., 2014*; *Zhu et al., 2019*). Investigating the DA neuron properties that are controlled by the mTOR pathway, and how these are regulated by the two mTOR complexes, is therefore important for understanding how mTOR regulates DA neuron biology in both health and disease.

Here we genetically manipulated the activity of mTORC1 and mTORC2 in DA neurons and defined how these manipulations affected key cellular properties. Our results reveal several main findings. First, we find that chronic inhibition of mTORC1 signaling strongly impacts multiple aspects of DA neuron biology with the most pronounced effects on SNc DA neurons. Second, we find that reducing mTORC2 signaling results in some phenotypes that overlap with mTORC1 inhibition. However, these phenotypes are milder and more selective for VTA neurons. Third, we observe that concurrent disruption of both mTORC1 and mTORC2 signaling strongly compromises DA neuron output, underscoring the importance of mTOR for proper DA system function. *Table 3* presents a summary of the main findings of this study.

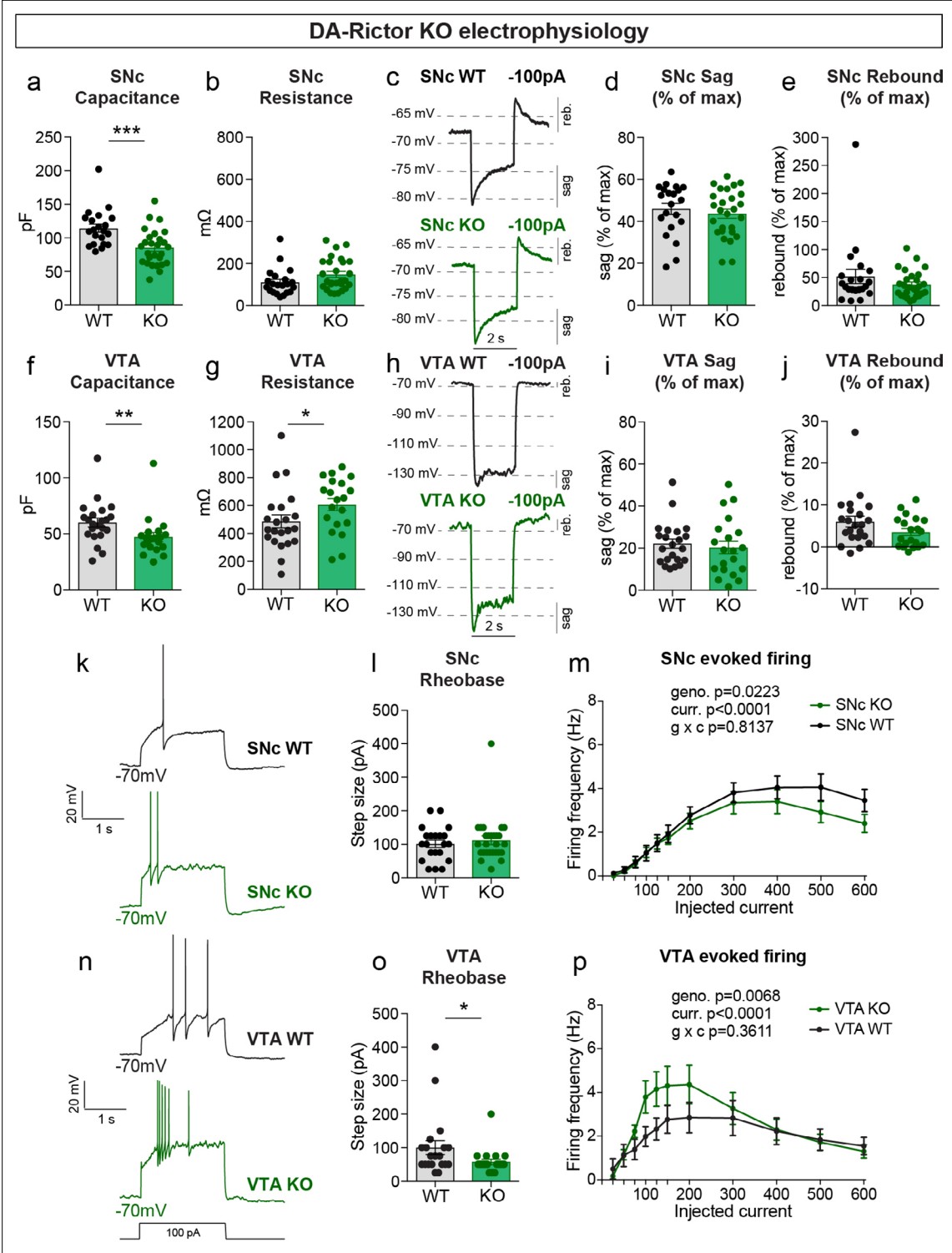

**Figure 5.** mTORC2 inhibition increases the excitability of VTA DA neurons. (**a,b**) Mean ± SEM membrane capacitance (**a**) and membrane resistance (**b**) of SNc DA neurons. DA-Rictor WT in black: n=21 neurons from four mice, DA-Rictor KO in green: n=27 neurons from eight mice. Capacitance (**a**), ***p=0.0006. Resistance (**b**), P=0.0772 (**b**). Mann–Whitney tests. (**c**) Example current-clamp recordings from SNc DA neurons of the indicated genotypes in response to a −100 pA current step. (**d,e**) Mean ± SEM sag (**d**) and rebound (**e**) amplitude expressed as a percentage of the maximum hyperpolarization from baseline in SNc DA neurons. DA-Rictor WT in black: n=21 neurons from four mice, DA-Rictor KO in green: n=27 neurons from eight mice. Sag (**d**), p=0.5006, two-tailed t-test. Rebound (**e**), p=0.4212, Mann–Whitney test. (**f,g**) Mean ± SEM membrane capacitance (**f**) and membrane resistance (**g**) of VTA DA neurons. DA-Rictor WT in black: n=22 neurons from four mice, DA-Rictor KO in green: n=20 neurons from six

*Figure 5 continued on next page*

*Figure 5 continued*

mice. Capacitance (**f**), **p=0.0014. Resistance (**g**), *p=0.0418. Mann-Whitney tests. (**h**) Example current-clamp recordings from VTA DA neurons of the indicated genotypes in response to a −100 pA current step. (**i,j**) Mean ± SEM sag (**i**) and rebound (**j**) amplitude expressed as a percentage of the maximum hyperpolarization from baseline in VTA DA neurons. DA-Rictor WT in black: n=22 neurons from four mice, DA-Rictor KO in green: n=20 neurons from six mice. Sag (**i**), p=0.3880. Rebound (**j**), P=0.1638. Mann–Whitney tests. (**k**) Examples of action potential firing elicited with a +100 pA current step in SNc DA neurons of the indicated genotypes. (**l**) Mean ± SEM rheobase of SNc DA neurons calculated as the current at which action potentials were first elicited. DA-Rictor WT in black: n=21 neurons from four mice, DA-Rictor KO in green: n=27 neurons from eight mice, p=0.7414, Mann–Whitney test. (**m**) Input-output curves showing the firing frequency of SNc DA neurons in response to current steps of increasing amplitude. Data are displayed as mean ± SEM. DA-Rictor WT in black: n=21 neurons from four mice, DA-Rictor KO in green: n=27 neurons from eight mice. Two-way ANOVA p values are shown. Sidak's multiple comparisons tests: $p_{25pA} > 0.9999$, $p_{50pA} > 0.9999$, $p_{75pA} > 0.9999$, $p_{100pA} > 0.9999$, $p_{125pA} > 0.9999$, $p_{150pA} > 0.9999$, $p_{200pA} > 0.9999$, $p_{300pA} = 0.9948$, $p_{400pA} = 0.9341$, $p_{500pA} = 0.2913$, $p_{600pA} = 0.4672$. (**n**) Examples of action potential firing elicited with a +100 pA current step in VTA DA neurons of the indicated genotypes. (**o**) Mean ± SEM rheobase of VTA DA neurons calculated as the current at which action potentials were first elicited. DA-Rictor WT in black: n=20 neurons from four mice, DA-Rictor KO in green: n=20 neurons from six mice, *p=0.0299, Mann–Whitney test. (**p**) Input-output curves showing the firing frequency of VTA DA neurons in response to current steps of increasing amplitude. Data are displayed as mean ± SEM. DA-Rictor WT in black: n=22 neurons from four mice, DA-Rictor KO in green: n=20 neurons from six mice. Two-way ANOVA p values are shown. Sidak's multiple comparisons tests: $p_{25pA} > 0.9999$, $p_{50pA} > 0.9999$, $p_{75pA} = 0.9842$, $p_{100pA} = 0.2640$, $p_{125pA} = 0.2515$, $p_{150pA} = 0.4871$, $p_{200pA} = 0.5233$, $p_{300pA} > 0.9999$, $p_{400pA} > 0.9999$, $p_{500pA} > 0.9999$, $p_{600pA} > 0.9999$. For all bar graphs, dots represent values for individual neurons. See also *Table 2*.

## mTORC1 is a potent regulator of DA neuron morphology and physiology

Here we found that chronic inhibition of mTORC1 signaling led to pronounced and widespread changes in the structure and function of midbrain DA neurons. Specifically, deletion of *Rptor* caused hypotrophy of multiple cellular compartments including the soma, dendrites and axons of both SNc and VTA neurons. Unexpectedly, we found that the striatal matrix compartment showed a greater DA axon innervation deficit compared to putative patch regions within DA-Raptor KO mice. While selective degeneration of patch-projecting SNc DA neurons in PD models has been reported (*Crittenden and Graybiel, 2011*; *Sgobio et al., 2017*), we found that Raptor loss more strongly reduced DA projections to the striatal matrix. Two possible explanations for this relate to the developmental timing and trophic support of DA axons in patch versus matrix. Dopaminergic innervation of striatal patches is established first, forming so-called 'DA islands' (*Moon Edley and Herkenham, 1984*; *Fishell and van der Kooy, 1987*; *Graybiel, 1984*). It could be the case that matrix-projecting axonal segments, which develop later, fail to properly form in the context of limited molecular resources due to mTORC1 suppression. Further, compared to the matrix compartment, patches provide additional trophic support in the form of glial cell derived neurotrophic factor (GDNF) (*López-Martín et al., 1999*; *Oo et al., 2005*). This, in turn, could enable DA neurons with limited resources to better maintain axonal branches in striatal patches of DA-Raptor KO mice.

The somatodendritic hypotrophy of DA-Raptor KO neurons is consistent with a large body of literature demonstrating the importance of mTORC1 signaling in controlling cell size, including prior work showing that reduction of Raptor expression causes somatodendritic hypotrophy in other neuron types (*Angliker et al., 2015*; *McCabe et al., 2020*; *Urbanska et al., 2012*). While the mechanisms downstream of mTORC1 that regulate DA neuron size are not well understood, cellular hypotrophy of Raptor KO neurons may result from reduced protein and lipid synthesis, chronic activation of autophagy, and/or changes to the cytoskeleton (*Fingar et al., 2002*). A reduction in dendritic arborization would have implications for the number and/or type of synaptic inputs onto DA-Raptor KO neurons, which may impact their response to activation of upstream circuits.

One of the most pronounced phenotypes observed in DA-Raptor KO neurons was a major impairment in their ability to release DA in response to electrical stimulation. Compromised DA release was likely due to impaired DA synthesis since total tissue DA content and TH expression were reduced in DA-Raptor KO mice. It is possible that mTORC1 is a direct regulator of TH protein synthesis as prior studies have shown that the translation of TH is controlled by cAMP (*Chen et al., 2008*) and that cAMP increases translation via mTORC1 (*Kim et al., 2010*). When mTORC1 is hyperactive, as in DA-Tsc1 KO mice, TH levels are increased (*Kosillo et al., 2019*). Therefore, TH protein levels are bidirectionally regulated by mTORC1 signaling. Taken together, our data show that chronic developmental suppression of mTORC1 signaling in DA neurons leads to pronounced cellular hypotrophy, aberrant striatal innervation, and severe deficits in DA production and release.

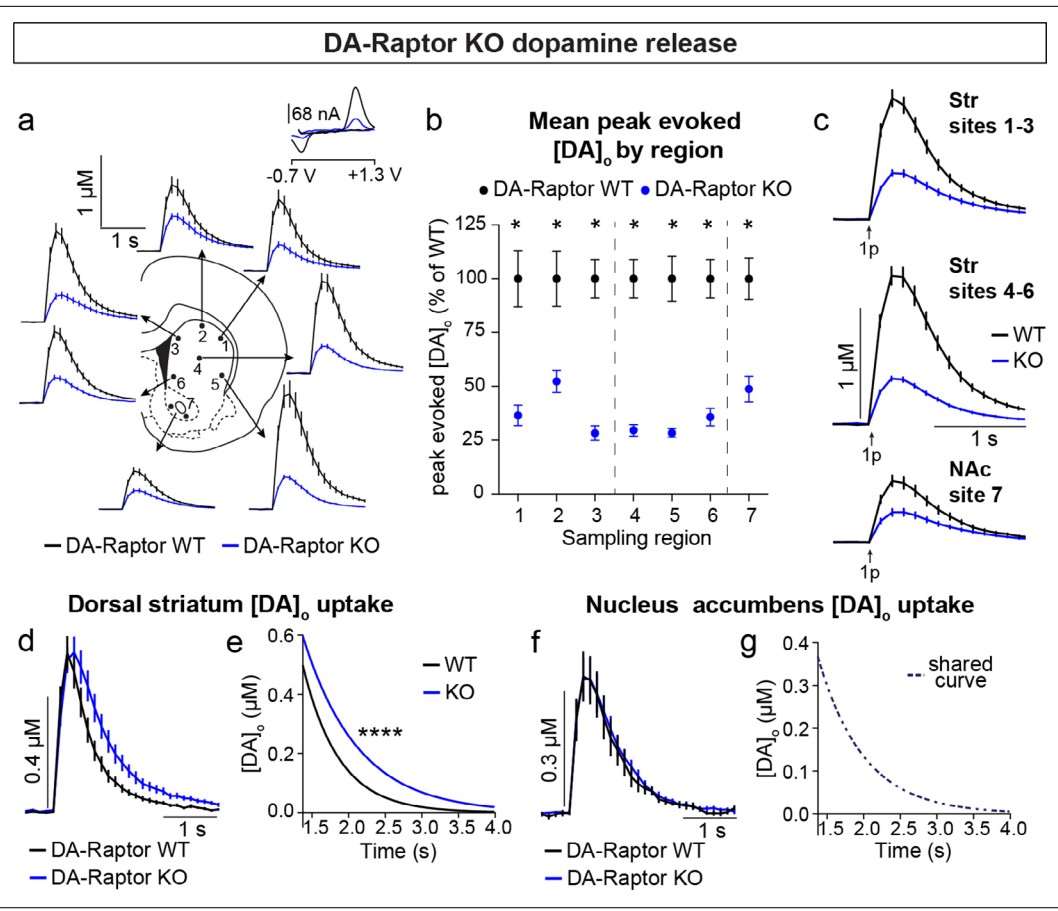

**Figure 6.** Deletion of *Rptor* reduces evoked DA release across all striatal regions. (**a**) Mean ± SEM $[DA]_o$ versus time evoked from different striatal subregions by a single electrical pulse. Traces are an average of n=20 (sites #1,2,4), n=18 (sites #3,6), n=19 (site #5) and n=34 (site #7) transients per sampling region from five mice per genotype. DA-Raptor WT in black, DA-Raptor KO in blue. Inset, typical cyclic voltammograms show characteristic DA waveform. (**b**) Mean ± SEM peak $[DA]_o$ by striatal subregion expressed as a percentage of WT (sampling region numbers correspond to the sites in panel **a**). n=20 transients (sites #1,2,4), n=18 transients (sites #3,6), n=19 transients (site #5) and n=34 transients (site #7) per sampling region from five mice per genotype. $*p_1=0.0002$, $*p_3 <0.0001$, $*p_7 <0.0001$, Wilcoxon's two-tailed t-tests; $*p_2=0.0047$, $*p_4 <0.0001$, $*p_5 <0.0001$, $*p_6 <0.0001$, paired two-tailed t-tests. (**c**) Mean ± SEM $[DA]_o$ versus time averaged across all transients from three striatal territories, dorsal striatum (Str) (sites #1–3), central-ventral striatum (sites #4–6) and NAc core (site #7). Traces are an average of n=58 transients (sites #1–3), n=57 transients (sites #4–6) and n=34 transients (site #7) per sampling territory from five mice per genotype. Statistical comparisons for the peak evoked $[DA]_o$ between genotypes by sub-region: $****p_{Str\ 1-3}<0.0001$, $****p_{NAc\ 7}<0.0001$, Wilcoxon's two-tailed t-tests; $****p_{Str\ 4-6}<0.0001$, paired two-tailed t-test. (**d**) Mean ± SEM $[DA]_o$ versus time from concentration- and site-matched FCV transients recorded in dorsal and central-ventral striatum (sites #1–6). DA-Raptor WT average of n=20 transients from five mice per genotype, DA-Raptor KO average of n=22 transients from five mice per genotype. (**e**) Single-phase exponential decay curve-fit of the falling phase of concentration- and site-matched striatal DA transients (from panel **d**). X-axis starts 375ms after stimulation. DA-Raptor WT average of n=20 transients from five mice per genotype, DA-Raptor KO average of n=22 transients from five mice per genotype. $****p<0.0001$, least-squares curve-fit comparison. (**f**) Mean ± SEM $[DA]_o$ versus time from concentration- and site-matched FCV transients recorded in NAc core (site #7). DA-Raptor WT average of n=9 transients from five mice per genotype, DA-Raptor KO average of n=10 transients from five mice per genotype. (**g**) Single-phase exponential decay curve-fit of the falling phase of concentration-matched NAc DA transients (from panel **f**). X-axis starts 375ms after stimulation onset. DA-Raptor WT average of n=9 transients from five mice per genotype, DA-Raptor KO average of n=10 transients from five mice per genotype. p=0.4377, least-squares curve-fit comparison. See also *Figure 6—figure supplement 1*.

The online version of this article includes the following figure supplement(s) for figure 6:

**Figure supplement 1.** mTORC1 inhibition reduces evoked striatal DA release in response to high frequency stimulation (related to *Figure 6*).

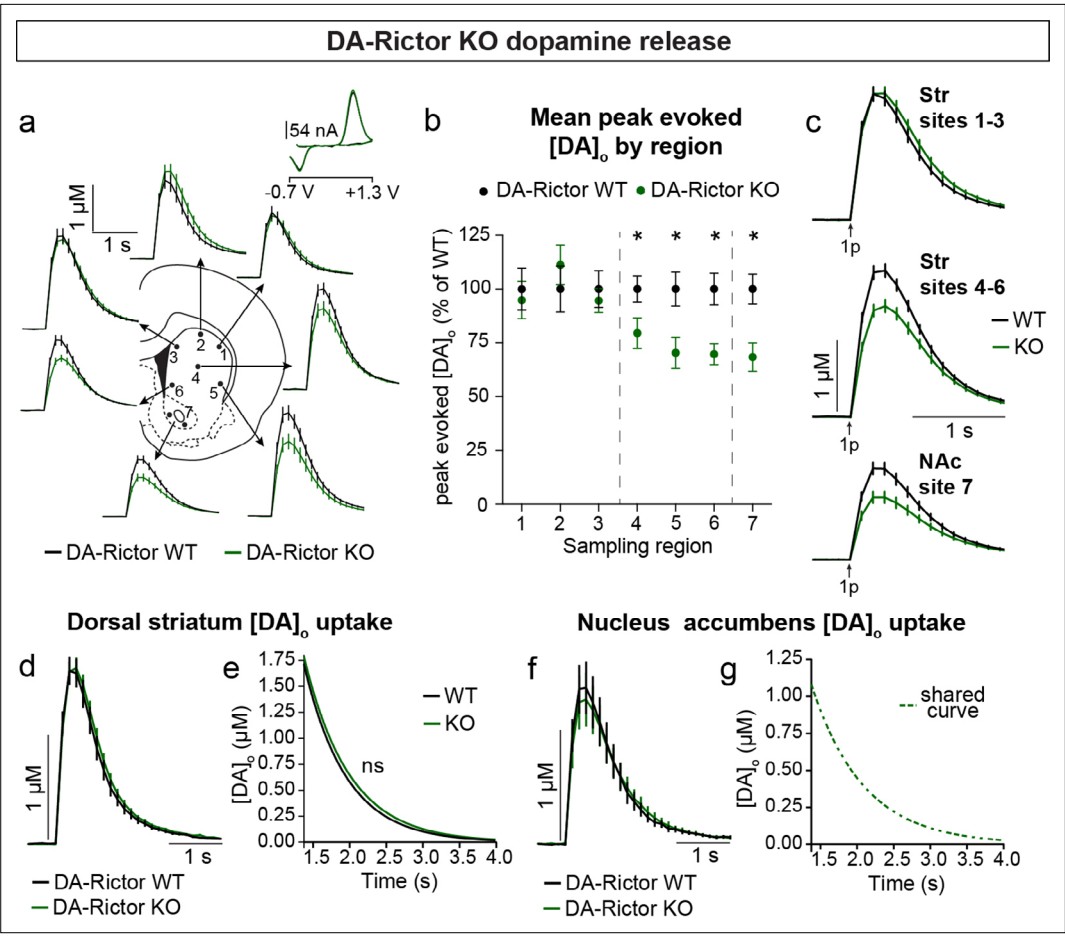

**Figure 7.** Deletion of *Rictor* reduces evoked DA release in central-ventral striatum and NAc. (**a**) Mean ± SEM [DA]$_o$ versus time evoked from different striatal subregions by a single electrical pulse. Traces are an average of n=20 (sites #1–6) or n=40 (site #7) transients per sampling region from five mice per genotype. DA-Rictor WT in black, DA-Rictor KO in green. Inset, typical cyclic voltammograms show characteristic DA waveform. (**b**) Mean ± SEM peak [DA]$_o$ by striatal subregion expressed as a percentage of WT (sampling region numbers correspond to the sites in panel **a**). n=20 (sites #1–6) or n=40 (site #7) transients per sampling region from five mice per genotype. $p_1$=0.5185, $p_2$=0.2858, $p_3$=0.4538, *$p_4$=0.0111, *$p_5$=0.0161, *$p_6$=0.0003, paired two-tailed t-tests; *$p_7$ <0.0001, Wilcoxon's two-tailed t-test. (**c**) Mean ± SEM [DA]$_o$ versus time averaged across all transients from three striatal territories, dorsal striatum (Str) (sites #1–3), central-ventral striatum (sites #4–6) and NAc core (site #7). Traces are an average of n=60 transients (sites #1–3 and #4–6) or n=40 transients (site #7) per sampling territory from five mice per genotype. Statistical comparisons for the peak evoked [DA]$_o$ between genotypes by sub-region: $p_{Str\ 1-3}$=0.9672, paired two-tailed t-test; ****$p_{Str\ 4-6}$<0.0001, ****$p_{NAc\ 7}$<0.0001, Wilcoxon's two-tailed t-tests. (**d**) Mean ± SEM [DA]$_o$ versus time from concentration- and site-matched FCV transients recorded in dorsal and central-ventral striatum (sites #1–6). DA-Rictor WT average of n=16 transients from five mice per genotype, DA-Rictor KO average of n=17 transients from five mice per genotype. (**e**) Single-phase exponential decay curve-fit of the falling phase of concentration- and site-matched striatal DA transients (from panel **d**). X-axis starts 375ms after stimulation. DA-Rictor WT average of n=16 transients from five mice per genotype, DA-Rictor KO average of n=17 transients from five mice per genotype. p=0.0594, least-squares curve-fit comparison. (**f**) Mean ± SEM [DA]$_o$ versus time from concentration- and site-matched FCV transients recorded in NAc core (site #7). DA-Rictor WT average of n=10 transients from five mice per genotype, DA-Rictor KO average of n=10 transients from five mice per genotype. (**g**) Single-phase exponential decay curve-fit of the falling phase of concentration-matched NAc DA transients (from panel **f**). X-axis starts 375ms after stimulation onset. DA-Rictor WT average of n=10 transients from five mice per genotype, DA-Rictor KO average of n=10 transients from five mice per genotype. p=0.8759, least-squares curve-fit comparison. See also *Figure 7—figure supplement 1*.

The online version of this article includes the following figure supplement(s) for figure 7:

**Figure supplement 1.** mTORC2 inhibition reduces evoked ventral striatal DA release in response to high frequency stimulation (related to *Figure 7*).

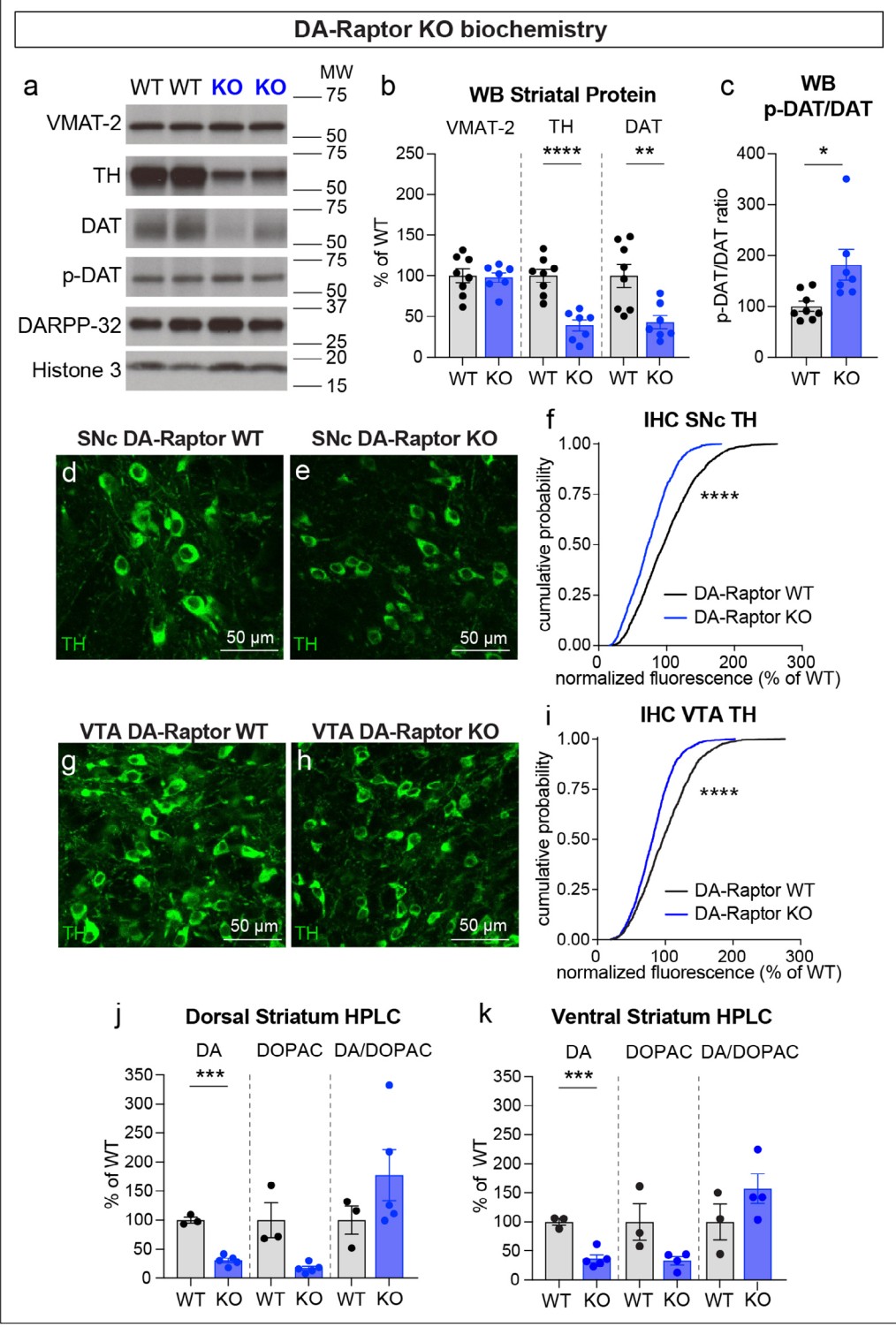

**Figure 8.** Deletion of *Rptor* reduces TH expression and DA synthesis. (**a**) Striatal lysates were harvested from DA-Raptor WT or KO mice. Representative western blots for vesicular monoamine transporter-2 (VMAT-2), tyrosine hydroxylase (TH), DA transporter (DAT), DAT phosphorylated on Thr53 (p-DAT), DARPP-32 and Histone 3. Two independent samples per genotype are shown. Observed molecular weight (MW) is noted on the right. See also *Figure 8—source data 1 and 2*. (**b**) Mean ± SEM striatal protein content of VMAT-2, TH and DAT. Dots represent values for individual mice (averaged from two samples per mouse). n=8 DA-Raptor WT mice in black and n=7 DA-Raptor KO mice in blue. VMAT-2, p=0.8476; TH, ****p<0.0001; DAT, **p=0.0053, Welch's two-tailed t-tests. (**c**)

*Figure 8 continued on next page*

*Figure 8 continued*

Mean ± SEM ratio of DAT phosphorylated at Thr53 (p-DAT) to total DAT protein, *p=0.0340, Welch's two-tailed t-test. (**d,e**) Representative confocal images of SNc neurons from DA-Raptor WT (**d**) or DA-Raptor KO (**e**) mice immunostained for TH, scale bars = 50 µm. (**f**) Cumulative distributions of TH levels in SNc DA neurons. DA-Raptor WT in black: n=1,024 neurons from three mice, DA-Raptor KO in blue: n=1045 neurons from three mice, ****p<0.0001, Kolmogorov–Smirnov test. (**g,h**) Representative confocal images of VTA neurons from DA-Raptor WT (**g**) or DA-Raptor KO (**h**) mice immunostained for TH, scale bars = 50 µm. (**i**) Cumulative distribution of TH levels in VTA DA neurons. DA-Raptor WT in black: n=1389 neurons from three mice, DA-Raptor KO in blue: n=1526 neurons from three mice, ****p<0.0001, Kolmogorov–Smirnov test. (**j,k**) Mean ± SEM total tissue content of DA, 3,4-dihydroxyphenylacetic acid (DOPAC) and the DOPAC/DA ratio per mouse assessed by HPLC from tissue punches from dorsal striatum (**j**) or ventral striatum (**k**). Dots represent values for individual mice (averaged from two samples per mouse). n=3 DA-Raptor WT mice in black and n=5 DA-Raptor KO mice in blue (n=4 DA-Raptor KO mice in blue for ventral striatum DOPAC). Dorsal striatum DA, ***p=0.0002, DOPAC, p=0.1084, DA/DOPAC ratio, p=0.1790, Welch's two-tailed t-tests. Ventral striatum DA, ***p=0.0005, DOPAC, p=0.1599, DA/DOPAC ratio, p=0.2190, Welch's two-tailed t-tests. See also *Figure 8—source data 3* for a summary of the raw HPLC measurement values.

The online version of this article includes the following source data for figure 8:

**Source data 1.** Western blot scans for the data presented in *Figure 8a–c* (DA-Raptor WT and KO).

**Source data 2.** Western blot scans for the data presented in *Figure 8a–c* (DA-Raptor WT and KO).

**Source data 3.** Summary of raw values for HPLC measurements for DA-Raptor WT and KO mice.

## Chronic mTORC1 inhibition or activation impairs striatal DA release via different mechanisms

Previous studies have investigated how activation of mTORC1 signaling affects DA neuron structure and function. Chronic mTORC1 activation has been achieved via deletion of genes encoding upstream negative regulators such as Pten or Tsc1 (*Diaz-Ruiz et al., 2009*; *Kosillo et al., 2019*) or by constitutive activation of positive regulators such as Akt and Rheb (*Cheng et al., 2011*; *Kim et al., 2012*; *Ries et al., 2006*). In general, these studies find opposing phenotypes to what we observed following Raptor loss. Namely, mTORC1 activation leads to an increase in somatodendritic size and complexity, enlarged axon terminal area, enhanced axonal sprouting, and increased TH expression and DA synthesis (*Cheng et al., 2011*; *Diaz-Ruiz et al., 2009*; *Kim et al., 2012*; *Kosillo et al., 2019*; *Ries et al., 2006*). Nevertheless, with constitutive mTORC1 activation in Tsc1 KO neurons, evoked DA release in the dorsal striatum is reduced to a similar degree as with Raptor loss (*Kosillo et al., 2019*). This is similar to phenotypes in mice with selective deletion of D2 autoreceptors from DA neurons, which have increased axonal arbors in the dorsal striatum but impaired evoked DA release (*Giguère et al., 2019*). In the case of Tsc1 loss, the evoked DA release deficits may result from structural alterations in DA axon terminals that render vesicular DA release less efficient, since the total tissue DA content is elevated (*Kosillo et al., 2019*). By contrast, with Raptor loss, the DA release deficits are likely due to reduced TH levels and impaired DA synthesis, causing broad deficits across all DA neurons. Therefore, as observed in other neuronal types (*Angliker et al., 2015*), both activation and inhibition of mTORC1 can produce similarly detrimental outcomes, albeit by distinct mechanisms.

## mTORC2 selectively alters the excitability and output of VTA DA neurons

The functions of mTORC2 have been less well studied; however, studies using mice with conditional deletion of *Rictor* from different neuron types have uncovered roles for mTORC2 in synaptic transmission, synaptic plasticity, actin dynamics and somatodendritic morphology (*Huang et al., 2013*; *McCabe et al., 2020*; *Thomanetz et al., 2013*; *Urbanska et al., 2012*; *Zhu et al., 2018*). In the DA system, conditional deletion of *Rictor* in VTA neurons using AAV-Cre moderately decreased the soma size of DA neurons (*Mazei-Robison et al., 2011*), similar to our observations. It was also shown that changes in VTA neurons induced by chronic morphine treatment are driven by suppression of mTORC2, as they can be phenocopied by *Rictor* deletion and rescued by *Rictor* overexpression (*Mazei-Robison et al., 2011*). The cellular changes observed with chronic morphine, namely reduced soma size, increased excitability, and reduced dopaminergic output to the NAc, are very similar to what we observed with DA neuron-specific deletion of *Rictor*. Here, we also examined SNc DA-Rictor

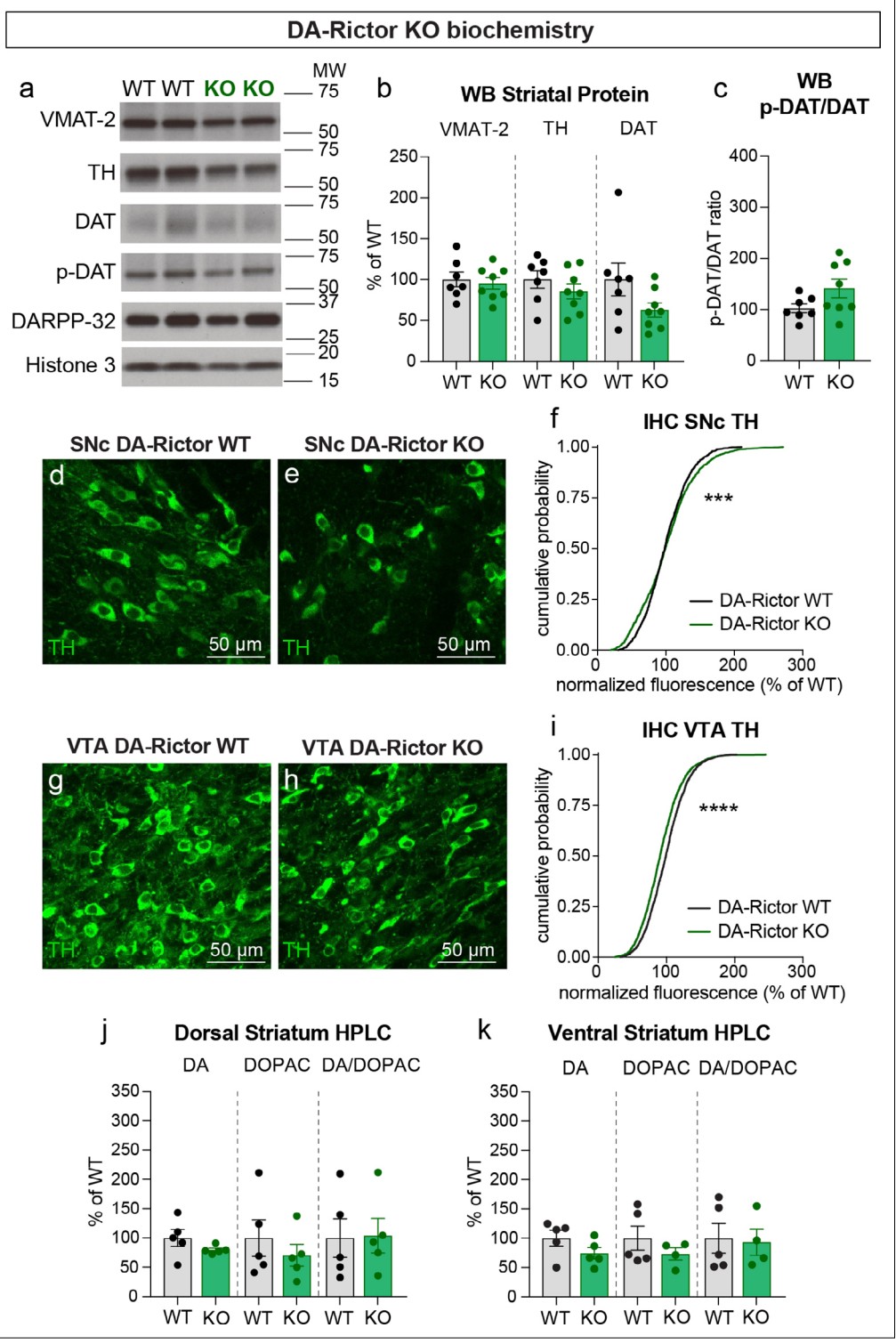

**Figure 9.** Deletion of *Rictor* does not alter striatal DA synthesis. (**a**) Striatal lysates were harvested from DA-Rictor WT or KO mice. Representative western blots for vesicular monoamine transporter-2 (VMAT-2), tyrosine hydroxylase (TH), DA transporter (DAT), DAT phosphorylated at Thr53 (p-DAT), DARPP-32 and Histone 3. Observed molecular weight (MW) is noted on the right. See also *Figure 9—source data 1–2*. (**b**) Mean ± SEM striatal protein content of VMAT-2, TH and DAT. Dots represent values for individual mice (averaged from two samples per mouse). n=7 DA-Rictor WT mice in black and n=8 DA-Rictor KO mice in green. VMAT-2, p=0.6882; TH, p=0.3281; DAT, p=0.1282 Welch's two-tailed t-tests. (**c**) Mean ± SEM ratio of DAT phosphorylated at Thr53 (p-DAT) to total

*Figure 9 continued on next page*

*Figure 9 continued*

DAT protein, p=0.0838, Welch's two-tailed t-test. (**d,e**) Representative confocal images of SNc neurons from DA-Rictor WT (**d**) or DA-Rictor KO (**e**) mice immunostained for TH, scale bars = 50 µm. (**f**) Cumulative distributions of TH levels in SNc DA neurons. DA-Rictor WT in black: n=1280 neurons from three mice, DA-Rictor KO in green: n=1550 neurons from four mice, ***p=0.0009, Kolmogorov–Smirnov test. (**g,h**) Representative confocal images of VTA neurons from DA-Rictor WT (**g**) or DA-Rictor KO (**h**) mice immunostained for TH, scale bars = 50 µm. (**i**) Cumulative distribution of TH levels in VTA DA neurons. DA-Rictor WT in black: n=1968 neurons from three mice, DA-Rictor KO in green: n=2370 neurons from four mice, ****p<0.0001, Kolmogorov–Smirnov test. (**j,k**) Mean ± SEM total tissue content of DA, 3,4-dihydroxyphenylacetic acid (DOPAC) and the DOPAC/DA ratio per mouse assessed by HPLC from tissue punches from dorsal striatum (**j**) or ventral striatum (**k**). Dots represent values for individual mice (averaged from two samples per mouse). n=5 DA-Rictor WT mice in black and n=5 DA-Rictor KO mice in green (n=4 DA-Rictor KO mice in green for ventral striatum DOPAC). Dorsal striatum DA, p=0.2356, DOPAC, p=0.4438, DA/DOPAC ratio, p=0.9289, Welch's two-tailed t-tests. Ventral striatum DA, p=0.1648, DOPAC, p=0.2896, DA/DOPAC ratio, p=0.8455, Welch's two-tailed t-tests. See also *Figure 9—source data 3* for a summary of the raw HPLC measurement values.

The online version of this article includes the following source data for figure 9:

**Source data 1.** Western blot scans for the data presented in *Figure 9a–c* (DA-Rictor WT and KO).

**Source data 2.** Western blot scans for the data presented in *Figure 9a–c* (DA-Rictor WT and KO).

**Source data 3.** Summary of raw values for HPLC measurements for DA-Rictor WT and KO mice.

KO neurons and found that like VTA neurons, they exhibit modest changes in somatodendritic architecture and reduced striatal axon density. However, in contrast to VTA neurons, *Rictor* KO SNc neurons did not show major changes in intrinsic excitability or DA release properties. Thus, VTA neurons are preferentially affected by suppression of mTORC2 signaling and SNc DA neurons are more resilient to mTORC2 perturbations.

The morphological changes observed in DA-Rictor KO neurons at the level of cell bodies and axons were similar to DA-Raptor KO neurons, albeit smaller in magnitude. However, we found that loss of Rictor differentially affected the dendritic morphology of DA neurons, causing increased proximal dendrite branching with no changes in total dendritic length. This increase in proximal dendrite branching is reminiscent of the phenotype caused by Rictor loss from cerebellar Purkinje cells (PCs). In PCs, deletion of *Rictor* decreases soma size and axon diameter but increases the number of primary dendrites (*Thomanetz et al., 2013*). It was suggested that the effects of Rictor loss on dendritic morphology may be due to alterations in PKC levels or activity as PKC is a phosphorylation target of mTORC2 and a known regulator of actin cytoskeletal dynamics (*Angliker and Rüegg, 2013*). Further studies could investigate the potential role of PKC signaling in DA neurons in mediating the effects of Rictor loss on cytoskeletal organization.

## Crosstalk between mTORC1 and mTORC2 signaling

In this study, we used a genetic strategy to delete obligatory components of mTORC1 and mTORC2 that are specific for each complex. While deletion of *Rptor* or *Rictor* may initially disrupt the formation or activity of each complex selectively, we find that these manipulations do not exclusively affect the signaling of just one complex. Specifically, we find that deletion of *Rictor* from DA neurons leads to a small but significant reduction in p-S6 levels, a canonical mTORC1 read-out. While several studies have reported no change in mTORC1 phosphorylation targets with deletion or reduction of *Rictor* (*Chen et al., 2019*; *Mazei-Robison et al., 2011*; *Thomanetz et al., 2013*), other studies have shown reduced p-S6, similar to what we observed here (*McCabe et al., 2020*; *Urbanska et al., 2012*). This discrepancy could arise due to neuron type, developmental timing, total duration, and/or extent of Rictor loss. Mechanistically, reduced mTORC2 activity could lead to inhibition of mTORC1 via suppression of Akt activity. Studies in non-neuronal cell lines show that Akt normally increases mTORC1 activity via phosphorylation of Tsc2 and removal of the Tsc1/2 complex's inhibitory control over Rheb (*Inoki et al., 2002*; *Manning et al., 2002*). In addition, Akt can phosphorylate the mTORC1 component PRAS40, which also releases its negative control over mTORC1 activity (*Wang et al., 2012*). In the case of mTORC2 suppression, Akt signaling is decreased; therefore, the positive effects of Akt on mTORC1 activity are reduced, potentially leading to lower levels of mTORC1 signaling.

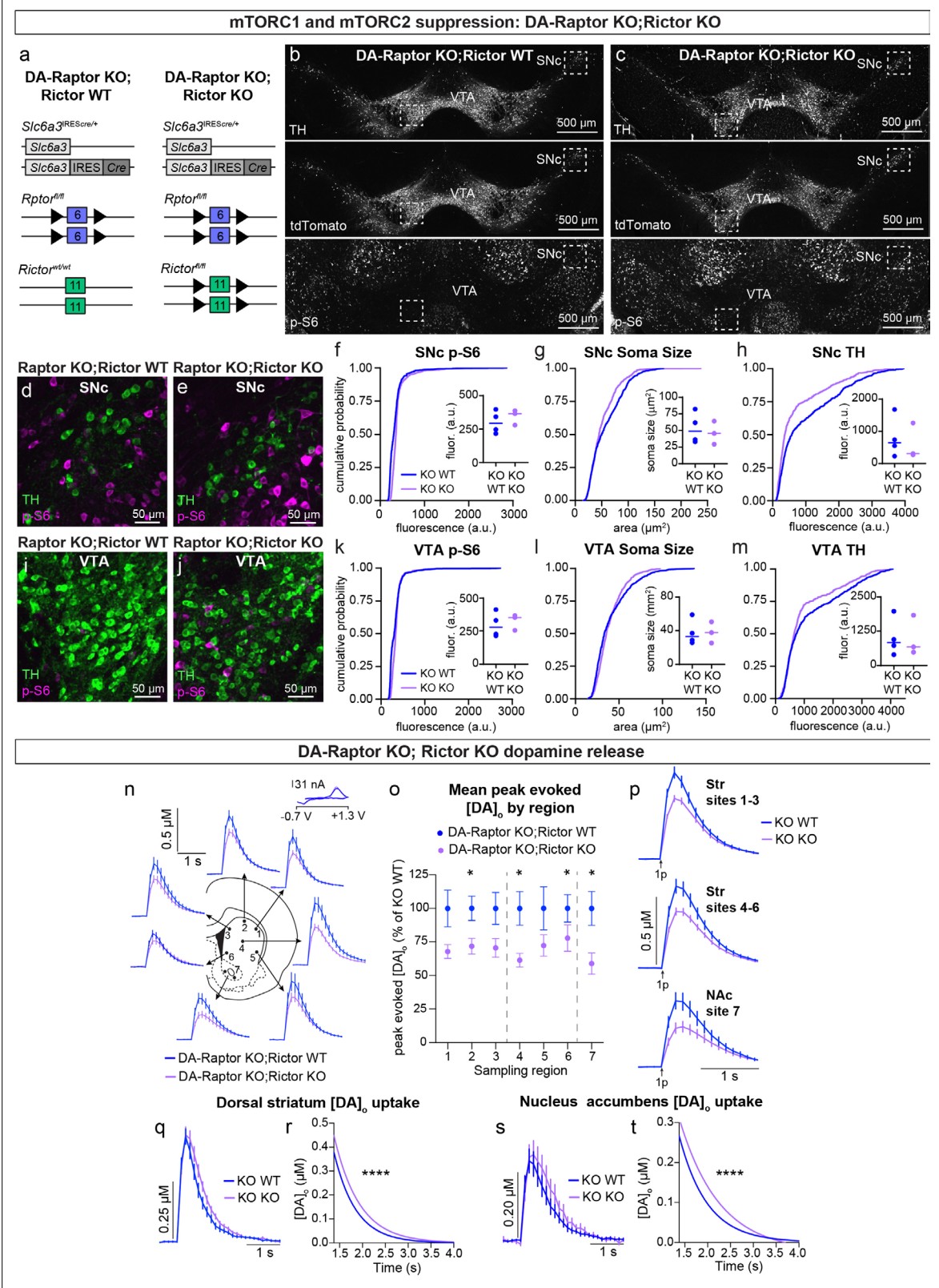

**Figure 10.** Double knock-out of *Rptor* and *Rictor* strongly impairs DA synthesis, release, and re-uptake. (**a**) Schematic of the genetic strategy to delete either *Rptor* alone or *Rptor* and *Rictor* selectively from DA neurons. Numbered boxes represent exons and triangles represent loxP sites. (**b,c**) Representative confocal images of midbrain sections from DA-Raptor KO;Rictor WT (**b**) and DA-Raptor KO;Rictor KO (**c**) mice with DA neurons visualized by Cre-dependent tdTomato expression and tyrosine hydroxylase (TH) immunostaining. Bottom panels show p-S6 (Ser240/244) immunostaining. Scale

*Figure 10 continued on next page*

*Figure 10 continued*

bars = 500 µm. (**d,e**) Higher magnification merged images of the boxed SNc regions in panels b and c, showing TH (green) and p-S6 (magenta), scale bars = 50 µm. (**f–h**) Cumulative distributions of SNc DA neuron p-S6 levels (**f**), soma area (**g**) and TH expression (**h**). DA-Raptor KO;Rictor WT in blue: n=1040 neurons from four mice, DA-Raptor KO;Rictor KO in purple: n=780 neurons from three mice, ****p<0.0001, Kolmogorov–Smirnov tests. Inset plots show the median per mouse, dots represent individual mice. Lines represent median per genotype. a.u.=arbitrary units. (**i,j**) Higher magnification merged images of the boxed VTA regions in panels b and c, showing TH (green) and p-S6 (magenta), scale bars = 50 µm. (**k–m**) Cumulative distributions of VTA DA neuron p-S6 levels (**k**), soma area (**l**) and TH expression (**m**). DA-Raptor KO;Rictor WT in blue: n=1,172 neurons from four mice, DA-Raptor KO;Rictor KO in purple: n=879 neurons from three mice. ****$P$<0.0001, Kolmogorov–Smirnov tests. Inset plots show the median per mouse, dots represent individual mice. Lines represent median per genotype. (**n**) Mean ± SEM [DA]$_o$ versus time evoked from different striatal subregions elicited by a single electrical pulse. Traces are an average of n=24 (sites #1–6) or n=48 (site #7) transients per sampling region from six mice per genotype. DA-Raptor KO;Rictor WT in blue and DA-Raptor KO;Rictor KO in purple. Inset, typical cyclic voltammograms show characteristic DA waveform. (**o**) Mean ± SEM peak [DA]$_o$ by striatal subregion expressed as a percentage of DA-Raptor KO;Rictor WT (sampling region numbers correspond to the sites in panel **n**). n=24 (sites #1–6) or n=48 (site #7) transients per sampling region from six mice per genotype. $p_1$=0.1208, $p_3$=0.0691, $p_5$=0.1875, *$p_6$=0.0425, *$p_7$=0.0002, Wilcoxon's two-tailed t-tests; *$p_2$=0.0238, *$p_4$=0.0098, paired two-tailed t-tests. (**p**) Mean ± SEM [DA]$_o$ versus time averaged across all transients from three striatal territories, dorsal striatum (Str) (sites #1–3), central-ventral striatum (sites #4–6) and NAc core (site #7). Traces are an average of n=72 transients (sites #1–3 and #4–6) or n=48 transients (site #7) per sampling territory from six mice per genotype. Statistical comparisons for the peak evoked [DA]$_o$ between genotypes by sub-region: ***$p_{Str\,1-3}$=0.0004, **$p_{Str\,4-6}$=0.0019, ***$p_{NAc\,7}$ = 0.0002, Wilcoxon's two-tailed t-tests. (**q**) Mean ± SEM [DA]$_o$ versus time from concentration- and site-matched FCV transients recorded in dorsal and central-ventral striatum (sites #1–6). DA-Raptor KO;Rictor WT ("KO WT") in blue and DA-Raptor KO;Rictor KO ("KO KO") in purple. DA-Raptor KO;Rictor WT average of n=26 transients from six mice per genotype, DA-Raptor KO;Rictor KO average of n=24 transients from six mice per genotype. (**r**) Single-phase exponential decay curve-fit of the falling phase of concentration- and site-matched striatal DA transients (from panel **q**). X-axis starts 375ms after stimulation. DA-Raptor KO;Rictor WT average of n=26 transients from six mice per genotype, DA-Raptor KO;Rictor KO average of n=24 transients from six mice per genotype. ****$P$<0.0001, least-squares curve-fit comparison. (**s**) Mean ± SEM [DA]$_o$ versus time from concentration- and site-matched FCV transients recorded in NAc core (site #7). DA-Raptor KO;Rictor WT ("KO WT") in blue and DA-Raptor KO;Rictor KO ("KO KO") in purple. DA-Raptor KO;Rictor WT average of n=10 transients from five mice per genotype, DA-Raptor KO; Rictor KO average of n=9 transients from five mice per genotype. (**t**) Single-phase exponential decay curve-fit of the falling phase of concentration-matched NAc DA transients (from panel **s**). X-axis starts 375ms after stimulation onset. DA-Raptor KO;Rictor WT average of n=10 transients from five mice per genotype, DA-Raptor KO;Rictor KO average of n=9 transients from five mice per genotype. ****p<0.0001, least-squares curve-fit comparison.

In addition to changes in p-S6, we found a small reduction in soma size in DA-Rictor KO mice. Reduced soma size is a phenotype observed in multiple cell types with mTORC1 suppression. Therefore, it is possible that reduced soma size in Rictor KO DA neurons is due to inhibition of mTORC1 activity. Consistent with this idea, activation of Akt or p70S6K in cultured hippocampal neurons can restore p-S6 levels and rescue somatodendritic hypotrophy in cells treated with *Rictor* shRNA (*Urbanska et al., 2012*). This suggests that some of the phenotypes associated with mTORC2 disruption may be driven by reduced mTORC1 activity. However, the reduced soma size of DA-Rictor KO neurons could arise from an mTORC1-independent mechanism. In support of the latter, chronic morphine reduces the soma size of VTA DA neurons via an mTORC2-dependent but mTORC1-independent mechanism (*Mazei-Robison et al., 2011*). The mechanisms underlying the signaling interactions between mTORC1 and mTORC2 in DA neurons and other cell types are not well understood and warrant further investigation.

## ProExM-TrailMap enables quantitative measurement of DA axon density and size

A key challenge with measuring the anatomical properties of DA axons has been difficulty in resolving the structure of individual DA axons within densely innervated regions, such as the striatum. In a classic study, single neuron reconstruction was performed to demonstrate the highly elaborate projection patterns of individual DA axon arbors within the striatum (*Matsuda et al., 2009*). Super resolution imaging approaches have been applied to DA axons to address this challenge including 3D-SIM (*Liu et al., 2018a*). Crittenden and colleagues used expansion microscopy (ProExM) to resolve the unique architecture of dendrite-axon bundles in the substantia nigra (*Crittenden et al., 2016*). Here we extended this approach to the analysis of striatal DA axons by combining ProExM and volumetric light sheet imaging with the computational method TrailMap (*Friedmann et al., 2020*), which allows automatic identification and segmentation of axonal processes. We first validated this workflow in DA-Tsc1 KO mice, which have established DA axon terminal hypertrophy as measured previously by EM (*Kosillo et al., 2019*). Importantly, EM cannot readily assess axon density since measurements are made on cross-sections of individual axon terminals. In turn, quantifying bulk striatal TH

**Table 3.** Summary of the major dopaminergic phenotypes associated with mTOR signaling manipulations.

| complex | direction of change | component | role in mTOR signaling | gene | type of manipulation | cell type affected | manipulation timing | pS6 levels | DA neuron number | DA neuron somato-dendritic morphology | DA neuron intrinsic excitability | DA neuron axonal morphology | DA release | TH expression | DA tissue content | DAT expression and function | Reference |
|---|---|---|---|---|---|---|---|---|---|---|---|---|---|---|---|---|---|
| mTORC1 | ↑ | Tsc1 | mTORC1 signaling suppression | Tsc1 gene (Tsc1 fl/fl) | homozygous conditional deletion | dopaminergic neurons (DAT-Cre driver line) | embryonic | Increased in SNc and VTA (IHC) | No changes in SNc or VTA (IHC) | Increased soma size and dendritic complexity in SNc and VTA (IHC) | Increased rheobase, decreased excitability in SNc and VTA neurons | Hypertrophic axon terminals, greater enlargement in dorsal than ventral striatum (EM) | ~60% decreased evoked release, more pronounced in dorsal than ventral striatum (FCV, electrical stim) | Increased in midbrain SNc and VTA and striatum (IHC, WB) | Increased in dorsal and ventral striatum (HPLC) | Increased striatal DAT function (FCV) | *Kosillo et al., 2019* |
| mTORC1 | ↓ | Rptor + Tsc1 | mTORC1 signaling suppression + mTORC1 obligatory protein | Rptor + Tsc1 genes (Tsc1 fl/fl; Rptor fl/fl) | homozygous conditional deletion | dopaminergic neurons (DAT-Cre driver line) | embryonic | Abolished in SNc and VTA (IHC) | No changes in SNc or VTA (IHC) | Decreased soma size in SNc and VTA (IHC) | — | — | ~60% decreased evoked release (FCV, electrical stim) | Decreased in midbrain SNc and VTA (IHC) | — | — | *Kosillo et al., 2019* |
| mTORC1 | ↓ | Rptor | mTORC1 obligatory protein | Rptor gene (Rtpor fl/fl) | homozygous conditional deletion | dopaminergic neurons (DAT-Cre driver line) | embryonic | Abolished in SNc and VTA (IHC) | Reduced in SNc and VTA (stereology) | Decreased soma size and dendritic complexity in SNc and VTA (IHC) | Decreased rheobase, increased excitability in SNc and VTA neurons at low currents, depolarization block with large depolarizing inputs | Reduced total axonal volume in dorsal and ventral striatum; reduction in both DA axon density and radius, matrix compartment most affected | ~60% decreased evoked release, dorsal and ventral striatum affected equally (FCV, electrical stim) | Decreased in midbrain SNc and VTA (IHC) and striatum (WB) | Decreased in dorsal and ventral striatum (HPLC) | Decreased striatal DAT function (FCV) and striatal DAT protein expression (WB) | this paper |

*Table 3 continued on next page*

Table 3 continued

| complex | direction of change | component | role in mTOR signaling | type of manipulation | cell type affected | manipulation timing | pS6 levels | DA neuron number | DA neuron somato-dendritic morphology | DA neuron intrinsic excitability | DA neuron axonal morphology | DA release | TH expression | DA tissue content | DAT expression and function | Reference |
|---|---|---|---|---|---|---|---|---|---|---|---|---|---|---|---|---|
| mTORC2 | ↓ | Rictor gene (Rictor fl/fl) | mTORC2 obligatory protein | homozygous conditional deletion | dopaminergic neurons (DAT-Cre driver line) | embryonic | Decreased in SNc and VTA (IHC) | No changes in SNc or VTA (stereology) | Decreased soma size in SNc and VTA, increased proximal dendrite complexity in SNc and VTA (IHC) | Unchanged excitability in SNc, decreased rheobase and increased excitability in VTA neurons | Reduced total axonal volume in dorsal striatum, reduced axonal density in both dorsal and ventral striatum | ~25% decreased in central-ventral and ventral striatum (FCV, electrical stim) | Unchanged in striatum (WB), slightly decreased in VTA (IHC) | Unchanged in dorsal and ventral striatum (HPLC) | Unchanged DAT function (FCV) and striatal DAT protein expression (WB) | this paper |
| total mTOR | ↓ | Rptor + Rictor genes (Rtpro fl/fl; Rictor fl/fl) | mTORC1 mTORC2 obligatory proteins | homozygous conditional deletion | dopaminergic neurons (DAT-Cre driver line) | embryonic | Abolished in SNc and VTA (IHC) | - | Decreased soma size in SNc and VTA (IHC) | — | — | ~20% decrease in evoked release in dorsal and ventral striatum compared to DA-Rptor KO alone (FCV, electrical stim) | Decreased in midbrain SNc and VTA (IHC) | — | Decreased DAT function in both striatum and NAc core (FCV) | this paper |

Abbreviations: DA Dopamine; DAT Dopamine Active Transporter; EM Electron Microscopy; FCV Fast-Scan Cyclic Voltammetry; HPLC High Performance Liquid Chromatography; KO Knock-out; NAc Nucleus Accumbens; SNc Substantia Nigra pars compacta; TH Tyrosine Hydroxylase; VTA Ventral Tegmental Area; WB Western Blot; IHC Immunohistochemistry; — no measurement

immunofluorescence as a proxy of axon density is confounded by potential changes in TH, or other fluorescent reporter expression, which may be dynamically regulated (*Chen et al., 2008*). ProExM-TrailMap overcomes these challenges and enables accurate quantification of striatal DA axon volume to determine whether alterations in this measure are due to changes in axon density, size, or both.

### Summary and relevance to disorders affecting the DA system

Our findings reveal that mTORC1 manipulations have the most pronounced effects on DA neuron structure and function, while the consequences of mTORC2 disruption are more modest. Nonetheless, balanced mTOR signaling is critically important to dopaminergic output, which is compromised by either chronic up- or down-regulation of mTORC1 or suppression of mTORC2 signaling. An important consideration is that the developmental timing and total duration of mTOR signaling manipulation will likely influence how changes in mTOR affect DA output. For example, altering mTOR signaling in embryonic development, as done in this study, likely leads to reduced DA output early in development, enabling post-synaptic circuits to compensate for low dopaminergic tone. This may allow for normal motor behavior in the face of significant deficits in DA output (*Delignat-Lavaud et al., 2021*; *Kosillo et al., 2019*). By contrast, it is known that reducing DA output in adulthood leads to significant motor impairments, as observed in PD (*Iancu et al., 2005*).

In addition, short- versus long-term mTOR signaling manipulations may lead to distinct outcomes. For example, in PD, acute upregulation of mTOR signaling via constitutively active Akt or Rheb may be beneficial to cell survival and axonal growth but prolonged activation, as with *Tsc1* or *Pten* deletion, is detrimental to DA output (*Diaz-Ruiz et al., 2009*; *Domanskyi et al., 2011*; *Kosillo et al., 2019*; *Zhu et al., 2019*). Similarly, acute inhibition of mTOR using rapamycin can suppress synthesis of pro-apoptotic proteins and may promote removal of misfolded proteins and damaged organelles by increasing autophagy (*Hernandez et al., 2012*; *Malagelada et al., 2010*). This may improve cell survival in response to neurotoxic injury. However, our results show that chronic mTORC1 suppression is detrimental to the long-term health and output of DA neurons. These factors will need to be taken into account when considering mTOR modulators as potential therapies.

In summary, this work, together with prior studies, demonstrates the importance of mTORC1 and mTORC2 signaling for proper DA neuron morphology, physiology, and output and reveals that unbalanced mTOR signaling can profoundly alter DA neuron structure and function.

## Materials and methods

### Key resources table

| Reagent type (species) or resource | Designation | Source or reference | Identifiers | Additional information |
|---|---|---|---|---|
| Genetic reagent (*Mus musculus*) | *Rptor*fl/fl | Jackson Laboratory | Stock #: 013188 | Male and female;C57Bl/6J background strain |
| Genetic reagent (*Mus musculus*) | *Rictor*fl/fl | Jackson Laboratory | Stock #: 020649 | Male and female;C57Bl/6 J, 129 SvJae and BALB/cJ mixed background strain |
| Genetic reagent (*Mus musculus*) | *Slc6a3*IREScre | Jackson Laboratory | Stock #: 0066600 | Male and female;C57Bl/6 J, CD1, 129 SvJae and BALB/cJ mixed background strain |
| Genetic reagent (*Mus musculus*) | *ROSA26*Ai9/+ | Jackson Laboratory | Stock #: 007909 | Male and female;C57Bl/6 J, CD1, 129 SvJae and BALB/cJ mixed background strain |
| Antibody | Anti-tyrosine hydroxylase (mouse monoclonal) | Immunostar | Cat #: 22,941 | (1:1000) IHC (1:2000) WB |
| Antibody | Anti-phospho-S6 ribosomal protein (rabbit monoclonal) | Cell Signaling | Cat #: 5,364 S | (1:1000) IHC |
| Antibody | Anti-RFP (rabbit polyclonal) | Rockland | Cat #: 600-401-379 | (1:500) IHC |
| Antibody | Anti-mu opioid receptor (rabbit polyclonal) | EMD Millipore | Cat #: AB5511 | (1:1000) IHC |

*Continued on next page*

*Continued*

| Reagent type (species) or resource | Designation | Source or reference | Identifiers | Additional information |
|---|---|---|---|---|
| Antibody | Goat anti-mouse Alexa 488 secondary (goat polyclonal) | Thermo Fisher | Cat #: A11001 | (1:500) IHC |
| Antibody | Goat anti-mouse Alexa 633 secondary (goat polyclonal) | Thermo Fisher | Cat #: A21050 | (1:500) IHC |
| Antibody | Goat anti-rabbit Alexa 488 secondary (goat polyclonal) | Thermo Fisher | Cat #: A11008 | (1:500) IHC |
| Antibody | Goat anti-rabbit Alexa 546 secondary (goat polyclonal) | Thermo Fisher | Cat #: A11035 | (1:500) IHC |
| Antibody | Goat anti-rabbit Alexa 633 secondary (goat polyclonal) | Thermo Fisher | Cat #: A11034 | (1:500) IHC |
| Antibody | Anti-DARPP32 (rabbit monoclonal) | Cell Signaling | Cat #: 2,306 S | (1:1500) WB |
| Antibody | Anti-Histone-3 (mouse monoclonal) | Cell Signaling | Cat #: 96C10 | (1:1500) WB |
| Antibody | Anti-VMAT2 (rabbit polyclonal) | Alomone Labs | Cat #: AMT-006 | (1:1000) WB |
| Antibody | Anti-DAT (mouse monoclonal) | Abcam | Cat #: 128,848 | (1:1000) WB |
| Antibody | Anti DAT phospho-T53 (rabbit polyclonal) | Abcam | Cat #: 183,486 | (1:1000) WB |
| Antibody | Goat anti-rabbit HRP secondary (goat polyclonal) | Bio-Rad | Cat #: 170–5046 | (1:5000) WB |
| Antibody | Goat anti-mouse HRP secondary (goat polyclonal) | Bio-Rad | Cat #: 170–5047 | (1:5000) WB |
| Commercial assay, kit | BCA assay | Fisher | Cat #: PI23227 | |
| Commercial assay, kit | Chemiluminesence substrate | Perkin-Elmer | Cat #: NEL105001EA | |
| Peptide, recombinant protein | Streptavidin Alexa 488 conjugate | Invitrogen | Cat #: S11223 | |
| Chemical compound, drug | GYKI 52466 dihydrochloride | Tocris | Cat #: 1,454 | (10 µM) final |
| Chemical compound, drug | (R)-CPP | Tocris | Cat #: 0247 | (10 µM) final |
| Chemical compound, drug | Picrotoxin | Tocris | Cat #: 1,128 | (50 µM) final |
| Software, algorithm | IMARIS | Oxford Instruments (https://imaris.oxinst.com) | | Version 9.2.1 |
| Software, algorithm | Stereoinvestigator | MBFBiosciences (https://www.mbfbioscience.com/stereo-investigator) | | |
| Software, algorithm | Igor Pro | Wavemetrics (https://www.wavemetrics.com/products/igorpro) | | Version 6.37 |
| Software, algorithm | GraphPad Prism | GraphPad Prism (https://graphpad.com) | | Version 6 + Version 8 |
| Software, algorithm | FIJI | ImageJ (http://imagej.nih.gov) | | |
| Software, algorithm | ScanImage | *Pologruto et al., 2003*; *Pologruto et al., 2019* (https://github.com/bernardosabatini/SabalabAcq) | | |
| Software, algorithm | TrailMap | https://github.com/kamodulin/TRAILMAP | https://doi.org/10.1073/pnas.191846511 | Modified from *Friedmann et al., 2020* |

| Reagent type (species) or resource | Designation | Source or reference | Identifiers | Additional information |
|---|---|---|---|---|
| Software, algorithm | Scale-invariant feature transform | https://github.com/kamodulin/expansion-microscopy | https://doi.org/10.1023/B:VISI.0000029664.99615.94 | Modified from *Lowe, 2004* |

## Mice

All animal procedures and husbandry were carried out in accordance with protocols approved by the University of California, Berkeley Institutional Animal Care and Use Committee (IACUC, protocol #AUP-2016-04-8684-2). Both male and female animals were used for all experiments. In general, young adult mice between 2 and 4 months of age were used, the ages of the animals used are indicated in the methods for each experiment. Mice were housed with same sex littermates in groups of 5–6 animals per cage and kept on a regular 12 hr light/dark cycle (lights on at 7am), with ad libitum access to food and water.

To generate DA neuron-specific Raptor-KO mice, *Rptor*fl/fl mice (Jackson Laboratories strain #013188) were bred to *Slc6a3*IREScre mice (Jackson Laboratories strain #0066600), on a C56Bl/6 J background. A *Rptor*fl/+;*Slc6a3*IREScre/+ × *Rptor*fl/+;*Slc6a3*IREScre/+ cross was used to generate experimental animals. Experimental mice were heterozygous for *Slc6a3*IREScre and either homozygous WT (*Rptor*+/+;*Slc6a3*IREScre/+, referred to as DA-Raptor WT) or homozygous floxed for *Rptor* (*Rptor*fl/fl;*Slc6a3*IREScre/+, referred to as DA-Raptor KO).

To generate DA neuron-specific Rictor-KO mice, *Rictor*fl/fl mice (Jackson Laboratories strain #020649) were bred to *Slc6a3*IREScre mice (Jackson Laboratories strain #0066600), on a mixed genetic background (C57Bl/6J, 129 SvJae and BALB/cJ). A *Rictor*fl/+;*Slc6a3*IREScre/+ × *Rictor*fl/+;*Slc6a3*IREScre/+ cross was used to generate experimental animals. Experimental mice were heterozygous for *Slc6a3*IREScre and either homozygous WT (*Rictor*+/+;*Slc6a3*IREScre/+, referred to as DA-Rictor WT) or homozygous floxed for *Rictor* (*Rictor*fl/fl;*Slc6a3*IREScre/+, referred to as DA-Rictor KO).

To generate DA neuron-specific double KO mice, *Rptor*fl/fl mice were crossed to *Rictor*fl/fl mice and their offspring were bred to *Slc6a3*IREScre mice. These mice were on mixed genetic background (C57Bl/6J, CD1, 129 SvJae and BALB/cJ). To generate experimental animals a *Rptor*fl/fl;*Rictor*fl/+;*Slc6a3*IREScre/+ × *Rptor*fl/fl;*Rictor*fl/+;*Slc6a3*IREScre/+ cross was used. Experimental mice were heterozygous for *Slc6a3*IREScre, homozygous for floxed *Rptor,* and either homozygous WT for *Rictor* (*Rptor*fl/fl;*Rictor*+/+;*Slc6a3*IREScre/+, referred to as DA-Raptor KO;Rictor WT) or homozygous floxed for *Rictor* (*Rptor*fl/fl;*Rictor*fl/fl;*Slc6a3*IREScre/+, referred to as DA-Raptor KO;Rictor KO).

For cell counting and axonal arborization experiments, *Rptor*fl/fl;*Slc6a3*IREScre/+ and *Rictor*fl/fl;*Slc6a3*IREScre/+ mice were bred to the Ai9 LSL-tdTomato Cre-reporter line (*ROSA26*Ai9) (Jackson Laboratories strain #007909). Mice used for experiments were heterozygous for *Slc6a3*IREScre and *ROSA26*Ai9.

## Stereological analysis of dopamine neuron number

Male and female P180-P240 mice were deeply anesthetized by isoflurane and transcardially perfused with ice cold 1 x PBS (~5–7 ml) followed by 4% paraformaldehyde (PFA) solution (Electron Microscopy Sciences: 15713) in 1 x PBS (~5–10 ml) using a perilstatic pump (Instech). The brains were removed and post-fixed by immersion in 4% PFA in 1 x PBS solution overnight at 4 °C. Brains were suspended in 30% sucrose in 0.1 M PB for cryoprotection. After brains descended to the bottom of the vial (typically 24–28 hr), 40 µm coronal sections of the midbrain were cut on a freezing microtome (American Optical AO 860), collected into serial wells, and stored at 4 °C in cryoprotectant (30% Ethylene glycol, 30% Glycerol, 0.2 M PB in ddH$_2$O).

Unbiased estimates of midbrain DA neuron number were obtained using the optical fractionator method as previously described (*Fortin et al., 2012*). The entire rostrocaudal extent of the midbrain was examined in 40-µm-thick coronal sections from DA-Raptor WT, DA-Raptor KO, DA-Rictor WT, and DA-Rictor KO mice that were heterozygous for the Ai9 tdTomato Cre-reporter allele by an observer who was blind to the genotype. A Zeiss AxioImager M1 with MicroBrightField BioSciences (MBFBiosciences) StereoInvestigator was used to count the number of tdTomato+ neurons in every sixth section at ×100 magnification using a 60 × 60 µm$^2$ counting frame. Sections counted corresponded to levels −2.54,−2.80, −3.08,−3.40, −3.64,−3.88, and −4.16 mm A/P (240 µm interval) with respect to bregma. A 10 µm optical dissector was placed 2 µm from the top of the section and counting sites were every 150 µM with a randomized start. Regions of interest were drawn separately for the VTA and SNc and

stereological estimates of the number of tdTomato+ neurons within each region were estimated using the StereoInvestigator software. Cell number estimates for the VTA and SNc regions were combined to calculate the total tdTomato+ cell number for each mouse.

## Neurobiotin-filled neuron reconstruction

Male and female mice (P56-P80) were deeply anesthetized by isoflurane, transcardially perfused with ice cold high $Mg^{2+}$ ACSF using a perilstatic pump (Instech) and decapitated. 275-µm-thick coronal midbrain slices were prepared on a vibratome (Leica VT1000 S) in ice cold high $Mg^{2+}$ ACSF containing in mM: 85 NaCl, 25 $NaHCO_3$, 2.5 KCl, 1.25 $NaH_2PO_4$, 0.5 $CaCl_2$, 7 $MgCl_2$, 10 glucose, and 65 sucrose. Slices recovered for 15 min at 34 °C followed by at least 50 min at room temperature (RT) in ACSF containing in mM: 130 NaCl, 25 $NaHCO_3$, 2.5 KCl, 1.25 $NaH_2PO_4$, 2 $CaCl_2$, 2 $MgCl_2$, and 10 glucose. All solutions were continuously bubbled with 95% $O_2$ and 5% $CO_2$. For whole cell recordings, 2.5–6 mΩ borosilicate glass pipettes (Sutter Instrument: BF150-86-7.5) were filled with a potassium-based internal solution containing in mM: 135 KMeSO₃, 5 KCl, 5 HEPES, 4 Mg-ATP, 0.3 Na-GTP, 10 phospho-creatine, 1 EGTA, and 4 mg/ml neurobiotin (Vector laboratories #SP-1120).

275 µm slices containing dopamine neurons filled with neurobiotin-containing internal solution (4 mg/ml) during patch-clamp experiments were fixed in 4% paraformaldehyde solution (Electron Microscopy Sciences: 15713) in 1 x PBS for 24–48 hr at 4 °C. With continuous gentle shaking, slices were washed in 1 x PBS 5 × 5 min and incubated with BlockAid blocking solution (Life Tech: B10710) for 1 hr at RT. Primary tyrosine hydroxylase antibody (1:1000, Immunostar: 22941) and streptavidin Alexa Flour 488 conjugate (1:750, Invitrogen: S11223) were applied overnight at 4 °C in 1 x PBS containing 0.25% (v/v) Trinton-X-100 (PBS-Tx). The following day, slices were washed in 1 x PBS 5 × 5 min, and Alexa-633 goat anti-mouse secondary antibody (1:500, ThermoFisher: A21050) in PBS-Tx was applied for 1 hr at RT. Slices were washed in cold 1 x PBS 5 × 5 min, mounted on SuperFrost slides (VWR: 48311–703) with the cell-containing side of the slice facing up, and coverslipped with either Prolong Gold antifade (Life Tech: P36935) or Vectashield hard-set (Vector Labs: H-1500) mounting media.

Filled cells in the mounted sections were imaged on a Zeiss LSM 880 NLO AxioExaminer confocal with 20 x/1.0 N.A. water immersion objective and 488 nm Argon laser using 1.53 µm steps to acquire a z-stack image spanning the entirety of the neurobiotin-filled cell body and dendritic arbor. 3D reconstruction of the cells was done using IMARIS 9.2.1 software (Bitplane) with automated filament tracing and manual editing. The mask generated by the automated filament tracing algorithm was continuously cross-referenced with the original z-stack image to ensure accuracy. Spurious segments created by the automated filament tracer due to background noise were removed, while processes with incomplete reconstruction were manually edited to incorporate missing segments.

## Immunohistochemistry

Male and female mice (P75-P90) were deeply anesthetized by isoflurane and transcardially perfused with ice cold 1 x PBS (~5–7 ml) followed by 4% paraformaldehyde (PFA) solution (Electron Microscopy Sciences: 15713) in 1 x PBS (~5–10 ml) using a perilstatic pump (Instech). The brains were removed and post-fixed by immersion in 4% PFA in 1 x PBS solution overnight at 4 °C. Brains were suspended in 30% sucrose in 0.1 M PB for cryoprotection. After brains descended to the bottom of the vial (typically 24–28 hr), 30 µm coronal sections of the midbrain and striatum were cut on a freezing microtome (American Optical AO 860), collected into serial wells, and stored at 4 °C in 1 x PBS containing 0.02% (w/v) sodium azide ($NaN_3$; Sigma Aldrich). All animals used for histology experiments were heterozygous for the Ai9 tdTomato Cre-reporter allele.

Brain sections for immunohistochemistry were batch processed to include matched control and experimental animals from a given line. Free-floating sections were washed with gentle shaking, 3 × 5 min in 1 x PBS followed by 1 hr incubation at RT with BlockAid blocking solution (Life Tech: B10710). Primary antibodies were applied at 4 °C in 1 x PBS containing 0.25% (v/v) Trinton-X-100 (PBS-Tx) for 48–72 hr to ensure penetration of the antibody throughout the slice. Sections were then washed with cold 1 x PBS 3 × 5 min and incubated for 1 hr at RT with secondary antibodies in PBS-Tx. Sections were washed in cold 1 x PBS 5 × 5 min, mounted on SuperFrost slides (VWR: 48311–703), and coverslipped with Vectashield hard-set (Vector Labs: H-1500) mounting media.

The following primary antibodies were used: tyrosine hydroxylase (1:1000, Immunostar: 22941) and phospho-S6 (Ser240/244) (1:1000, Cell Signaling: 5,364 S). The following secondary antibodies were used: Alexa-488 goat anti-mouse (1:500, Thermo Fisher: A-11001) and Alexa-633 goat anti-rabbit (1:500, Thermo Fisher: A-11034).

## Confocal microscopy

Images of 30 µm sections processed for immunohistochemistry were acquired using a Zeiss LSM 880 NLO AxioExaminer confocal microscope fitted with a motorized XY-stage for tile scanning. A 20 x/1.0 N.A. water objective was used to generate tile scans (scan area 425.1 × 425.1 µm per tile) of the entire midbrain (9 × 3 grid). 488 nm, 561 nm, and 633 nm lasers were used. Z-stack images captured the entire thickness of the slice at 1.70 µm steps. Laser power and gain settings were kept constant between control and experimental slice imaging.

Example images of 40 µm sections from mice used for cell counting experiments were acquired with an Olympus FV3000 confocal microscope using 405 and 561 nm lasers and a motorized stage for tile imaging. Z-stack images captured the entire thickness of the section at 1–2 µm steps for images taken with a 20 X air (Olympus #UCPLFLN20X) objective.

## Electrophysiology

Male and female adult mice (P56–P80) were deeply anesthetized by isoflurane, transcardially perfused with ice-cold high $Mg^{2+}$ artificial cerebrospinal fluid (aCSF) using a peristaltic pump (Instech) and decapitated. Coronal midbrain slices (275 µm thick) were prepared on a vibratome (Leica VT1000 S) in ice-cold high $Mg^{2+}$ aCSF containing (in mM): 85 NaCl, 25 NaHCO3, 2.5 KCl, 1.25 NaH2PO4, 0.5 CaCl2, 7 MgCl2, 10 glucose, and 65 sucrose. Slices were recovered for 15 min at 34 °C, followed by at least 50 min at room temperature (RT) in aCSF containing (in mM): 130 NaCl, 25 NaHCO3, 2.5 KCl, 1.25 NaH2PO4, 2 CaCl2, 2 MgCl2, and 10 glucose. All solutions were continuously bubbled with 95% $O_2$ and 5% $CO_2$.

Recordings were performed at 32 °C in the presence of AMPA, NMDA, and GABA-A receptor blockers (10 µM GYKI 52466 cat #1454, 10 µM CPP cat #0247, 50 µM picrotoxin cat #1128, final concentrations, all from Tocris), with a bath perfusion rate of ~2 ml/min. Dopaminergic neurons in the SNc and VTA were identified by tdTomato fluorescence. For whole-cell recordings, 2.5–6 mΩ boro-silicate glass pipettes (Sutter Instruments: BF150-86-7.5) were filled with a potassium-based internal solution containing (in mM): 135 KMeSO3, 5 KCl, 5 HEPES, 4 Mg-ATP, 0.3 Na-GTP, 10 phospho-creatine, 1 EGTA, and 4 mg/ml neurobiotin (Vector Laboratories #SP-1120). Recordings were obtained using a MultiClamp 700B amplifier (Molecular Devices) and ScanImage software (https://github.com/bernardosabatini/SabalabAcq). Passive membrane properties were recorded in voltage clamp with the membrane held at −70 mV. Negative current steps (2 s, −50 to −200 pA) were applied to measure the sag amplitude and rebound following hyperpolarization. Positive current steps (2 s,+25 to+600 pA) were applied to generate an input–output curve from a baseline membrane potential of −70 mV as maintained by current clamp. For whole-cell recordings, series resistance was <30 MΩ and liquid junction potential was not corrected.

Electrophysiology data were acquired using ScanImage software, written and maintained by Dr. Bernardo Sabatini (https://github.com/bernardosabatini/SabalabAcq). Data were analyzed in Igor Pro (Wavemetrics). Rheobase was calculated as the current at which APs were first elicited. Passive properties were calculated from an RC check in voltage clamp recordings at −70 mV.

## Fast-scan cyclic voltammetry (FCV)

DA release was monitored using fast-scan cyclic voltammetry (FCV) in acute coronal slices. Male and female mice (P56-P90) were deeply anesthetized by isoflurane and decapitated. 275-µm-thick coronal striatal slices were prepared on a vibratome (Leica VT1000 S) in ice cold high $Mg^{2+}$ ACSF containing in mM: 85 NaCl, 25 NaHCO₃, 2.5 KCl, 1.25 NaH₂PO₄, 0.5 CaCl₂, 7 MgCl₂, 10 glucose, and 65 sucrose. Slices recovered for 1 hr at RT and were recorded from in ACSF containing in mM: 130 NaCl, 25 NaHCO₃, 2.5 KCl, 1.25 NaH₂PO₄, 2 CaCl₂, 2 MgCl₂, and 10 glucose. All solutions were continuously saturated with 95% $O_2$ and 5% $CO_2$. Slices between +1.5 mm and +0.5 mm from bregma containing dorsal striatum and nucleus accumbens were used for experimentation (*Franklin and Paxinos, 2008*).

Slices from both genotypes were prepared and recorded from on the same day, the order of brain dissection was counterbalanced between experiments.

In the recording chamber, slices were maintained at 32 °C with a perfusion rate of 1.2–1.4 ml/min. Slices used for recording were anatomically matched between animals and position within the recording chamber was counterbalanced between experiments. Extracellular DA concentration ([DA]$_o$) evoked by local electrical stimulation was monitored with FCV at carbon-fiber microelectrodes (CFMs) using a Millar voltammeter (Julian Millar, Barts and the London School of Medicine and Dentistry). CFMs were fabricated in-house from epoxy-free carbon fiber ~7 μm in diameter (Goodfellow Cambridge Ltd) encased in a glass capillary (Harvard Apparatus: GC200F-10) pulled to form a seal with the fiber and cut to final tip length of 70–120 μm. The CFM was positioned ~100 μm below the tissue surface at a 45° angle. A triangular waveform was applied to the carbon fiber scanning from −0.7 V to +1.3 V and back, against an Ag/AgCl reference electrode at a rate of 800 V/s. Evoked DA transients were sampled at 8 Hz, and data were acquired at 50 kHz using AxoScope 10.5–10.7 (Molecular Devices). Oxidation currents evoked by electrical stimulation were converted to [DA]$_o$ from post-experimental calibrations. Recorded FCV signals were confirmed to be DA by comparing oxidation (+0.6 V) and reduction (−0.2 V) potential peaks from experimental voltammograms with currents recorded during calibration with 2 μM DA dissolved in ACSF.

A concentric bipolar stimulating electrode (FHC: CBAEC75) used for electrical stimulation was positioned on the slice surface with minimal tissue disturbance within 100 μm of the CFM. Following 30 min slice equilibration in the recording chamber, DA release was evoked using square wave pulses (0.6 mA pulse amplitude, 2ms pulse duration) controlled by an Isoflex stimulus isolator (A.M.P.I., Jerusalem, Israel) delivered out of phase with voltammetric scans. At each sampling site, DA release was evoked every 2.5 min. Three stimulations were delivered (in the following order: single pulse, 4 pulses at 100 Hz, single pulse) before progressing to the corresponding site in the other slice.

## High-performance liquid chromatography (HPLC)

Tissue DA content was measured by HPLC with electrochemical detection in tissue punches from dorsal and ventral striatum. Male and female mice (P60-P90) were deeply anesthetized by isoflurane and decapitated. 300-μm-thick coronal slices of striatum were prepared on a vibratome (Leica VT1000 S) in ice cold high $Mg^{2+}$ ACSF containing in mM: 85 NaCl, 25 $NaHCO_3$, 2.5 KCl, 1.25 $NaH_2PO_4$, 0.5 $CaCl_2$, 7 $MgCl_2$, 10 glucose, and 65 sucrose. Slices recovered for 1 hr at RT in ACSF containing in mM: 130 NaCl, 25 $NaHCO_3$, 2.5 KCl, 1.25 $NaH_2PO_4$, 2 $CaCl_2$, 2 $MgCl_2$, and 10 glucose. All solutions were continuously bubbled with 95% $O_2$ and 5% $CO_2$. Following slice recovery, tissue punches from the dorsal (2.5 mm diameter) and ventral striatum/NAc (1.5 mm diameter) from two brain slices per animal were taken and stored at –80 °C in 200 μl 0.1 M $HClO_4$. On the day of analysis, samples were thawed, homogenized by sonication, and centrifuged at 16,000 x g for 15 min at 4 °C. The supernatant was analyzed for DA content using HPLC with electrochemical detection. Analytes were separated using a 4.6 × 150 mm Microsorb C18 reverse-phase column (Varian or Agilent) and detected using a Decade II SDS electrochemical detector with a Glassy carbon working electrode (Antec Leyden) set at +0.7 V with respect to a Ag/AgCl reference electrode. The mobile phase consisted of 13% methanol (v/v), 0.12 M NaH2PO4, 0.5 mM OSA, 0.8 mM EDTA, pH 4.8, and the flow rate was fixed at 1 ml/min. Analyte measurements were normalized to tissue punch volume (pmol/mm$^3$). HPLC analysis was repeated in two independent experiments. Raw HPLC measurements were normalized to WT control values for each line to enable percent-change quantification of DA and DOPAC availability in KO samples. Raw summary values for HPLC are reported *Figure 8—source data 3* and *Figure 9—source data 3*.

## Western blotting (WB)

Male and female mice (P60-P90) were deeply anesthetized by isoflurane and decapitated. Bilateral striata were rapidly dissected out on ice, flash-frozen in liquid nitrogen and stored at –80 °C. On the day of analysis, frozen samples were sonicated until homogenized (QSonica Q55) in 500 μl lysis buffer containing 1% SDS in 1 x PBS with Halt phosphatase inhibitor cocktail (Fisher: PI78420) and Complete mini EDTA-free protease inhibitor cocktail (Roche: 4693159001). Sample homogenates were then boiled on a heat block at 95 °C for 10 min, allowed to cool to RT and total protein content was determined by BCA assay (Fisher: PI23227). Following the BCA assay, protein homogenates were mixed with 4 x Laemmli sample buffer (Bio-Rad: 161–0747) and 10–15 μg of protein were

loaded onto 4%–15% Criterion TGX gels (Bio-Rad: 5671084) in running buffer (3.03 g Tris base, 14.41 g glycine, 1 g SDS in 1 L ultrapure dH$_2$O). Proteins were transferred to a PVDF membrane (BioRad: 1620177) in transfer buffer (3.03 g Tris base, 14.41 g glycine in 1 L ultrapure dH$_2$O) at 4 °C overnight using a Bio-Rad Criterion Blotter (12 V constant voltage). The membranes were blocked in 5% milk in 1 x TBS with 1% Tween (TBS-T) for one hour at RT, and incubated with primary antibodies diluted in 5% milk in TBS-T overnight at 4 °C. The following day, after 3 × 10 min washes with TBS-T, the membranes were incubated with HRP-conjugated secondary antibodies (1:5,000 dilution in 5% milk in TBS-T solution) for 1 hr at RT. Following 6 × 10 min washes, the membranes were incubated with chemiluminesence substrate (Perkin-Elmer: NEL105001EA) for 1 min and signal was developed on GE Amersham Hyperfilm ECL (VWR: 95017–661). Membranes were stripped by 2 × 7 min incubations in stripping buffer (6 M guanidine hydrochloride (Sigma: G3272) with 1:150 β-mercaptoethanol) with shaking, followed by 4 × 2 min washes in 1 x TBS with 0.05% NP-40 to re-blot on subsequent days.

The following primary antibodies were used: mouse anti-Tyrosine Hydroxylase (1:2000, Immunostar: 22941); rabbit anti-DARPP32 (1:1500, Cell Signaling: 2,306 S); mouse anti-Histone-3 (1:1500, Cell Signaling: 96C10); rabbit anti-VMAT2 (1:1000, Alomone Labs: AMT-006); mouse anti-DAT (1:1000, Abcam: 128848) and rabbit anti DAT phospho-T53 (1:1000, Abcam: 183486). Secondary antibodies were goat anti-rabbit HRP (1:5000, Bio-Rad: 170–5046) and goat anti-mouse HRP (1:5000, Bio-Rad: 170–5047). Bands were quantified by densitometry using FIJI (NIH). Phospho-proteins were normalized to their respective total proteins. Total proteins were normalized to Histone 3 loading control. Uncropped western blot scans are provided in *Figure 8—source data 1 and 2* and *Figure 9—source data 1 and 2*.

## Protein-retention expansion microscopy (ProExM)

PFA fixed coronal 30 µm striatal brain sections were washed 3 × 5 min in 1 x PBS and followed by 1 hr incubation with Blockaid at RT with gentle nutation. Rabbit anti-RFP antibody (1:500; Rockland: 600-401-379) was applied at 4 °C in 1 x PBS containing 0.25% Triton-X-100 (PBS-Tx) for 24 hr. Sections were washed with 1 x PBS 5 × 5 min, incubated for 24 hr at 4 °C with a goat anti-rabbit Alexa 546 secondary antibody (1:500; Thermo: A-11035) in PBS-Tx, and washed with cold PBS 5 × 5 min.

Sections were then processed with ProExM protocol as previously described (*Asano et al., 2018*; *Tillberg et al., 2016*). Briefly, sections were incubated with anchoring solution for 16 hr at RT with no shaking. The anchoring solution consisted of 10 mg/mL acryloyl-X SE (Thermo-Fisher: A20770) dissolved in anhydrous DMSO (Thermo-Fisher: D12345) diluted to 0.1 mg/mL in 1 x PBS. Slices were then washed 2 × 15 min in 1 x PBS. Monomer solution consisting of 1 x PBS, 2 M NaCl, 8.55% (w/v) sodium acrylate (Combi-Blocks: QC-1489), 2.5% (w/v) acrylamide (Sigma: A9099), and 0.15% (w/v) N,N'-methylenebisacrylamide (Invitrogen: M7279) was frozen in aliquots and thawed at 4 °C before use. A gelling solution was made by combining monomer solution with 0.01% (w/v) 4-hydroxy-TEMPO (4HT, Sigma 176141), 0.2% (w/v) tetra-methylethylenediamine (TEMED, Sigma: T7024) and 0.2% (w/v) ammonium persulfate (APS, Sigma: A3678). All chemical concentrations refer to their final ratio in the gelling solution (monomer solution with 4HT, TEMED, and APS).

Slices were incubated in the gelling solution for 30 min at 4 °C, transferred to gelation chambers (22 × 25 x 0.15 mm) and placed in a humidity chamber for 2 hr at 37 °C for polymerization (see Figure 4 in *Asano et al., 2018* for depiction of the chamber). Gels were then placed in Proteinase K (New England Biolabs: P8107S) diluted 1:100 (8 U/mL) in digestion buffer composed of 50 mM Tris (pH 8), 1 mM EDTA, 0.5% (w/v) Triton X-100, 1 M NaCl for 16 hr at RT and then stored in 1 x PBS at 4 °C.

Post-expansion imaging of DA axons was carried out as follows. No. 1.5 coverslip strips cut to 2.75 × 0.5 mm with a diamond scriber (VWR: 52865–005) were cleaned with distilled water followed by pure ethanol before 20 min treatment with 0.1% (w/v) poly-L-lysine. Following 3 rinses with distilled water, coverslips were left to air-dry for 1 hr. Coverslips were superglued onto the inset surface of a custom-made light-sheet adapter (*Asano et al., 2018*) printed on a Form 2 (Formlabs) using standard resin. Expansion gels were incubated 6 × 15 min in distilled water, and following full expansion were trimmed to obtain regions containing dorsolateral striatum or ventromedial striatum and nucleus accumbens core (regions of interest, ROI). Trimmed gels containing ROIs were carefully placed onto the poly-L-lysine-coated coverslip strip and set aside to adhere for 5 min. Gels were imaged on a Zeiss Lightsheet Z.1 using a 20 x/1.0 N.A. water immersion objective and a single-side illumination

10 x/0.2 N.A. objective. Images were acquired with a 546 nm laser line using 0.522 μm steps to acquire a z-stack image of entire expanded region of interest.

To quantify the expansion factor, pre- and post-expansion images of the same field of view were acquired on an Olympus FV3000 confocal with a 4 x/0.16 N.A. air objective (Olympus #UPLXAPO4X). Expansion factor calculation was performed using an implementation of the scale-invariant feature transform (SIFT) algorithm (*Lowe, 2004*). OpenCV package for Python (https://github.com/opencv/opencv-python) was used to generate SIFT keypoints and estimate a partial 2D affine transformation between pre- and post-expansion keypoints, restricting image alignment to rotation, translation, and uniform scaling. Analysis code is available at https://github.com/kamodulin/expansion-microscopy. Average (mean ± SEM) expansion factors per genotype were: DA-Tsc1 WT: 4.356+/-0.022, DA-Tsc1 KO: 4.231+/-0.017, DA-Raptor WT: 4.225+/-0.014, DA-Raptor KO: 4.255+/-0.015, DA-Rictor WT: 4.344+/-0.019, DA-Rictor KO: 4.221+/-0.039.

## TrailMap axon segmentation

To identify DA axons within striatal sub-regions, expansion microscopy volumes were segmented with a convolutional neural network, TrailMap (*Friedmann et al., 2020*), with a few modifications. Three separate volumes were acquired for each ROI, two of which were held out for inference (428 volumes total). The remaining volumes were randomly assigned to training (88 volumes) or validation (25 volumes) datasets and the rest were not used. 3D sub-volumes (400 × 400 pixels) that spanned the entire depth of the original volume were randomly cropped from training and validation datasets. These volumes were sparsely annotated with a spacing of ~30–50 slices along the Z axis. From these sub-volumes, $64^3$ pixel cubes were randomly cropped to generate 8800 training and 2500 validation volumes. The model was trained with real-time data augmentation and the model with the lowest validation loss (epoch 22) was selected for inference on the main dataset. Our best trained model and a fork of the original TrailMap package with a modified data augmentation procedure are available at https://github.com/kamodulin/TRAILMAP.

TrailMap predictions were thresholded at p>0.7 to generate binary volumes and objects less than 256 voxels in size were removed. Total axon volume was calculated relative to the total image volume. Binary volumes were skeletonized in three dimensions and the total length of the axon armature was measured and normalized by the total image volume. A 3D Euclidean distance transform was computed on the binary volume for every point along the axon skeleton. The average of these distances was used to calculate the mean axon radius per image. All measurements were normalized by expansion factor. Analysis code is available at https://github.com/kamodulin/expansion-microscopy; *Ahmed, 2021*.

## Striatal patch-matrix ROI processing

PFA fixed coronal 30 μm striatal brain sections were washed 3 × 5 min in 1 x PBS and followed by 1 hr incubation with Blockaid (Life Tech - B10710) at RT with gentle rocking. Rabbit anti-mu opioid receptor (MOR) antibody (1:1000; EMD Millipore: AB5511) was applied at 4 °C in 1 x PBS containing 0.25% Triton-X-100 (PBS-Tx) for 24 hr. Sections were washed with 1 x PBS 5 × 5 min, incubated for 1 hr at RT with a goat anti-rabbit Alexa 488 secondary antibody (1:500, Thermo: A-11008) in PBS-Tx, and washed with cold 1 x PBS 5 × 5 min. Processed sections were mounted on SuperFrost slides (VWR: 48311–703), and coverslipped with Prolong Gold antifade (Life Tech: P36935) mounting media.

Striatal sections containing tdTomato labelled DA axons and patch (striosome) regions visualized by high density MOR signal were imaged on an Olympus FV3000 confocal microscope equipped with 488 and 640 nm lasers and a motorized stage for tile imaging. Z stack images captured the entire thickness of the section at 2.75 μm steps with a 10 x air/0.4 N.A. objective (UPLXAPO10X).

Patch ROIs were manually drawn around the high intensity MOR regions in FIJI (NIH). The same ROIs were then moved outside the high intensity MOR signal to get fluorescence measurements in striatal matrix. Quantification of tdTomato fluorescence intensity signal for patch and matrix ROIs was done on max-projected Z stack images.

## Quantification and statistical analyses

No explicit power analysis was used to compute samples sizes. Sample sizes were based on prior studies. Whenever possible, quantification and analyses were performed blind to genotype. Samples

were allocated into groups based on genotype. All data were included and there was no exclusion of outliers. Statistical analyses and graphing were performed using the GraphPad Prism 8 software. All datasets were first analyzed using D'Agostino and Pearson normality test, and then parametric or non-parametric two-tailed statistical tests were employed accordingly to determine significance. If the variances between two groups was significantly different, a Welch's correction was applied. Significance was set as *p<0.05, **p<0.01, ***p<0.001, and ****p<0.0001. p Values were corrected for multiple comparisons. Statistical details including sample sizes for each experiment are reported in the figure legends.

## Acknowledgements

Confocal and light sheet imaging experiments were conducted, in part, at the CRL Molecular Imaging Center, RRID:SCR_017852, supported by the Helen Wills Neuroscience Institute. We thank Holly Aaron and Feather Ives for their microscopy advice and support. We thank Tien (Mary) Chiu and Victoria Du for their assistance in maintaining the mouse colony. We thank Dr. Drew Friedmann for his advice in adapting the TrailMap method to our analysis pipeline. The Zeiss AxioImager M1 with MicroBright-Field BioSciences (MBFBiosciences) StereoInvestigator is maintained through a generous gift from Ms. Joan Lam. We thank Alexander J Ehrenberg for assistance with setting up the StereoInvestigator protocol. We thank the members of the Bateup lab for their feedback on this work.

## Additional information

### Funding

| Funder | Grant reference number | Author |
|---|---|---|
| National Institutes of Health | R01NS105634 | Helen S Bateup |
| Tuberous Sclerosis Alliance | 381490 | Polina Kosillo |
| Brain and Behavior Research Foundation | 25073 | Helen S Bateup |
| Brain and Behavior Research Foundation | 27458 | Polina Kosillo |
| Chan Zuckerberg Biohub | | Helen S Bateup |
| Clarendon Fund | | Bradley M Roberts |
| Parkinson's UK | G-1504 | Stephanie J Cragg |

The funders had no role in study design, data collection and interpretation, or the decision to submit the work for publication.

### Author contributions

Polina Kosillo, Conceptualization, Formal analysis, Funding acquisition, Investigation, Visualization, Methodology, Writing – original draft, Writing – review and editing; Kamran M Ahmed, Conceptualization, Software, Formal analysis, Investigation, Visualization, Methodology, Writing – review and editing; Erin E Aisenberg, Investigation, Writing – review and editing; Vasiliki Karalis, Investigation, Methodology, Writing – review and editing; Bradley M Roberts, Investigation; Stephanie J Cragg, Supervision, Funding acquisition, Writing – review and editing; Helen S Bateup, Conceptualization, Formal analysis, Supervision, Funding acquisition, Writing – original draft, Project administration, Writing – review and editing

### Author ORCIDs

Polina Kosillo http://orcid.org/0000-0002-1944-9460
Kamran M Ahmed http://orcid.org/0000-0002-7204-5991
Bradley M Roberts http://orcid.org/0000-0002-5192-2545
Stephanie J Cragg http://orcid.org/0000-0001-9677-2256
Helen S Bateup http://orcid.org/0000-0002-0135-0972

### Ethics

All animal procedures and husbandry were carried out in accordance with protocols approved by the University of California, Berkeley Institutional Animal Care and Use Committee (IACUC, protocol #AUP-2016-04-8684-2).

### Decision letter and Author response

Decision letter https://doi.org/10.7554/eLife.75398.sa1
Author response https://doi.org/10.7554/eLife.75398.sa2

---

# Additional files

### Supplementary files

• Transparent reporting form

### Data availability

All data generated or analysed during this study are included in the manuscript and supporting files. Source data files have been provided for Figures 8 and 9.

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
