## [Editor Report]

This manuscript by Kosillo and colleagues presents a series of carefully carried out experiments evaluating the impact of perturbing the mTORC1 and mTORC2 protein complexes selectively in mouse dopamine neurons. By utilizing dopamine neuron-specific Raptor and Rictor cKO mice, this paper elucidated which of these mTOR complexes are responsible for the regulation of dopamine neuronal functions, revealing the importance of mTORC1/2 signaling for the structure and function of dopamine neurons. This paper provided comprehensive data including structural, physiological, and biochemical alterations by genetic deletion of Raptor/Rictor in dopamine neurons.

---

## [Decision Letter]

**Decision letter after peer review:**

Thank you for submitting your article "Dopamine neuron morphology and output are differentially controlled by mTORC1 and mTORC2" for consideration by *eLife*. Your article has been reviewed by 3 peer reviewers, and the evaluation has been overseen by a Reviewing Editor and Lu Chen as the Senior Editor. The following individuals involved in review of your submission have agreed to reveal their identity: Louis-Eric Trudeau (Reviewer #2); Jae-Ick Kim (Reviewer #3).

Essential revisions:

The reviewers have raised the following suggestions, and the reviewers agree that these are essential for the revision:

1. The authors conclude that there is no change in the number of dopamine neurons in the two lines of conditional KO mice. This should be done with unbiased stereological counting.

2. Provide more support, or tune down the conclusion – "mTORC2 inhibition leads to distinct cellular changes not observed following mTORC1 suppression, suggesting some independent actions of the two mTOR complexes in DA neurons."

3. Improve introduction and discussion: modify some inconsistent statements, cut down reiterations of the results in the discussion, provide additional discussion on mTORC1 vs mTORC2, quantification, etc.

The suggestions on DAT staining and behavioral consequences are excellent, but the reviewers agree that these data are not essential for the revision.

*Reviewer #1 (Recommendations for the authors):*

I have the following specific comments:

1. The authors state that "Together, these data show that mTORC2 suppression reduces mTORC1 activity in DA neurons, which is associated with a small but significant decrease in soma size. In contrast to mTORC1 inhibition, suppression of mTORC2 increases proximal dendrite branching." However, in the Sholl analysis shown in Figure 1p, WT peaks at 8 intersections approximately 100 μm from the soma, while in Figure 2n, the same value for WT is 6 intersections. The example WT neurons in Figure 2m also have much smaller dendritic arbors as compared to WT neurons shown in Figure 1o, and are clumped together, which is not typical of DA neurons.

2. Related to the above, a change in just proximal dendrite branching (as opposed to overall reduction in dendritic arbor) is typically due to reduced number of primary dendrites. Do the authors see this phenotype?

3. In Figure 1r, the KO neuron examples have about half the dendritic arbor of WT neurons, while the summary data in Figure 1t only show a roughly 20% reduction.

4. In SFigure 2b, the DA-Tsc1 KO dl striatum image is much dimmer than WT. This image is inconsistent with the increased axon measurements shown in SFig2 c-e.

5. To better understand the data presented in Figure 3h-3j, the authors need to provide data examining differences between axon density, volume and radius in WT mice between patch and matrix.

6. The electrophysiological measurements reported in Figure 5 (Rictor KO) are in general in the same direction as those of Figure 4 (Raptor KO), albeit of smaller magnitude. This does not fit with the increased proximal dendritic arbor size reported in Figure 2. I also note that in Kosillo et al., NC, 2019, Tsc1 KO mice with more proximal dendrites have large SNc capacitance.

*Reviewer #2 (Recommendations for the authors):*

1. The authors conclude that there is no change in the number of dopamine neurons in the two lines of conditional KO mice. However, this is based on quantification of cell number in a very small subset of mice and using a technique this is not state of the art. The authors need to perform unbiased stereological counting. This is only technique that can reliably detect small changes in dopamine neuron number.

2. In the first paragraph of the introduction, the authors also state that dopamine neuron differentiation is unaltered. This is not an adequate statement, especially based on the data shown in the rest of the manuscript. The sentence should be modified.

3. In the discussion of the manuscript, the authors point out the surprising observation of increased axonal density, but reduced dopamine release in the Tsc-1 KO mice. The authors may also find it relevant to point out that this is a phenotype similar to that previously reported in conditional D2 KO mice, in which dopamine release is also similarly reduced, but axonal density increased (PMID: 31449520).

4. There are some words missing in the first sentence of the paragraph in the discussion on the use of expansion microscopy. In this same section, the authors mention that the approach of simply quantifying the density of TH signal is not appropriate because TH levels are subject to regulation. This is certainly true. But the authors may perhaps wish to mention that this problem would be seen for quantification of reporter protein expression (such as tdTomato expression driven by DAT-Cre).

*Reviewer #3 (Recommendations for the authors):*

I think the conclusion of this paper is unbiased, just reflecting the data presented, and the major findings are significant enough to be published in this journal. I have just several comments as follows.

1. It would be much better if the authors can provide direct (or indirect) evidence explaining how the deletion of Raptor can lead to TH/DAT/p-DAT/structural changes in dopamine neurons? The authors don't have to show all of these evidence. Experiments targeting just one or two of these alterations would be enough.

2. Given that the authors related the major findings of this manuscript to brain disorders such as PD, what are the behavioral consequences (especially motor functions) caused by the genetic deletion of Raptor in dopamine neurons?

3. Checking the level of p-473 Akt is the only way by which we can validate the inhibition of mTORC2 signaling in DA-Rictor KO mice? Are there any alternative, exclusive downstream targets for mTORC2 signaling?

[Editors' note: further revisions were suggested prior to acceptance, as described below.]

Thank you for resubmitting your work entitled "Dopamine neuron morphology and output are differentially controlled by mTORC1 and mTORC2" for further consideration by *eLife*. Your revised article has been evaluated by Lu Chen (Senior Editor) and a Reviewing Editor.

The manuscript has been improved but there are some remaining issues that need to be addressed, as outlined below:

The reviewers did not request additional experiments. However, a few points need to be clarified. In addition, a couple of statements need to be toned down. Please see Reviewer #2's detailed comments.

*Reviewer #1 (Recommendations for the authors):*

The authors have sufficiently addressed my concerns. I have no further comments.

*Reviewer #2 (Recommendations for the authors):*

The authors have done a good job at revising the manuscript.

The results presented globally support previous work documenting a role of the mTOR pathway in cell growth and differentiation. As such, conceptually, the results are not especially surprising. But the work is thorough and provides a complete overview of the impact of interfering with the mTOR1 and mTOR2 pathways on the structure and function of dopamine neurons.

A few additional points to consider:

1. In the abstract, the authors state that mTOR is involved in brain disorders including Parkinson's. This is a strong statement. The introduction of the manuscript does not refer to evidence showing direct involvement of deficits in this pathway in PD. There are some hypotheses that link this pathway to mechanisms related to PD, but direct involvement has not been established. I suggest that such statements be toned down in the manuscript.

2. The way the results are presented in the section that relates to a difference in the density of dopamine neuron axon terminals in the patch versus matrix is not very clear. The authors state that they could not quantify the density of TdTomato-positive axon terminals in the two compartments in the WT mice because the signal was too homogeneous. What they mean by this needs to be clarified. Normally, a simple TH staining shows quite clear distinctions between the patch and matrix sectors of the striatum. Why was this not the same for the TdTomato signal in the WT mice? Without a comparison with WT mice, it is difficult to conclude that there was a larger decrease in axon terminal density in the matrix, as suggested by the authors. If additional data on WT mice is not provided, this conclusion may need to be reconsidered.

3. In figure 4, reporting the results comparing DA neuron excitability, it is very surprising that panel 4O reports a significant increase in rheobase for the VTA neurons. The data show quite clearly that only one data point was higher for the KO cells. I am very surprised that this comes out significantly. Perhaps the authors should double-check their statistical analysis.

4. In the description of the results of figure 8, the authors state that "The ratio of p-DAT to total DAT was increased (Figure 8c), suggesting that the remaining DAT is functional.". What this sentence really means is unclear. The authors should specify what Thr53 phosphorylation means for DAT function. As far as I know, this phosphorylation is typically associated with increased DAT activity. So, would the observation of an increase in the ratio of phospho-DAT to total DAT not imply an increased relative DAT activity in the KO mice? If so, this would be counterintuitive considering the authors' results showing a slowing down of reuptake in the KO mice. This needs to be clarified.

*Reviewer #3 (Recommendations for the authors):*

I don't have any further comments about the revised manuscript. I'm satisfied with the revised manuscript and the authors' responses. I think the manuscript is now ready for publication.

---

## [Author Response]

Essential revisions:The reviewers have raised the following suggestions, and the reviewers agree that these are essential for the revision:1. The authors conclude that there is no change in the number of dopamine neurons in the two lines of conditional KO mice. This should be done with unbiased stereological counting.

We have now performed unbiased stereological counting of midbrain dopamine neurons in the DA-Raptor and DA-Rictor conditional knock-out (KO) mouse lines. We find that deletion of *Rictor* does not affect the number of tdTomato+ DA neurons in the midbrain of *Rictor^fl/fl^;DATCre+/-;Ai9+/-* mice compared to *Rictor^wt/wt^;DAT-Cre+/-;Ai9+/-* (WT) controls. However, we do find a significant reduction in the number of tdTomato+ neurons in both the SNc and VTA of DA-Raptor KO mice. This data is included in revised Figure 1—figure supplement 1.

2. Provide more support, or tune down the conclusion – "mTORC2 inhibition leads to distinct cellular changes not observed following mTORC1 suppression, suggesting some independent actions of the two mTOR complexes in DA neurons."

We agree that the majority of changes observed in Rictor cKO dopamine neurons are mild versions of the Raptor cKO phenotypes, which may be driven by the moderate suppression of mTORC1 signaling we observed with *Rictor* deletion (see Figure 2i,k). However, we do find differential effects on proximal dendrite branching within ~50-100 µm of the soma, as measured by Sholl analysis. In this case, the number of intersections is increased in DA-Rictor KO neurons but decreased in DA-Raptor KO cells (see Figures 1o-r and 2m-p). We also find that while loss of Raptor impairs dopamine release in both the dorsal and ventral striatum, deletion of *Rictor* only affects release in the central and ventral striatum. Given that these phenotypes are relatively minor, we have toned down the conclusion that “mTORC2 inhibition leads to distinct cellular changes”, as suggested by the reviewers. Please see the revised Discussion section in the manuscript.

3. Improve introduction and discussion: modify some inconsistent statements, cut down reiterations of the results in the discussion, provide additional discussion on mTORC1 vs mTORC2, quantification, etc.

We have significantly revised the introduction and discussion per the reviewers’ suggestions. Please see specific comments below.

The suggestions on DAT staining and behavioral consequences are excellent, but the reviewers agree that these data are not essential for the revision.

We appreciate that these research directions are interesting; however, we believe that they are beyond the scope of the current study. Please see our responses to the individual reviewer comments below.

Reviewer #1 (Recommendations for the authors):I have the following specific comments:1. The authors state that "Together, these data show that mTORC2 suppression reduces mTORC1 activity in DA neurons, which is associated with a small but significant decrease in soma size. In contrast to mTORC1 inhibition, suppression of mTORC2 increases proximal dendrite branching." However, in the Sholl analysis shown in Figure 1p, WT peaks at 8 intersections approximately 100 μm from the soma, while in Figure 2n, the same value for WT is 6 intersections. The example WT neurons in Figure 2m also have much smaller dendritic arbors as compared to WT neurons shown in Figure 1o, and are clumped together, which is not typical of DA neurons.

The reviewer is correct that we do see differences in the properties of WT dopamine neurons across the two transgenic strains. This is why we always compare WT and KO mice from the same transgenic line, using littermates when possible. We and others have previously observed significant differences in dopamine neuron properties across different mouse lines and strains, which is likely driven by differences genetic background (for example, see PMIDs: 33979604 and 23524098). Compared to WT mice from the same line, we find that *Rptor* deletion reduces overall dendritic arborization. By contrast, *Rictor* deletion increases proximal dendritic branching without affecting the number of primary dendrites or total dendritic length. We have updated our example images in Figure 2m with more representative cells.

2. Related to the above, a change in just proximal dendrite branching (as opposed to overall reduction in dendritic arbor) is typically due to reduced number of primary dendrites. Do the authors see this phenotype?

We have quantified the number of primary dendrites and find that this is not significantly altered by deletion of *Rptor* or *Rictor.* This data is included in the revised Figures 1 and 2.

3. In Figure 1r, the KO neuron examples have about half the dendritic arbor of WT neurons, while the summary data in Figure 1t only show a roughly 20% reduction.

While we do find some DA-Raptor KO VTA neurons with strongly reduced dendritic arbors, we have updated the example images to show cells that are more similar to the group mean.

4. In SFigure 2b, the DA-Tsc1 KO dl striatum image is much dimmer than WT. This image is inconsistent with the increased axon measurements shown in SFig2 c-e.

The reviewer is correct that the endogenous tdTomato fluorescence in striatal dopamine axons is consistently lower in Tsc1 KO mice compared to WT. We have also observed reduced tdTomato fluorescence in Tsc1 KO somas, whereas tyrosine hydroxylase expression is increased in DA-Tsc1 KO neurons in both compartments. These types of fluctuations in fluorescence intensity for different markers can indeed complicate quantification, which is why we used TrailMap to segment the images and identify DA axons. This allows us to accurately quantify the volume, density and radius of DA axons without bias from the intensity of fluorescent markers. We have included example segmented images of striatal DA axons for DATsc1 KO and WT mice in Figure 3—figure supplement 1 to highlight this point.

5. To better understand the data presented in Figure 3h-3j, the authors need to provide data examining differences between axon density, volume and radius in WT mice between patch and matrix.

We agree that this would be nice data to have; however, we do not have a way to reliably distinguish patch and matrix compartments in WT mice in the ProExM experiments. In the DARaptor KO striatum, we see clearly observable “patches” with higher density of tdTomato-expressing DA axons (see Figure 3f). Using IHC and confocal imaging, we find that these tdTomato-dense patches correspond to striatal patches defined by high mu opioid receptor (MOR) expression (see Figure 3—figure supplement 1g). We therefore separately quantified DA axon properties in these two compartments in DA-Raptor KO mice. In all other genotypes, we find that the tdTomato signal is relatively uniform throughout the dorsal striatum and we cannot determine whether our field of view is within a patch or matrix region (see top left panel in Figure 3—figure supplement 1g showing WT tdTomato expression). The MOR antibody used for IHC has not yet been optimized or validated for use with expansion microscopy. However, we do show with confocal imaging that the ratio of tdTomato fluorescence in patch versus matrix (as defined by MOR staining) is significantly higher in DA-Raptor KO striatum compared to WT (see new Figure 3—figure supplement 1j).

6. The electrophysiological measurements reported in Figure 5 (Rictor KO) are in general in the same direction as those of Figure 4 (Raptor KO), albeit of smaller magnitude. This does not fit with the increased proximal dendritic arbor size reported in Figure 2. I also not that in Kosillo et al., NC, 2019, Tsc1 KO mice with more proximal dendrites have large SNc capacitance.

In DA-Rictor KO neurons, we find significantly decreased soma size and no change in total dendritic length. Therefore, we believe the small reduction in membrane capacitance is likely driven by the change in cell body size, which is in the same direction as in DA-Raptor KO neurons. We find that the small but significant increase in membrane resistance in Rictor KO VTA neurons is sufficient to reduce rheobase and increase excitability to positive current injection. In the Kosillo et al., 2019 study (PMID: 31780742), we observed an increase in both cell body size and total dendritic length, which drove a strong increase in membrane capacitance, reduction in resistance, and reduction in intrinsic excitability.

Reviewer #2 (Recommendations for the authors):1. The authors conclude that there is no change in the number of dopamine neurons in the two lines of conditional KO mice. However, this is based on quantification of cell number in a very small subset of mice and using a technique this is not state of the art. The authors need to perform unbiased stereological counting. This is only technique that can reliably detect small changes in dopamine neuron number.

We appreciate the reviewer’s suggestion. We have performed unbiased stereological counting of DA neurons, based on published methods. We find that deletion of *Rictor* does not affect the number of tdTomato+ DA neurons in the midbrain of *Rictor^fl/fl^;DAT-Cre+/-;Ai9+/-* mice compared to *Rictor^wt/wt^;DAT-Cre+/-;Ai9+/-* (WT) controls. However, we do find a significant reduction in the number of tdTomato+ neurons in both the SNc and VTA of DA-Raptor KO mice. This data is included in revised Figure 1—figure supplement 1.

2. In the first paragraph of the introduction, the authors also state that dopamine neuron differentiation is unaltered. This is not an adequate statement, especially based on the data shown in the rest of the manuscript. The sentence should be modified.

We have revised this sentence and updated our conclusions based on the new stereology analysis.

3. In the discussion of the manuscript, the authors point out the surprising observation of increased axonal density, but reduced dopamine release in the Tsc-1 KO mice. The authors may also find it relevant to point out that this is a phenotype similar to that previously reported in conditional D2 KO mice, in which dopamine release is also similarly reduced, but axonal density increased (PMID: 31449520).

We thank the reviewer for pointing out this study. We have revised the discussion significantly and mentioned the D2 KO study in the context of the DA Tsc1-KO axonal phenotypes.

4. There are some words missing in the first sentence of the paragraph in the discussion on the use of expansion microscopy. In this same section, the authors mention that the approach of simply quantifying the density of TH signal is not appropriate because TH levels are subject to regulation. This is certainly true. But the authors may perhaps wish to mention that this problem would be seen for quantification of reporter protein expression (such as tdTomato expression driven by DAT-Cre).

We agree and have revised this section as suggested (see also response to Reviewer #1, point #4 in the recommendations to the authors).

Reviewer #3 (Recommendations for the authors):I think the conclusion of this paper is unbiased, just reflecting the data presented, and the major findings are significant enough to be published in this journal. I have just several comments as follows.1. It would be much better if the authors can provide direct (or indirect) evidence explaining how the deletion of Raptor can lead to TH/DAT/p-DAT/structural changes in dopamine neurons? The authors don't have to show all of these evidence. Experiments targeting just one or two of these alterations would be enough.

We agree that investigating the mechanistic basis for the changes in TH, DAT and neuronal morphology are important avenues for future studies. Regarding TH, we hypothesize that it is under translational control by mTORC1 as hyperactivation of mTORC1 increases TH levels and DA tissue content (PMID: 31780742), while chronic suppression of mTORC1 reduces TH and DA levels (Figure 8a,b). In the case of DAT, it is possible that it is also under translational control by mTORC1 as its protein levels and function are reduced in DA-Raptor KO mice (Figure 8a,b) but increased in DA-Tsc1 KO mice (PMID: 31780742). Alternatively, DAT expression may be reduced in DA-Raptor KO neurons as a compensatory mechanism to boost and prolong DA actions in the face of reduced synthesis. This could potentially be resolved in future studies by examining the developmental time course of changes in DA production and re-uptake in DA-Raptor KO mice.

In terms of the morphological changes, while numerous studies have demonstrated structural alterations in many cell types with mTOR manipulations, to our knowledge, the downstream effectors responsible are largely unknown. Prior studies have implicated mTORC1-depedent translation, via regulation of its downstream targets p70S6K and 4E-BP1, in the control of dendritic complexity during development (PMID: 16339025). In addition, mTORC2 is known to regulate PKC phosphorylation and total levels and PKC, which has also been shown to regulate dendrite branching (PMIDs: 15217371 and 23569215). In the discussion we speculate that the dramatic changes in dopamine neuron size (across all subcellular domains examined) associated with mTORC1 up- or down-regulation could arise due to changes in protein and lipid synthesis, chronic activation of autophagy, and/or changes to the cytoskeleton.

2. Given that the authors related the major findings of this manuscript to brain disorders such as PD, what are the behavioral consequences (especially motor functions) caused by the genetic deletion of Raptor in dopamine neurons?

We agree that this would be a very interesting future direction; however, behavioral analysis is beyond the scope of this study, which focuses on cellular and sub-cellular structural and functional changes.

3. Checking the level of p-473 Akt is the only way by which we can validate the inhibition of mTORC2 signaling in DA-Rictor KO mice? Are there any alternative, exclusive downstream targets for mTORC2 signaling?

p-473 Akt is the most well-characterized read-out for mTORC2 activity; however, we were not able to get a reliable signal by immunostaining. We have used the same *Rictor* conditional knock-out mice for another study (in progress), in which we examine mTORC1 and mTORC2 signaling dynamics in hippocampal cultures using western blotting. For reference, we see strong downregulation of p-Akt in Rictor KO hippocampal cultures treated with AAV-Cre, validating this mouse model as effective for manipulating mTORC2 activity (see Author response image 1). Notably, in Rictor KO hippocampal cultures we also find moderately reduced p-S6, similar to what we observed here for dopamine neurons, suggesting that inhibition of mTORC2 inhibition can reduce mTORC1 activity in multiple cell types.

**Author response image 1. sa2fig1:** Conditional deletion of Rictor suppresses Akt and S6 phosphorylation. Primary hippocampal cultures were prepared from Rictorfl/fl mice and AAV-GFP (WT) or AAV-Cre (KO) was added on DIV 2. Cultures were harvested for western blotting on DIV 14. (a) Representative western blots for the indicated proteins (two independent samples are shown). (b-d) Quantification of western blots (mean +/- SEM) for Rictor (b), p-473 Akt normalized to total Akt (c) and p-240/244 S6 normalized to total S6 (d). Dots represent individual culture wells. ****, p<0.0001, Mann-Whitney tests. n=11 Rictor WT and 14 Rictor KO samples.

[Editors' note: further revisions were suggested prior to acceptance, as described below.]

Reviewer #1 (Recommendations for the authors):The authors have sufficiently addressed my concerns. I have no further comments.

We thank the reviewer for their time and positive evaluation of our work.

Reviewer #2 (Recommendations for the authors):The authors have done a good job at revising the manuscript.The results presented globally support previous work documenting a role of the mTOR pathway in cell growth and differentiation. As such, conceptually, the results are not especially surprising. But the work is thorough and provides a complete overview of the impact of interfering with the mTOR1 and mTOR2 pathways on the structure and function of dopamine neurons.

We appreciate the reviewer’s careful review and positive comments on our work. While some of the results we find with mTORC1 suppression are indeed consistent with prior studies in other cell types, we provide an extensive characterization of a number of DA neuron properties in DA-Raptor and DA-Rictor KO mice (somatic, dendritic and axonal architecture, intrinsic excitability, DA release and reuptake properties, DA synthesis capacity, and more) that have not previously been investigated. In addition, we establish an experimental and computational workflow for analyzing the properties of individual DA axons within the striatum using expansion microscopy. We therefore hope that this study will be an important resource for the field.

A few additional points to consider:1. In the abstract, the authors state that mTOR is involved in brain disorders including Parkinson's. This is a strong statement. The introduction of the manuscript does not refer to evidence showing direct involvement of deficits in this pathway in PD. There are some hypotheses that link this pathway to mechanisms related to PD, but direct involvement has not been established. I suggest that such statements be toned down in the manuscript.

We appreciate the reviewer’s point and have removed this sentence from the abstract. The other parts of the manuscript that discuss mTOR in the context of PD cite studies showing that activation or inhibition of mTOR, either pharmacologically or genetically, can be beneficial in PD models. We agree that a causal link for mTOR in the pathophysiology of PD has not been established although several of the cellular processes thought to be deregulated in PD (e.g. autophagy, mitochondria function, lysosome function) are known to be controlled by mTOR signaling.

2. The way the results are presented in the section that relates to a difference in the density of dopamine neuron axon terminals in the patch versus matrix is not very clear. The authors state that they could not quantify the density of TdTomato-positive axon terminals in the two compartments in the WT mice because the signal was too homogeneous. What they mean by this needs to be clarified. Normally, a simple TH staining shows quite clear distinctions between the patch and matrix sectors of the striatum. Why was this not the same for the TdTomato signal in the WT mice? Without a comparison with WT mice, it is difficult to conclude that there was a larger decrease in axon terminal density in the matrix, as suggested by the authors. If additional data on WT mice is not provided, this conclusion may need to be reconsidered.

We apologize that this section was not clear, we have now revised it for clarity and provide more context below. We have also renumbered the Videos to align with the text revisions.

In our hands, we do not see distinct patch/matrix compartments with TH staining or with tdTomato reporter expression in DAT-Cre or DAT-Flp mice. The relatively uniform distribution of these markers in young adult (2-4 months old) mice across different lines can be seen in the current study in Figure 3-Figure Sup.1g, as well as in Figure 5f,o in Kosillo et al., 2019 (PMID: 31780742) and in Figures 3k and 4d in Kramer et al., 2021 (PMID: 33979604). It is possible that some lines may show more distinct TH patterning, but in our hands, we are not able to reliably distinguish patch/matrix without an additional marker.

We did not a priori expect the DA-Raptor KO mice to show differential expression of tdTomato in the patch/matrix compartments. However, in the course of imaging sections from this line on the lightsheet microscope, we could clearly observe regions with brighter tdTomato signal in the dorsal striatum. We therefore decided to take separate images of the tdTomato “bright” and “dim” regions in the DA-Raptor KO DLS sections. No other genotypes had clearly discernable bright or dim regions of tdTomato expression. We confirmed that the tdTomato bright regions in the dorsal striatum of DA-Raptor KO mice corresponded to bona fide patch regions based on mu opioid receptor (MOR) staining using confocal microscopy (see Figure 3-Figure Sup.1g). We therefore refer to them as patches in the manuscript.

In Figure 3-Figure Sup.1i we defined ROIs of patches by MOR staining with conventional confocal microscopy and measured the intensity of tdTomato expression in these regions compared to equivalently sized ROIs in the neighboring matrix regions. We found a ratio of 1.065 +/- 0.0131 in DA-Raptor WT mice suggesting similar tdTomato fluorescence in patches and matrix (with a slight increase in patches). However, this ratio was significantly larger in DA-Raptor KO mice (1.209 +/- 0.0109, p<0.0001, Welch’s t-test) indicating differential innervation of patch versus matrix in this genotype (Figure 3-Figure Sup.1i). Due to this large difference, we were able to distinguish these two compartments with tdTomato in ProExM expanded sections from DA-Raptor KO mice and image them separately to quantify axonal properties (Figure 3g-j). For Figure 3g-j, our quantitative comparison of axons in patch vs. matrix compartments was only done within DA-Raptor KO mice and not between KO and WT mice. For Figure 3c-e, we averaged values obtained from the patch and matrix regions within each DA-Raptor KO section and compared that to values obtained from WT sections, which contain both patch and matrix. We concluded that in the DLS of DA-Raptor KO mice, there is overall reduced DA axonal innervation compared to WT, and that matrix regions have less innervation than patch regions in KO mice.

3. In figure 4, reporting the results comparing DA neuron excitability, it is very surprising that panel 4O reports a significant increase in rheobase for the VTA neurons. The data show quite clearly that only one data point was higher for the KO cells. I am very surprised that this comes out significantly. Perhaps the authors should double-check their statistical analysis.

We have checked the statistics for this graph and confirm that the P value is 0.0190 using a two-tailed Mann-Whitney test. This analysis is reporting a significant decrease in rheobase in Raptor KO VTA neurons (WT median = 62.50 vs. KO median = 50.00). To see this more clearly, we plotted the data without the outlier in the KO group at 400 pA (see graph in Author response image 2). If we analyze this dataset, there is still a significant reduction in rheobase in DA-Raptor KO cells (p=0.0066, two-tailed Mann-Whitney test; WT median = 62.50 vs. KO median = 50.00). Since we don’t have a clear reason to exclude the 400pA value in the KO group, we have chosen to keep it in the dataset in the manuscript.

4. In the description of the results of figure 8, the authors state that "The ratio of p-DAT to total DAT was increased (Figure 8c), suggesting that the remaining DAT is functional.". What this sentence really means is unclear. The authors should specify what Thr53 phosphorylation means for DAT function. As far as I know, this phosphorylation is typically associated with increased DAT activity. So, would the observation of an increase in the ratio of phospho-DAT to total DAT not imply an increased relative DAT activity in the KO mice? If so, this would be counterintuitive considering the authors' results showing a slowing down of reuptake in the KO mice. This needs to be clarified.

We appreciate that this may have been confusing since we see changes that go in opposite directions. The main effect we observe is a strong down-regulation of total DAT protein in the striatum of DA-Raptor KO mice (Figure 8a,b). This down-regulation of total DAT is consistent with the slower DA re-uptake we observed in the DLS of these mice by FCV (Figure 6d). We also examined the phosphorylation of DAT as phosphorylation at this site has been shown to affect DAT function and, as the reviewer points out, is typically thought to reflect increased DAT activity. We find a relative increase in the phosphorylation state of the remaining DAT protein (i.e. the ratio of phosphorylated DAT to total DAT) in DA-Raptor KO mice. However, given our FCV results, this relative increase in phosphorylation is not sufficient to counteract the significant loss of total DAT protein. We have revised this section to make this clearer.

Reviewer #3 (Recommendations for the authors):I don't have any further comments about the revised manuscript. I'm satisfied with the revised manuscript and the authors' responses. I think the manuscript is now ready for publication.

We thank the reviewer for their time and positive evaluation of our work.